 Select

# Quantum analytic Langlands correspondence

**Davide Gaiotto[1] and Jörg Teschner[2,3]**

**1** Perimeter Institute for Theoretical Physics,
31 Caroline Street North, Waterloo, ON N2L 2Y5, Canada
**2** Department of Mathematics, University of Hamburg,
Bundesstrasse 55, 20146 Hamburg, Germany
**3** Deutsches Elektronen-Synchrotron DESY,
Notkestr. 85, 22607 Hamburg, Germany

## Abstract

The analytic Langlands correspondence describes the solution to the spectral problem for the quantised Hitchin Hamiltonians. It is related to the S-duality of $\mathcal{N} = 4$ super Yang-Mills theory. We propose a one-parameter deformation of the Analytic Langlands Correspondence, and discuss its relations to quantum field theory. The partition functions of the $H_3^+$ WZNW model are interpreted as the wave-functions of a spherical vector in the quantisation of complex Chern-Simons theory. Verlinde line operators generate a representation of two copies of the quantised skein algebra on generalised partition functions. We conjecture that this action generates a basis for the underlying Hilbert space, and explain in which sense the resulting quantum theory represents a deformation of the Analytic Langlands Correspondence.

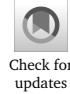

# 1 Introduction

A *complex integrable system* is a complex symplectic phase space equipped with a maximal collection of commuting holomorphic Hamiltonians $H_r$, such that generic orbits of the corresponding flows are compact complex tori. The geometric structures associated to complex integrable systems can encode the low-energy physics of four-dimensional $\mathcal{N} = 2$ supersymmetric field theories [1–3]. The complex integrable systems represent moduli spaces of vacua of four-dimensional $\mathcal{N} = 2$ gauge theories compactified on a circle in this context [4].

The moduli spaces of Higgs bundles $\mathcal{M}_H$ are prototypical examples of complex integrable systems. They occur in the context of theories of class $\mathcal{S}$ [5], but also as moduli spaces of classical vacua of four-dimensional $\mathcal{N} = 4$ gauge theories compactified on a Riemann surface [6]. This observation provides a starting point for the study of relations between four-dimensional gauge theory and the Langlands correspondence [7].

The problem of quantizing complex integrable systems has a rich history. An important first step to the quantisation of the Hitchin system has been taken in the work of Beilinson and Drinfeld on the geometric Langlands correspondence [8], leading to the construction of differential operators $\mathsf{H}_r$ on the moduli spaces $\mathrm{Bun}_G$ of holomorphic $G$-bundles quantising the Hitchin Hamiltonians. The theory of affine Lie algebras at critical level was the basis of this construction. From the perspective of the quantisation of the Hitchin system, the geometric Langlands correspondence implies that the eigenvalues $h_r$ of $\mathsf{H}_r$ correspond to values of coordinates on the spaces of opers associated to $^L\mathfrak{g}$, the Langlands dual of the Lie algebra $\mathfrak{g}$ of $G$. The spaces of opers form distinguished half-dimensional Lagrangian subspaces in the moduli spaces of $G$-bundles $E$ with holomorphic connection $\nabla$.

Important aspects of the geometric Langlands correspondence have been related to the $\mathcal{N} = 4$ supersymmetric Yang-Mills theories in [7]. Mirror symmetry of the Hitchin moduli spaces reflects the S-duality of this supersymmetric QFT theory compactified on a Riemann surface.

Complex integrable systems are characterised by a geometrical structure called special geometry [3]. The data characterising special geometry can be encoded into a collection of locally defined coordinate functions $\boldsymbol{a} = (a_1, \ldots, a_d)$, together with holomorphic functions $\mathcal{F}(\boldsymbol{a})$. Generalised action coordinates can be obtained as $a_k^\mathrm{D} = \partial_{a_k} \mathcal{F}(\boldsymbol{a})$.

The classical orbits of the integrable system can be quantized semi-classically, predicting a discrete spectrum of eigenvalues $h_r$ for the quantum Hamiltonians. The equations determining semi-classical spectra take a particularly simple form in terms of the action coordinates. For complex integrable systems which are complexifications of integrable systems defined over the real numbers, for example, one may express the quantisations conditions with the help of the generalised action coordinates in the form $a_k^\mathrm{D} \in i\hbar\mathbb{Z}$, $k = 1, \ldots, d$.[1]

Going beyond the semi-classical approximation is usually very hard. The relations to QFT mentioned above offer valuable hints. Four-dimensional $\mathcal{N} = 2$-supersymmetric QFTs admit natural deformations called $\Omega$-deformation [9]. The partition functions $\mathcal{Y}(\boldsymbol{a}, \epsilon)$ in the presence of a partial $\Omega$-deformation with a single parameter $\epsilon$ deform the function $\mathcal{F}(\boldsymbol{a})$ in a natural way, in the sense that $\mathcal{Y}(\boldsymbol{a}, \epsilon) = \frac{1}{\epsilon}(\mathcal{F}(\boldsymbol{a}) + \mathcal{O}(\epsilon))$. Investigation of effective two-dimensional descriptions of these four-dimensional $\mathcal{N} = 2$ SUSY QFT has motivated the conjecture [10] that the extremisation conditions $\partial_{a_k} \mathcal{Y}(\boldsymbol{a}, \epsilon) \in i\hbar\mathbb{Z}$ describe the exact quantisation conditions of the quantum theory obtained by quantising the classical integrable system characterised by the functions $\mathcal{F}(\boldsymbol{a})$, assuming that the parameter $\epsilon$ is equal to the quantisation parameter $\hbar$.

The partition function $\mathcal{Y}(\boldsymbol{a}, \epsilon)$ can be characterised geometrically as the generating function of the Lagrangian submanifold of opers [11].[2] This result can be explained using the mirror symmetry of Hitchin's moduli spaces [13].

In quantum integrable models one usually identifies the quantisation conditions as consequences of the square-integrability of the eigenfunctions of the commuting Hamiltonians. It has been verified in the example of the pure $\mathcal{N} = 2$ gauge theory with gauge group $U(N)$ that the conditions $\partial_{a_k} \mathcal{Y}(\boldsymbol{a}, \epsilon) \in i\hbar\mathbb{Z}$ indeed describe the spectrum of the closed Toda chain [14], as predicted in [10]. The corresponding classical integrable systems are real slices of the Hitchin systems associated to the Riemann sphere with two irregular singularities. However, for general Hitchin systems it was not clear for a while how to translate square-integrability conditions of the solutions to the eigenvalue equations $\mathsf{H}_r\Psi = E_r\Psi$ into extremisation conditions.

As a step in this direction, reference [15] investigated the more basic quantisation condition implied by single-valuedness of the solutions $\Psi$ on the whole $\mathrm{Bun}_G$ to the pairs of eigenvalue

---

[1]We are here assuming convenient normalisations making the formulae as simple as possible. The conventions used later will differ slightly.

[2]Combining the AGT-correspondence with CFT-arguments had independently led to an equivalent characterisation in [12].

equations $\mathsf{H}_r \Psi = E_r \Psi$ and $\bar{\mathsf{H}}_r \Psi = \bar{E}_r \Psi$, $r = 1, \ldots, d$, with $\bar{\mathsf{H}}_r$ being the complex conjugate of $\mathsf{H}_r$. It has been argued in [15] that the solutions to the single-valuedness condition are in one-to-one correspondence to real opers, opers with real holonomy. This condition can be expressed as an extremisation condition for the imaginary part of $\mathcal{Y}(\boldsymbol{a}, \epsilon)$ rather than the holomorphic function $\mathcal{Y}(\boldsymbol{a}, \epsilon)$ itself:

$$\operatorname{Im} a_k \in \hbar\mathbb{Z}, \qquad \operatorname{Im} a_k^{\mathrm{D}} \in \hbar\mathbb{Z}. \tag{1}$$

The analytic Langlands correspondence proposed in [16] completes the picture by introducing natural scalar products strengthening the quantisation conditions for general Hitchin systems from single-valuedness to normalizability of a wave-function, and formulating conjectures relating the solutions to these conditions to real opers. The relation of the analytic Langlands correspondence to the $\mathcal{N} = 4$ supersymmetric Yang-Mills theory has been clarified in [17].[3]

Taken together, one arrives at a striking picture. The Hitchin moduli spaces $\mathcal{M}_{\mathrm{H}}$ can be represented in one complex structure as a complex torus fibration, exhibiting a classically integrable structure. There exist moduli spaces $^L\mathcal{M}_{\mathrm{H}}$ Langlands dual to $\mathcal{M}_{\mathrm{H}}$, with $^L\mathcal{M}_{\mathrm{H}}$ related to $\mathcal{M}_{\mathrm{H}}$ by mirror symmetry. Quantisation of the integrable structure of $\mathcal{M}_{\mathrm{H}}$ leads to quantum integrable systems. The spectra of these quantum integrable systems are encoded in distinguished sub-manifolds of $^L\mathcal{M}_{\mathrm{H}}$ which are Lagrangian with respect to a holomorphic symplectic form related to the one describing $^L\mathcal{M}_{\mathrm{H}}$ as torus fibration by a hyperkähler rotation. Mirror symmetry of hyperkähler manifolds thereby furnishes the geometric background for a precise geometric characterisation of the solutions of a large class of quantum integrable models.

In this paper we will study a natural deformation of the picture outlined above, in which the role of the Hitchin eigenvalue equations is taken by the KZB equations from conformal field theory. We will argue that this deformation admits an interpretation as a quantum theory obtained by quantising the Hitchin moduli space in a holomorphic symplectic structure that differs from the one considered in relation to the analytic Langlands correspondence. Solutions to the KZB-equations which are single-valued on $\operatorname{Bun}_G$ (but not necessarily on the space of complex structures!) will represent states in this quantum theory. The result is an analytic version of the Quantum Langlands correspondence [18–20].

The conformal field theory relevant in this context is known in the literature as the $H_3^+$-WZNW model.[4] The correlation functions of the $H_3^+$-WZNW model are solutions to the KZB-equations which are single-valued both on $\operatorname{Bun}_G$ and on the space of complex structures. A key ingredient in our proposal will be a family of topological defects modifying the correlation functions of the $H_3^+$-WZNW model which generalise the Verlinde line operators previously defined in other conformal field theories [21–23]. Correlation functions with Verlinde line insertions give solutions of the KZB-equations which are single-valued on $\operatorname{Bun}_G$, and the mapping class group is simply represented by its natural action on the support of the Verlinde line operators. We conjecture that these operators generate a basis for the space of states of this quantum theory which carries a natural action of two copies of the Skein algebra associated to $G$ and $C$.

As support for this conjecture we are going to demonstrate that the insertion of Verlinde line operators is related in the critical level limit to a geometric operation called grafting, constructing a new real oper from a given one. It is a known mathematical result that all real opers can be obtained by applying the grafting operation to the real oper associated to the hyperbolic metric uniformising a given complex structure. The real analytic Langlands conjectures include the completeness of the set of eigenstates of the Hitchin-Hamiltonians corresponding

---

[3]There is also a well-defined "real" generalization involving square-integrable functions on real slices of $\operatorname{Bun}_G$, which makes contact with and generalizes the earlier attempts to quantize real slices of the complex integrable system.

[4]For the case where the gauge group is $G = SL(2)$. More generally, a $G/G_c$ WZNW model.

to real opers. This completeness turns out to be a limiting case of the conjectured completeness of the set of single-valued KZB solutions created with the help of the Verlinde line operators.

The quantum theory defined in this way can be interpreted as a quantisation of complex Chern-Simons theory, as in [24], with the $H_3^+$ WZW model playing an analogous role as the rational WZW models in conventional Chern-Simons theory. A companion paper [25] discusses this interpretation and relations to other quantization strategies for complex CS theory in more depth.

Our results also have interesting implications for class $\mathcal{S}$ theories. Solutions to the KZ-equations can in this context be represented by partition functions in the presence of surface defects of co-dimension two [26, 27]. A new insight offered by the results discussed in this paper is that additional line defects can represent operators creating arbitrary eigenstates from a reference state, playing a role analogous to the Bethe creation operators known in the theory of integrable spin chains. Taken together, these results indicate that the quantum integrable structures of the class $\mathcal{S}$ theories are best revealed by using the *interplay* of all the supersymmetric defects.

## 1.1 Structure of the paper

We begin in the following Section 2 with a brief review of the quantisation of the Hitchin system, and how this quantisation appears in the context of quantum field theory.

Section 3 reviews background on the analytic Langlands correspondence, relating single-valued and normalisable eigenfunctions of the quantised Hitchin's Hamiltonians to real opers. The following Section 4 begins with a review of the classification of real opers using grafting. This classification is reformulated in terms of the generating function $\mathcal{Y}$.

Section 5 formulates conjectures following from the relations to quantum field theory.

In order to verify these conjectures we apply techniques from conformal field theory, described in Sections 6-8. Section 6 summarises the required background on the $H_3^+$-WZNW model. A first definition of the Verlinde line operators is proposed in this section. The next Section 7 reviews the relation between the correlation functions of the $H_3^+$-WZNW model and Liouville theory. It takes the form of an integral transformation mapping single-valued solutions to the BPZ-equations to single-valued solutions to the KZB equations. We will observe that this transformation maps Liouville correlation functions modified by insertions of Verlinde line operators to solutions to the KZB equations. This will be used in Section 8 to deduce crucial properties of correlation functions of these solutions, interpreted as wave-functions.

Section 9 analyses the critical level limit of correlation functions of the $H_3^+$-WZNW model modified by Verlinde line operators. We argue that this limit can be represented by eigenfunctions of the quantised Hitchin Hamiltonians corresponding to real opers obtained from the uniformising oper by grafting.

The concluding Section 10 offers an outlook towards generalizations associated to groups of higher ranks, formulated in the language of 2d CFT.

## 2 Quantisation of the Hitchin system

This section briefly reviews the quantum integrable system obtained by quantisation of the Hitchin system, and the associated spectral problem formulated in [16]. We also review the physical origin of the spectral problem and its gauge theory interpretation.

## 2.1 Hitchin's integrable system

Let $C$ be a Riemann surface. We shall consider surfaces $C = C_{g,n}$ which may have genus $g$ and $n$ punctures, often restricting to cases with $g = 0$ to give concrete examples.

The Hitchin moduli space $\mathcal{M}_H(C)$ is the moduli space of pairs $(\mathcal{E}, \varphi)$, where $\mathcal{E}$ is an holomorphic $G = SL(2)$-bundle on $C$ and $\varphi \in H^0(C, \mathrm{End}(\mathcal{E}) \otimes K)$. When the number $n$ of punctures is non-zero, we will assume that $\varphi$ has only regular singularities, represented by first order poles in local trivialisations.[5]

$\mathcal{M}_H(C)$ has a canonical Poisson structure. The restriction of this Poisson structure to the open dense subset $T^* \mathrm{Bun}_G$, with $\mathrm{Bun}_G$ being the moduli space of stable holomorphic $G$-bundles, coincides with the canonical cotangent bundle Poisson structure.

Let $\theta := \frac{1}{2} \mathrm{tr}(\varphi^2) \in H^0(C, K^2)$, and let $\{Q_r ; r = 1, \ldots, 3g - 3 + n\}$ be a basis for $H^0(C, K^2)$. The expansion

$$\theta(z) = \sum_{r=1}^{3g-3+n} H_r Q_r(z),$$

defines Hamiltonians $H_r$ satisfying $\{H_r, H_s\} = 0$. The fibers of $\pi : \mathcal{M}_H(C) \to H^0(C, K^2)$, $\pi(E, \varphi) = \theta$ are complex tori for generic $\theta$. This defines the integrable structure on $\mathcal{M}_H(C)$.

The Hitchin system has been quantised on an algebraic level by Beilinson and Drinfeld:

*There exists a square-root $K_{\mathrm{Bun}}^{1/2}$ of the canonical bundle on $\mathrm{Bun}_G$, and differential operators $\mathsf{H}_r$, $r = 1, \ldots, 3g - 3 + n$ on $K_{\mathrm{Bun}}^{1/2}$ satisfying*

$$[\,\mathsf{H}_r, \mathsf{H}_s\,] = 0, \qquad r, s = 1, \ldots, 3g - 3 + n,$$

*and having symbols which coincide with the functions $H_r$ on $\mathcal{M}_H(C)$.*

The differential operators $\mathsf{H}_r$ therefore represent a quantisation of the Hamiltonians $H_r$.

## 2.2 Natural quantisation conditions

For given pair $(\boldsymbol{E}, \bar{\boldsymbol{E}})$, where $\boldsymbol{E}, \bar{\boldsymbol{E}} \in \mathbb{C}^d$, $d := 3g - 3 + n$, we may consider the pair of complex conjugate eigenvalue equations

$$\begin{aligned}\mathsf{H}_r \Psi &= E_r \Psi, \\ \bar{\mathsf{H}}_r \Psi &= \bar{E}_r \Psi,\end{aligned} \qquad r = 1, \ldots, d, \qquad \begin{aligned}\boldsymbol{E} &= (E_1, \ldots, E_d), \\ \bar{\boldsymbol{E}} &= (\bar{E}_1, \ldots, \bar{E}_d).\end{aligned} \tag{2}$$

The Hitchin Hamiltonian have singularities at wobbly bundles, which are bundles admitting nilpotent Higgs fields. Let $\mathrm{Bun}_G^{\mathrm{vs}}$: moduli space of stable bundles which are not wobbly. We may look for smooth solutions on $\mathrm{Bun}_G^{\mathrm{vs}}$ which are *single-valued* in the following sense. Choosing local coordinates $\boldsymbol{x} = (x_1, \ldots, x_d)$ on $\mathrm{Bun}_G^{\mathrm{vs}}$ one may represent the solutions $\Psi$ of (2) locally in the form

$$\Psi(\boldsymbol{x}, \bar{\boldsymbol{x}}) = \sum_{k,l} C_{kl} \psi_k(\boldsymbol{x}) \bar{\psi}_l(\bar{\boldsymbol{x}}), \qquad \begin{aligned}\mathsf{H}_r \Psi &= E_r \Psi, \\ \bar{\mathsf{H}}_r \Psi &= \bar{E}_r \Psi,\end{aligned} \qquad r = 1, \ldots, d.$$

Single-valuedness requires that the monodromies of the local sections $\psi_r(\boldsymbol{x})$ and $\bar{\psi}_s(\bar{\boldsymbol{x}})$ must cancel each other.

The work of Etingof, Frenkel and Kazhdan [16, 29, 30] introduces a Hilbert space framework for the system of eigenvalue equations (2). One may first introduce a smooth algebraic moduli space $\mathrm{Bun}_G^{\mathrm{rs}}(C)$ by considering the stack of bundles $\mathrm{Bun}_G^{\circ}(C)$ with automorphisms in the center of $G$, and forgetting automorphisms. A Hilbert space associated to $\mathrm{Bun}_G^{\mathrm{rs}}(C)$ can be introduced as follows. Let

---

[5]The regular singularities also modify the definition of $\mathcal{E}$ by reducing the structure group at the point to a parabolic subgroup. For brevity, this modification is implied but not explicitly mentioned in the discussion below.

- $\Omega_{\mathrm{Bun}}^{1/2} := |K_{\mathrm{Bun}}|$ be the line bundle of half-densities, where $|\mathcal{L}|^2 = \mathcal{L} \otimes \mathcal{L}$,

- $\mathcal{S}$ be the space of smooth compactly supported sections of $\Omega_{\mathrm{Bun}}^{1/2}$, and

- let us define a Hermitian form $\langle . , . \rangle$ on $\mathcal{S}$ by

$$\langle v, w \rangle := \int_{\mathrm{Bun}_G^{\mathrm{rs}}} v \, \bar{w} , \qquad v, w \in \mathcal{S} . \tag{3}$$

The Hilbert space $\mathcal{H}_{\mathrm{Bun}}$ is then defined as the completion of $\mathcal{S}$ with respect to $\langle . , . \rangle$.

The collection of results presented in [16, 30] supports the following conjecture:

(i) The eigenspaces $\mathcal{H}_{E,\bar{E}}$ generated by single-valued solutions to the pair of eigenvalue equations with eigenvalues $(E, \bar{E})$ contained in $\mathcal{H}$ are at most one-dimensional, and non-vanishing only if $\bar{E}$ is complex conjugate to $E$.

(ii) The Hilbert spaces $\mathcal{H}$ admit an orthogonal direct sum decomposition into the eigenspaces $\mathcal{H}_{E,\bar{E}}$ (completeness).

The question is formulated in [16] if the conditions of single-valuedness and $\bar{E}$ being the complex conjugate of $E$ imply square-integrability.

One may furthermore introduce integral operators called Hecke operators of the form [29]

$$(\mathbf{H}_{P,\lambda} f)(\mathcal{E}) := \int_{Z_{P,\lambda}(\mathcal{E})} q_1^*(f) . \tag{4}$$

- $Z_{P,\lambda}(\mathcal{E})$ – space of all Hecke modifications of the bundle $\mathcal{E}$ at the point $P$, isomorphic to the closure $\overline{\mathrm{Gr}}_\lambda$ of the orbit $\mathrm{Gr}_\lambda = G[[t]] \cdot \lambda(t) / G[[t]]$.

- $q_1^*(f)$ is the pull-back of $f$ under the correspondence between holomorphic bundles defined by the Hecke modifications parameterised by $\overline{\mathrm{Gr}}_\lambda$.

In [16, 29, 30] is conjectured and in some cases proven that

- the Hecke operators extend to a family of commuting compact self-adjoint operators on $\mathcal{H}$,

- $\mathcal{H}$ decomposes into eigenspaces of the Hecke operators,

- the eigenspaces coincide with the eigenspaces $\mathcal{H}_{E,\bar{E}}$ of the Hitchin Hamiltonians.

From the point of view of quantum integrable models one may regard the differential operators $\mathsf{H}_r$, $\bar{\mathsf{H}}_r$ as local conserved quantities, while the Hecke operators represent certain non-local conserved quantities analogous to Baxter Q-operators.

## 2.3 Quantum-field theoretical realisations of quantum Hitchin systems

In this sub-section we will elaborate on the quantum field-theoretical interpretation of the analytic Langlands program. The comparison has three levels of complexity, with increasing predictive power and decreasing mathematical rigour. Our discussion here is not specific to $G = SL(2)$ but is expected to apply to any reductive $G$, with Lie algebra $\mathfrak{g}$.

1. The most natural mathematical definition of the quantum Hitchin Hamiltonians involves the center of the $\mathfrak{g}_{\kappa_c}$ Kac-Moody vertex algebra at critical level $\kappa_c$. Precisely at critical level, the Sugawara vertex operator $\hat{\theta} \equiv J \cdot J$ becomes central, together with a larger

collection of higher polynomial elements. The action of these central elements on conformal blocks for the Kac-Moody chiral algebra coupled to $G$ bundles on $C$ reproduces the commuting quantum Hitchin Hamiltonians. The holomorphic half of the integrand for Hecke operators can also be reproduced with the help of spectral flow modules for the Kac-Moody algebra (see [12, 17] for a physical discussion). In order to formulate the quantization conditions and Hecke operators in a 2d language, we clearly need to combine chiral- and anti-chiral Kac-Moody algebras. Mathematically this could be done, say, using the language of factorization algebras. The main hurdle is to include elements of functional analysis in the definition of conformal blocks/factorization homology, allowing one to define spaces of conformal blocks related to wave-functions in $\mathcal{H}$. This 2d setting is far from being a fully-fledged 2d QFT: it lacks a stress tensor and it does not assign, say, a partition function to a 2d manifold. This motivates the next steps.

2. The action of operators on the Hilbert spaces of states of physical models motivates the study of spectral problems. Accordingly, we will now review a construction involving a three-dimensional quantum field theory ("complex $G$ HT-BF theory") whose space of states on $C$ is conjecturally $\mathcal{H}$, with local operators which are identified with the generating functions of the Hitchin Hamiltonians including $\hat{\theta}(z)$, and their complex conjugates, or with Hecke operators $\mathbf{H}_\lambda[z, \bar{z}]$ supported at points $z \in C$. In a sense which is still to be made mathematically precise, the three-dimensional theory is the "center" of the above two-dimensional setup: it is equipped with a family of boundary conditions labelled by $G$ bundles on $C$ which support a chiral and an anti-chiral copy of the $\mathfrak{g}_{\kappa_c}$ Kac-Moody chiral algebra. Bulk local operators brought to the boundary reproduce the 2d construction of Hitchin Hamiltonians and Hecke operators.

   The above three-dimensional theory combines an "holomorphic-topological factorization algebra" with its complex conjugate in a non-trivial fashion, analogous to the way a 2d CFT combines holomorphic and anti-holomorphic chiral algebras. The HT factorization algebra may already be a mathematically rigorous object, see e.g. [31], but the combination is not. Physically, the complex HT-BF theory is certainly well-defined at a perturbative level, but the action is somewhat unconventional and it is unclear if a non-perturbative definition should exist. This point is addressed by increasing again the dimensionality of the system.

3. There is a further four-dimensional lift of the construction which embeds the 3d theory into the A-type Kapustin-Witten twist of four-dimensional $\mathcal{N} = 4$ SYM with gauge group $G_c$, the compact subgroup of $G$. This setup is physically well-defined: the physical 4d theory is expected to exist non-perturbatively and the the twist is not expected to change this fact. More practically, it relates many phenomena in the ALC correspondence to S-duality.[6]

### 2.3.1 The 2d construction

The 2d CFT interpretation of the spectral problem follows from the observation that holomorphic Kac-Moody conformal blocks can be defined for any Riemann surface equipped with a $G$ bundle. The conformal blocks are closely related to the D-modules of holomorphic differential operators on $\text{Bun}_G$. In order to make contact with QFT constructions, we discuss this relation in gauge-theoretic terms: a smooth wave-functions on $\text{Bun}_G$ can be identified, essentially

---

[6]An important open question in the four-dimensional setup is to demonstrate that the space of states of the system is indeed an Hilbert space with a positive-definite inner product. This is not a property which could be derived directly from the 4d KW setup, but would follow if one could demonstrate that the embedding of the twisted theory into the physical $\mathcal{N} = 4$ SYM theory is compatible with reflection positivity.

by definition, with a gauge-covariant functional on the space of $(0,1)$ $G$-connections $\mathcal{A}_{\bar{z}}$. We can thus take functional derivatives $\frac{\delta}{\delta\mathcal{A}_{\bar{z}}}(z)$ and $\frac{\delta}{\delta\mathcal{A}_{\bar{z}}}(\bar{z})$ of $\Psi$. A simple calculation shows that the derivatives depend (anti)holomorphically on the point in $C$ and that repeated derivatives satisfy the same Ward identities as insertions on $C$ of critical Kac-Moody currents $J(z)$ and $\bar{J}(\bar{z})$.

The output of a general collection of functional derivatives will not be a wavefunction, as the current insertions modify the behaviour under gauge transformations. The insertion of central elements in the Kac-Moody chiral algebra such as the Sugawara element $\hat{\theta}(z) \equiv J(z) \cdot J(z)$, though, does not change the action of gauge transformations and thereby produces a new wave-function: it maps to a globally defined twisted differential operator on $\mathrm{Bun}_G$. Centrality also guarantees that insertions at different points do not affect each other, so the differential operators associated to different points in $C$ commute. This leads to the commutativity of quantum Hitchin Hamiltonians.

A local Hecke modification by an element of $\overline{\mathrm{Gr}}_\lambda$ can be applied verbatim to the gauge-covariant functional associated to $\Psi$. The result, again, is not a wavefunction on $\mathrm{Bun}_G$, as a specific choice of Hecke modification does not commute with gauge transformations. If we act with functional derivatives, current Ward identities will be modified as in the presence of the corresponding spectral flow image of the vacuum module inserted at the point.[7] The averaging procedure which defines the Hecke operators produces averaged spectral flow operators, which are central and map wavefunctions to wavefunctions. This definition can be employed to derive the BTZ-like differential equations satisfied by Hecke operators, which are an important part of the story. For example, if $G = SL(2)$ one has

$$(\partial_z^2 + \hat{\theta}(z))\mathbf{H}_{\frac{1}{2}}[z,\bar{z}] = 0, \qquad (\bar{\partial}_{\bar{z}}^2 + \hat{\bar{\theta}}(\bar{z}))\mathbf{H}_{\frac{1}{2}}[z,\bar{z}] = 0, \tag{5}$$

to be understood as operator equations valid within expectation values.

### 2.3.2 The 3d construction

One possible way to demonstrate the existence of commuting quantum Hitchin Hamiltonians uses quantum field theory models whose phase space is the Hitchin moduli space $\mathcal{M}_H(C,G)$, and quantizes them it in a way which preserves the commutativity of the Hamiltonians. This strategy was implemented in [17]: $\mathcal{M}_H(C,G)$ can be identified as the phase space of a QFT called Complex Holomorphic-Topological BF theory, which is a three-dimensional gauge theory defined on $C \times \mathbb{R}$.

The elementary fields in the 3d gauge theory are 3d versions of the Higgs field $\varphi$ and a complex Holomorphic-Topological 3d connection combining the $(0,1)$ component $A_{\bar{z}}$ along $C$ and a component $A_t$ along $\mathbb{R}$. The action is simply

$$\frac{i}{4\pi}\int_{C\times\mathbb{R}} \mathrm{tr}\,\varphi F_{t\bar{z}} - \mathrm{c.c.} \tag{6}$$

Classical solutions $(\varphi, A_{\bar{z}})$ of the equations of motion modulo gauge transformations coincide with time-independent Higgs bundles on $C$. Notice that the action is the sum of two terms, each involving either $\varphi$ and the complex connection $A$ or their Hermitean conjugates $\bar{\varphi}$ and $\bar{A}$.

The complex HT-BF theory is topological along $\mathbb{R}$. Local operators in the 3d theory placed at different points on $C$ can be freely shifted along $\mathbb{R}$ and in particular act as a collection of commuting operators on the Hilbert space of the theory, which we would identify with $\mathcal{H}$.

Gauge-invariant polynomials of the Higgs field, such as $\theta(z)$, survive as *holomorphic* local operators $\hat{\theta}(z)$ in the quantum theory. They transform as appropriate (affine) holomorphic

---

[7]The insertion of a spectral flow image of the vacuum module at a point on $C$ is indeed equivalent to a modification of the bundle at that point [12, 17].

bundles on $C$ and can be expanded into a basis of sections to give the quantum Hitchin Hamiltonians. Conjugate Hamiltonians arise from polynomials in $\bar{\varphi}$. It is also possible to define disorder local operators, i.e. monopole operators, which reproduce the action of Hecke modification and thus demonstrate that Hecke modifications exist, commute at different points and also commute with the quantum Hitchin Hamiltonians.

The connection with the Kac-Moody perspective involves a "Dirichlet" boundary condition $(A_{\bar{z}})_\partial = \mathcal{A}_{\bar{z}}^{2d}$, which fixes the bundle at the boundary and forces gauge transformations to be the identity at the boundary. The boundary values $\varphi_\partial$ of the bulk fields map to the 2d Kac-Moody currents. The Dirichlet boundary conditions define distributional functionals on $\mathcal{H}$, identified with the evaluation of a smooth wavefunction or its derivatives at a specific bundle. Formally,

$$\Psi(\mathcal{A}_{\bar{z}}) = \langle \mathrm{Dir}[\mathcal{A}_{\bar{z}}] | \Psi \rangle \,. \tag{7}$$

This is closely analogous to, say, the relation between the Hilbert space of a compact Chern-Simons theory and the conformal blocks of a chiral WZW model. The main difference is that $(A_{\bar{z}})_\partial = \mathcal{A}_{\bar{z}}^{2d}$ is a *real* polarization of the classical phase space of the complex HT-BF theory, so that normalizable states in $\mathcal{H}$ map to $L^2$-normalizable wavefunctions on Bun.

### 2.3.3 The 4d construction

The complex HT-BF theory can be further lifted to the compactification of A-twisted four-dimensional $\mathcal{N} = 4$ Super Yang Mills on a segment $[-1, 1]$. This is a special case of a general construction [32] we refer to as "analytic continuation of a path integral". Given a $D$-dimensional action $S$ which is only perturbatively well-defined, one builds a topological field theory in $D+1$ dimension equipped with a canonical boundary condition $B_{cc}$, such that boundary local operators satisfy the Schwinger-Dyson equations derived from $S$. If one can also find a second boundary condition $B_{top}$ which is topological, a "slab" geometry of the form $[0, 1] \times \cdots$

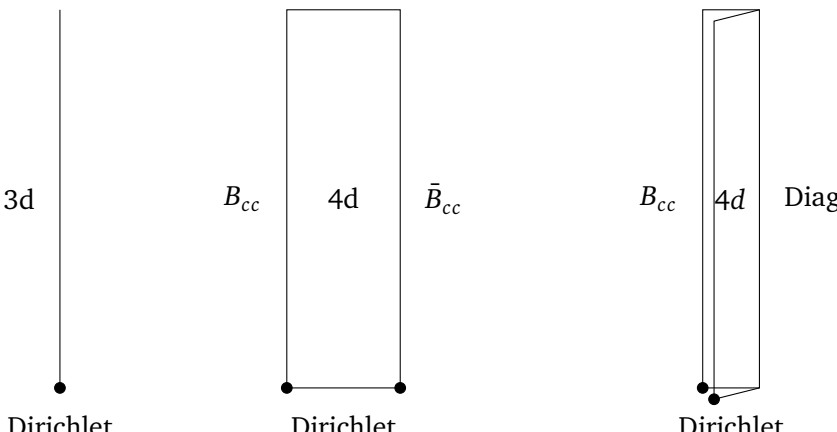

Figure 1: Left: the 3d complex HT-BF theory compactified on $C$ gives the desired Hilbert space $\mathcal{H}$. Dirichlet boundary conditions give distributional boundary states labelled by a bundle and support holomorphic and anti-holomorphic critical Kac-Moody currents. Right: The analytic continuation procedure gives an embedding in a $G \times G$ A-twisted Kapustin-Witten theory compactified on $[0, 1] \times C$, with $B_{cc} \times B_{cc}$ boundary conditions at one end of the segment and a diagonal reflection boundary condition at the other end. Middle: this can be unfolded to the 4d theory compactified on $[-1, 1] \times C$. In either case, the Kac-Moody currents live at the corners between the $B_{cc}$ and Dirichlet boundary conditions.

with $B_{cc}$ at 1 and $B_{top}$ at 0 engineers a well-defined system which behaves as a $D$-dimensional "contour path integral" with action $S$. The integration contour is encoded in $B_{top}$.

Here $S$ is the difference of an action for $\varphi$ and $A$ and an action for their complex conjugates $\bar{\varphi}$ and $\bar{A}$. Analytic continuation promotes the 3d fields and their complex conjugates to independent 4d fields. The topological boundary condition $B_{top}$ can be produced by a reflection trick, essentially enforcing that the path integral is effectively done along $\bar{\varphi} = \varphi^\dagger$ and $\bar{A} = A^\dagger$. The result is the 4d KW A-twist compactified on a segment $[-1, 1]$ with $B_{cc}$ boundary conditions we will refer to as "chiral" and "anti-chiral" Neumann boundary conditions, which effectively support the chiral and anti-chiral halves of the 3d HT-BF theory. The Hitchin Hamiltonian generators $\hat{\theta}(z)$, etc. appear as holomorphic boundary local operators. Hecke operators arise from 't Hooft line defects stretched between the two boundaries.

S-duality maps the setting to the B-twisted four-dimensional $\mathcal{N} = 4$ Super Yang Mills with Langlands dual gauge group $G^\vee$. Strongly interacting, non-perturbative features of the A-twist map to semi-classical calculations in the B-twist. The chiral and anti-chiral Neumann boundary conditions map to "oper" or "anti-oper" boundary conditions imposing certain singularities in the fields at the boundary. The bulk 't Hooft lines map to bulk Wilson lines. The relations satisfied by Hecke operators, such as the differential equation (5), become classical properties of the Wilson lines stretched between the two boundaries.

The B-twist description of $\mathcal{H}$ involves the solution of certain classical equations of motion: find flat $G^\vee$ connections which are both "opers" and "anti-opers". We will review the meaning of these terms in the next section, but in practice the equations of motion describe eigenvalues $t(z)$, $\bar{t}(z)$ and $\chi(z, \bar{z})$ for the quantum operators $\hat{\theta}(z)$, $\hat{\bar{\theta}}(\bar{z})$ and $\mathbf{H}_{\frac{1}{2}}[z, \bar{z}]$, compatible with (5). Each solution thus gives a state in the B-twist Hilbert space which is automatically an eigenstate of Hitchin and Hecke operators and viceversa. Assuming S-duality, this leads to a precise but non-constructive prediction for the spectrum of Hitchin's Hamiltonians.[8]

Another payoff of the 4d construction is that it extends naturally to a "real" version of the spectral problem, involving bundles on Riemann surfaces with boundaries or cross-caps.

# 3 Analytic Langlands correspondence

This section introduces the geometric Langlands correspondence along with a strengthened form of this correspondence called analytic (geometric) Langlands correspondence [16]. The analytic Langlands correspondence is closely related to the quantisation of the Hitchin system discussed in the previous section.

## 3.1 Geometric Langlands correspondence

The geometric Langlands correspondence can be schematically represented as a correspondence between two types of objects:

$$\boxed{{}^L\mathfrak{g} \text{ local systems } (\mathcal{E}, \nabla) \text{ on } C} \quad \longleftrightarrow \quad \boxed{\mathcal{D}\text{-modules on } \mathrm{Bun}_G(C),} \qquad (8)$$

with $G$ being a complex group[9] with Lie algebra $\mathfrak{g}$, ${}^L\mathfrak{g}$ being the Langlands dual Lie algebra of $\mathfrak{g}$, and local systems $(\mathcal{E}, \nabla)$ being pairs consisting of holomorphic $G$-bundles $\mathcal{E}$, and holomorphic connections $\nabla$ on $\mathcal{E}$.

An important special case of the geometric Langlands correspondence considers local systems $(\mathcal{E}, \nabla_\eta)$ called opers. In the case $\mathfrak{g} = \mathfrak{sl}_2$ one considers pairs $\eta = (\mathcal{E}, \nabla_\eta)$, where

---

[8]Up to an important subtlety: unitarity is a non-obvious property of the B-twisted setup.
[9]Split, reductive.

- $\mathcal{E} = \mathcal{E}_{\mathrm{op}}$ is the unique up to isomorphism extension $0 \to K^{\frac{1}{2}} \to \mathcal{E}_{\mathrm{op}} \to K^{-\frac{1}{2}} \to 0$, and

- $\nabla_\eta$ locally gauge equivalent to the form

$$\nabla_\eta = dz\left(\partial_z + \begin{pmatrix} 0 & -t_\eta \\ 1 & 0 \end{pmatrix}\right), \qquad \text{where } t_\eta \text{ transforms as a projective connection,}$$

$$t_\eta(z) = \left(\varphi'(z)\right)^2 \tilde{t}_\eta\left(\varphi(z)\right) + \frac{1}{2}\{\varphi, z\}, \qquad \{\varphi, z\} := \frac{\varphi'''}{\varphi'} - \frac{3}{2}\left(\frac{\varphi''}{\varphi'}\right)^2.$$

In other words, an oper connection is defined by a Schrödinger operator $\partial_z^2 + t_\eta(z)$ defined on sections of $K^{-\frac{1}{2}}$.

Note that $q_\eta = t_\eta - t_0 \in H^0(C, K^2)$ for $t_0$ fixed, giving coordinates on $\mathrm{Op}(C)$,

$$q_\eta = t_\eta - t_0 = \sum_{r=1}^{d} E_r(\eta)\theta_r, \tag{9}$$

where $\{\theta_r; r = 1, \ldots, d\}$ is a basis for $H^0(C, K^2)$.

The geometric Langlands correspondence can be represented in this case more explicitly as the correspondence between the opers $\eta$ and the $\mathcal{D}$-modules represented by the equations

$$\mathsf{H}_r \Psi = E_r(\eta)\Psi,$$

or, in terms of generating functions,

$$\hat{\theta}(z)\Psi = t_\eta(z)\Psi.$$

This means in particular that there are natural ways to fix the ordering ambiguities in the quantisation of Hitchin's Hamiltonians exhibiting a direct relation to the ambiguity in the definition of the coordinate functions $E_r(\eta)$ using (9), represented by the choice of reference projective connection $t_0$ in (9).

## 3.2 Analytic Langlands correspondence

Variants of the geometric Langlands correspondence can be formulated by imposing additional conditions. A natural condition is the condition of single-valuedness [15]. The additional condition of square-integrability has been considered in [16]. This must imply restrictions on the objects on the left side of the correspondence (8). It has been proposed in [15, 16] that the corresponding objects on the left of (8) are opers with real holonomy, schematically

$$\boxed{\text{Real opers}} \quad \longleftrightarrow \quad \boxed{\begin{Bmatrix} \text{single-valued and} \\ L^2\text{-normalisable} \end{Bmatrix} \text{Hitchin eigen-}\mathcal{D}\text{-modules.}} \tag{10}$$

The eigenfunctions of the Hitchin eigen-$\mathcal{D}$-modules on the right play a role analogous to the automorphic forms in the usual Langlands correspondence.

This statement has several non-trivial aspects:

- In order to define an eigenvalue problem, the holomorphic Hitchin Hamiltonians should be normal operators, adjoint to the anti-holomorphic Hamiltonians. Proving existence of a normal extension of the Hitchin Hamiltonians is a non-trivial functional-analytic problem.

- To each real oper there should correspond a unique solution to the spectral problem. The eigenspaces of the Hitchin Hamiltonians are claimed to be one-dimensional, and to be eigenspaces of the Hecke operators at the same time.

- The Hitchin eigenvectors associated to real opers are complete in $\mathcal{H}_{\text{Bun}}$, they generate a dense subset of this Hilbert space.

We will denote the eigenvalue of the Hecke operators of minimal charge $\mathbf{H}_{\frac{1}{2}}[u,\bar{u}]$ as $\chi_\eta(u,\bar{u})$. It satisfies

$$(\partial_u^2 + t_\eta(u))\chi_\eta(u,\bar{u}) = 0, \qquad (\bar{\partial}_{\bar{u}}^2 + \bar{t}_\eta(\bar{u}))\chi_\eta(u,\bar{u}) = 0. \tag{11}$$

It can be shown that single-valuedness of $\chi_\eta$ requires that the oper $\partial_u^2 + t_\eta(u)$ is real [28].

We will next describe the arguments given in [15], which allow one to reconstruct the Hitchin eigenfunctions from the data $\chi_\eta(u,\bar{u})$ encoding the real oper.

## 3.3 Separation of variables

As a preparation we will now briefly summarise results from past and upcoming work on the Separation of Variables for the Hitchin system forming relevant background for our work.

The Separation of Variables (SOV) method for the Hitchin system has been developed in the series of works [33–39]. It can be described geometrically using a representation of the holomorphic bundles $E$ as extensions:[10]

$$0 \to L \to E \to \Lambda L^{-1} \to 0,$$

allowing us to represent the Higgs fields locally as $\varphi = \left(\begin{smallmatrix} a & b \\ c & d \end{smallmatrix}\right)$ with $c \in H(C, KL^{-2}\Lambda)$. In the quantised Hitchin system one may represent $c$ as a first order differential operator. It can be diagonalised by a Fourier-transform on $V = H(C, KL^{-2}\Lambda)$. The key insight forming the basis of the SOV method is that the change of variables from linear coordinates on the vector space $V$ to the coordinates provided by the zeros of sections of $KL^{-2}\Lambda$ simplifies the eigenvalue equations considerably.

A similar simplification has first been observed in the case of the Gaudin model in the pioneering work [33]. The Gaudin models can be regarded as variants[11] of the Hitchin systems on surfaces of genus zero. It has been observed in [34] that Drinfeld's first construction of the geometric Langlands correspondence can be understood as an analog SOV method for quantised Hitchin systems, leading to the geometric picture outlined above. The relation between Drinfeld's first construction and the classical version of the SOV method has been made explicit in [39].

### 3.3.1 Representation as an integral transformation

The SOV method can be used to construct an explicit integral transformation which intertwines the Hitchin eigenvalue equations with a system of ordinary differential equations, each of which coincides with the oper differential equations (11) involving only a single variable. This has first been observed [38] for Hitchin systems associated to surfaces $C = C_{g,n}$ with $g = 0$, and generalisations to surfaces with genus $g > 0$ will be treated in the upcoming work [40] based on [39]. In all these cases one finds an integral transformation of the following form:

$$\Psi(\boldsymbol{x},\bar{\boldsymbol{x}}) = \int_{S^d C} d\mu(\boldsymbol{u},\bar{\boldsymbol{u}}) \, K(\boldsymbol{x},\bar{\boldsymbol{x}} \,|\, \mathbf{u},\bar{\mathbf{u}}) \Phi(\mathbf{u},\bar{\mathbf{u}}), \tag{12}$$

where $S^d C$ is the symmetric product of $d$ copies of C, and $\mathbf{u} = (u_1,\dots,u_d)$, $\bar{\mathbf{u}} = (\bar{u}_1,\dots,\bar{u}_d)$. Note that $d = 3g - 3 + n$ coincides with the dimension of the space of holomorphic $SL(2)$-bundles on $C$ having parabolic structures at the $n$ punctures, while $d\mu(\boldsymbol{u},\bar{\boldsymbol{u}})$ is a measure

---

[10]This basically amounts to describing the bundles in terms a family of local trivialisation on an open cover related by upper-triangular transition functions. We include in the definition the possibility of a fixed non-trivial determinant line bundle $\Lambda$, which is instrumental to define some useful variations of the notion of $SL(2)$ bundles.

[11]The models differ in the treatment of the global symmetries.

briefly discussed below. The integral transformation (12) is essentially characterised by the property that (2) is equivalent to

$$(\partial_{u_r}^2 + t(u_r))\chi(u_r, \bar{u}_r) = 0, \qquad (\bar{\partial}_{\bar{u}_r}^2 + \bar{t}(\bar{u}_r))\chi(u_r, \bar{u}_r) = 0, \qquad r = 1, \ldots, d.$$  (13)

It follows that the transform (12) yields a Hitchin eigenfunction if the function $\Phi$ has the form,

$$\Phi(\mathbf{u}, \bar{\mathbf{u}}) = \prod_{r=1}^{d} \chi(u_r, \bar{u}_r), \qquad \text{where } \chi \text{ satisfies} \qquad \begin{array}{l} (\partial_u^2 + t(u))\chi(u, \bar{u}) = 0, \\ (\bar{\partial}_{\bar{u}}^2 + \bar{t}(\bar{u}))\chi(u, \bar{u}) = 0. \end{array}$$  (14)

The explicit formulae for the kernel $K(\mathbf{x}, \bar{\mathbf{x}}|\mathbf{u}, \bar{\mathbf{u}})$ are known for genus 0 [38] and for $g > 0$ [40]. It is a single-valued function of the variables $\mathbf{x}, \bar{\mathbf{x}}$ and $\mathbf{u}, \bar{\mathbf{u}}$. It follows that the transform (12), (14) maps single-valued solutions $\chi(u, \bar{u})$ of the holomorphic and anti-holomorphic oper equations (13) to single-valued solutions of the Hitchin eigenvalue equations. As real opers correspond to single-valued solutions of (13), one may regard the transform (12) as an explicit realisation of an important part of the analytic Langlands correspondence (10).

### 3.3.2 Unitarity

Up to now we have only considered the conditions coming from single-valuedness of the solutions to the Hitchin eigenvalue equations. Recalling that the scalar product on $\text{Bun}_G$ defined by (3) provides a functional-analytic framework for the analytic Langlands correspondence, it is natural to ask if the SOV-transformation (12) can represent a unitary map between suitable spaces of square-integrable functions.[12]

One may notice that the SOV method leads to a natural choice of the measure of integration in (12). It is induced from the Lebesgue-measure on the vector space $V = H(C, KL^{-2}\Lambda)$ by the change of variables from linear coordinates on $V \simeq \mathbb{C}^{d+1}$ to the coordinates provided by the zeros of sections of $KL^{-2}\Lambda$. This measure will be referred to as the Sklyanin measure, following the terminology customary in the literature on completely integrable systems.

The Sklyanin measure leads to the definition of a natural scalar product structure on the space of functions on the symmetric product $S^d C$,

$$\left\langle \Phi_2, \Phi_1 \right\rangle_{\text{SOV}} = \int_{S^d C} d\mu(\mathbf{u}, \bar{\mathbf{u}}) \, \bar{\Phi}_2(\mathbf{u}, \bar{\mathbf{u}}) \Phi_1(\mathbf{u}, \bar{\mathbf{u}}).$$  (15)

The Hilbert space defined by this scalar product will be denoted as $\mathcal{H}_{\text{SOV}}$. It will be established in upcoming work that the SOV-transformation (12) defines a unitary map $\text{SOV} : \mathcal{H}_{\text{SOV}} \to \mathcal{H}_{\text{Bun}}$.

We may propose an indirect approach to determine the measure of integration defining the scalar product in $\mathcal{H}_{\text{SOV}}$. It will be useful to bear in mind that a quadratic differential $q$ on $C$ is uniquely determined by specifying its values at $d = 3g - 3 + n$ distinct points $u_1, \ldots, u_d$ on $C$. This is expressed by the interpolation formula

$$q(z) = \sum_{i=1}^{d} \frac{\Theta(u_1, \ldots, u_{i-1}, z, u_{i+1}, \ldots, u_d)}{\Theta(u_1, \ldots, u_{i-1}, u_i, u_{i+1}, \ldots, u_d)} q(u_i),$$  (16)

where $\Theta$ is the determinant of the matrix having matrix elements $\det(\theta_r(u_i))$, $r, i = 1, \ldots, d$, with $\{\theta_r; r = 1, \ldots, d\}$ being a basis for $H^0(C, K^2)$. The system of equations

$$\left(\partial_{u_i}^2 + t_0(u_i) - q(u_i)\right)\Phi(u, \bar{u}) = 0, \qquad q(u) := \sum_{r=1}^{d} E_r \theta_r(u_r), \qquad i = 1, \ldots, d,$$

---

[12]This question has been brought up by D. Kazhdan in a discussion.

is equivalent to the validity of the eigenvalue equations

$$\mathsf{T}(z)\Phi(u,\bar{u}) = q(z)\Phi(u,\bar{u}), \tag{17}$$

for all $z$, where $\mathsf{T}(z)$ is the family of differential operators

$$\mathsf{T}(z) := \sum_{i=1}^{d} \frac{\Theta(u_1,\ldots,u_{i-1},z,u_{i+1},\ldots,u_d)}{\Theta(u_1,\ldots,u_{i-1},u_i,u_{i+1},\ldots,u_d)} \big(\partial_{u_i}^2 + t_0(u_i)\big). \tag{18}$$

We may next observe that $\mathsf{T}(z)$ is the representation of the generating function $\hat{\theta}(z)$ of the quantised Hitchin Hamiltonians in the SOV representation. Unitarity of the SOV transformation would imply that $\mathsf{T}(z)$ is normal for all $z$, and this would furthermore imply orthogonality of eigenfunctions to different eigenvalues. Setting $D(\boldsymbol{u}) := \Theta(u_1,\ldots,u_d)$ one may observe that $\mathsf{T}(z)$ is formally normal with respect to the inner product

$$\big\langle \Phi_2,\Phi_1 \big\rangle_{\det} = \int_{S^d C} \prod_i d^2 u_i \, \big|D(\boldsymbol{u})\big|^2 \, \bar{\Phi}_2(\mathbf{u},\bar{\mathbf{u}})\Phi_1(\mathbf{u},\bar{\mathbf{u}}), \tag{19}$$

assuming that the integration by parts is possible without boundary terms. This inner product is well-defined: The functions $\Phi_i$, $i = 1,2$, represent sections of $|K|^{-1}$, and $|D(\boldsymbol{u})|^2$ a section of $|K|^4$ with respect to each integration variable $u_i$, so the integrand is a density. If the measure $|D(\boldsymbol{u})|^2$ is uniquely determined by the property that $\mathsf{T}(z)$ is normal for all $z$, we could reach the conclusion that (19) must coincide with the Sklyanin measure.

## 4 Real opers

This section starts with a review of geometric background on the classification of real opers that will be used in the following. We will then provide important preparations for the following sections. New results that will be important later include the parameterisation of real opers with the help of Fenchel-Nielsen type coordinates, and their characterisation in terms of the generating function of the subspace of opers within the character variety. Preliminary versions of some of these results had been proposed in [15, 28].

### 4.1 Opers, projective structures, and uniformisation

Opers are in one-to-one correspondence to projective structures on $C$. One may use the ratios

$$A(z) = \frac{\chi_1(z)}{\chi_2(z)}, \tag{20}$$

where $\chi_i(z)$ are the two linearly independent solutions of $(\partial_z^2 + t(z))\chi_i(z) = 0$, to define new local coordinates $w = A(z)$ on $C$. The local coordinates $w$ defined in this way have the following features:

- The oper is represented with respect to the coordinate $w$ by $\tilde{t}(w) = 0$, and
- changes of coordinates $w'(w)$ are represented by Möbius transformations.

An atlas $\{U_\iota; \iota \in \mathcal{I}\}$ for $C$ with transition functions represented by Möbius transformations defines a projective structure. A projective structure naturally defines a representation of the fundamental group of $C$ by Möbius transformations, referred to as the holonomy of the projective structure. If a projective structure is obtained from an oper $\partial_z^2 + t(z)$ with the help of (20), it has holonomy given by the projectivised monodromy of the oper $\partial_z^2 + t(z)$.

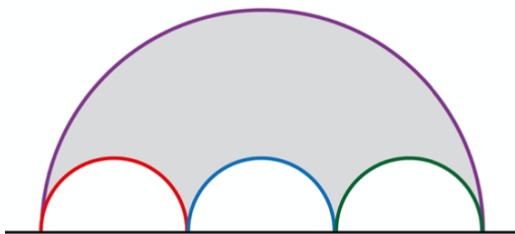

Figure 2: A fundamental domain for a once-punctured torus.

The dimension of the moduli space $\mathcal{P}(C)$ of projective structures on a surface $C = C_{g,n}$ is equal to $6g - 6 + 2n$. This dimension coincides with the dimension of the total space of the fibration formed by fibering the spaces of opers over the Teichmüller space $\mathcal{T}(C)$ of deformations of complex structures on $C$. Constructing the holonomy of an oper, or of the holonomy of the corresponding projective structure, defines a map to the character variety $\mathfrak{X}(C)$, the space of representations of the fundamental group $\pi_1(C) \to \mathrm{PSL}(2, \mathbb{C})$ modulo conjugation. These three spaces are related by local biholomorphisms relating the natural complex structures on these spaces, respectively.[13]

Real projective structures are usually defined to be projective structures having holonomy in $\mathrm{PGL}(2, \mathbb{R})$. The projective structures obtained from real opers form a subset of the set of all real projective structures characterised by the property that the holonomy is conjugate to a subgroup of $\mathrm{PSL}(2, \mathbb{R}) \subset \mathrm{PGL}(2, \mathbb{R})$.

### 4.1.1 Uniformising oper

The most basic case of a real projective structure is well-known, being related to the uniformisation of Riemann surfaces. The functions $A(y)$ defined in (20) may map $C$ to a fundamental domain in the upper half plane, with boundary curves mapped to each other by Möbius transformations being identified. A fundamental domain for a once-punctured torus is depicted in Figure 2 above.[14]

A natural hyperbolic metric on $C$ is obtained by pullback from the upper half plane metric. It can be represented in terms of the corresponding oper by finding two linearly independent solutions $\chi_i$, $i = 1, 2$, to $(\partial_y^2 + t(y))\chi(y) = 0$, and defining

$$g = \frac{dy \, d\bar{y}}{(\chi(y, \bar{y}))^2}, \qquad \chi = \mathrm{i}(\chi_1 \bar{\chi}_2 - \chi_2 \bar{\chi}_1). \tag{21}$$

It is manifest in the representation (21) that $g$ is single-valued, given that the monodromy of $\partial_y^2 + t(y)$ is contained in $\mathrm{PSU}(1, 1) \simeq \mathrm{PSL}(2, \mathbb{R})$.

### 4.1.2 Single-valuedness

We are now going to outline the argument presented in [28] that single-valued combinations of $\chi_i$ and $\bar{\chi}_j$ of the form

$$\chi = \sum_{i,j=1}^{2} c_{ij} \, \chi_i \bar{\chi}_j,$$

can only exist if the monodromy of $\partial_y^2 + t(y)$ is contained in $\mathrm{PSU}(1, 1) \simeq \mathrm{PSL}(2, \mathbb{R})$.

---

[13]We refer to [41, 42] for recent papers on this foundational issue containing discussion of previous results.
[14]Figures 2-5 are taken from [28].

Indeed, as we continue $\chi$ along non-trivial cycles on $C$, the monodromy of the solutions $\chi_i$ must preserve the hermitian form $(.,.)$. By a change of basis one can diagonalise $c_{ij}$. Depending on the signature of the hermitian form $(.,.)$, the monodromy will have to lie either in PSU(2), in PSU(1,1), or would have to be reducible.

In the first case one could represent $\chi$ in the form $\chi = |f_1|^2 + |f_2|^2$ which is everywhere positive. Inserting $\chi$ into (21) would yield a metric of constant positive curvature on $C$, contradicting the Gauss-Bonnet theorem. One may similarly argue that reducible monodromy would lead to a metric of vanishing constant curvature, again contradicting the Gauss-Bonnet theorem. The only possibility is therefore that the monodromy lies in $\mathrm{PSU}(1,1) \simeq \mathrm{PSL}(2,\mathbb{R})$.

We are thereby led to the conclusion that single-valued solutions to the oper equations and their complex conjugates are in a one-to-one correspondence with real opers.

### 4.1.3 Singularity lines

We can thus assume that the monodromy is valued in $\mathrm{PSL}(2,\mathbb{R})$ and therefore defines a *real projective structure* on $C$. Note that $\chi_i$, $i = 1, 2$, and consequently $\chi$, can be represented as

$$\chi_1 = (\partial A)^{-\frac{1}{2}}, \qquad \chi_2 = (\partial A)^{-\frac{1}{2}} A, \qquad \chi = \mathrm{i}(\chi_1 \bar{\chi}_2 - \chi_2 \bar{\chi}_1) = 2\frac{\mathrm{Im}(A)}{|\partial A|}. \tag{22}$$

$\chi$ vanishes precisely if $A(z)$ is real. This observation associates a useful piece of data to a real oper: the shape of the $\chi = 0$ locus. Indeed, in each chart of a projective atlas the $\chi = 0$ locus has the form of a straight line, with no singular points where it may be branching or ending. Globally, the locus must consist of a collection of disjoint closed curves on $C$.

The corresponding hyperbolic metric, associated to $\chi$ using (21), will be singular along the locus where $\chi$ vanishes. The resulting singularities are of a special type, being related to the divergence of the metric on the upper half plane along the real axis.

As the sign of $\chi$ changes when crossing the zero locus of $\chi$, it is necessary that the number of curves in the zero locus crossed by any closed curve on the surface is even. One may, more generally, consider the possibility that $\chi$ is only defined up to a sign. In this way one may describe the hyperbolic metrics associated to real projective structures with holonomy in $\mathrm{PGL}(2,\mathbb{R})$ which may not be conjugate to a subgroup of $\mathrm{PSL}(2,\mathbb{R})$. The number of curves in the zero locus which are crossed by any closed loop on the surface will then be unconstrained.

This naturally raises the question if one can classify real opers in terms of their singularity lines. If so, this would lead to a topological classification of the spectrum of the quantised Hitchin Hamiltonians through the analytic Langlands correspondence: The choice of a lamination would play a role analogous to the Bethe quantum numbers for more conventional integrable systems.

## 4.2 Classification of real projective structures

An important step to address the classification of real opers is a theorem of Goldman classifying real projective structures in terms of data called laminations which will be introduced next.

### 4.2.1 Classification by laminations

An integer-measured lamination $\Lambda$ is defined as the homotopy class of a finite collections of simple non-intersecting non-contractible closed curves on $C$ with integral weights, such that

- The weight of a curve is non-negative unless the curve is peripheral.
- A lamination containing a curve of weight zero is equivalent to the lamination with this curve removed.

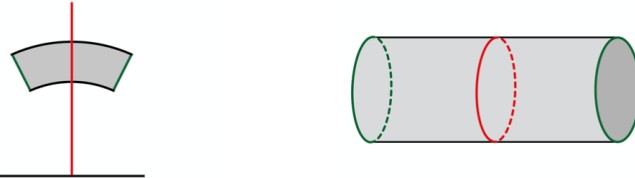

Figure 3: Left side: Representation of an annular neighbourhood of a geodesic (red) in the upper half plane model.

- A lamination containing two homotopic curves of weights $k$ and $l$ is equivalent to a lamination with one of the two curves removed and weight $k + l$ assigned to the other.

The set of all such laminations on $C$ is denoted as $\mathcal{ML}_C(\mathbb{Z})$. Half-integer measured laminations $\Lambda \in \mathcal{ML}_C(\frac{1}{2}\mathbb{Z})$ can be defined in a very similar way.

We can map a real oper to an half-integer lamination by assigning weight $\frac{1}{2}$ to simple zeros of $\chi$. The homotopy class of the zero locus thereby defines a half-integer measured lamination $\Lambda \in \mathcal{ML}_C(\frac{1}{2}\mathbb{Z})$. Goldman's theorem asserts a converse to this statement:

**Theorem 1.** *(Goldman [43]) Real projective structures are in one-to-one correspondence to half-integer measured laminations $\Lambda \in \mathcal{ML}_C(\frac{1}{2}\mathbb{Z})$.*

One mainly needs to establish the existence of a real projective structure associated to any given lamination $\Lambda \in \mathcal{ML}_C(\frac{1}{2}\mathbb{Z})$. The main tool for the proof is a geometric surgery called grafting in the mathematical literature, creating real projective structures labelled by laminations from the projective structure from uniformization.

In the rest of this subsection we will first introduce the grafting operation, and then outline how it leads to the classification of real projective structures by half-integer measured laminations.

### 4.2.2 Grafting

Grafting is a surgery operation on real projective structures. The following description is a re-interpretation of its definition in terms of the singular hyperbolic metrics associated to real projective structures according to the discussion in Section 4.1 above.

In order to define the grafting operation, one may cut $C$ open along a simple closed geodesic, and insert an arbitrary number of "trumpets" before re-gluing. A trumpet is a hyperbolic metric on an annulus bounded by a finite length geodesic on one side, and the boundary of the Poincare disc on the other side. In the upper half plane model a trumpet would be represented by the shaded region in one of the quadrants of the diagram on the left in Figure 4, bounded by the red segment representing the geodesic on one side and the blue segments representing the boundary of the Poincare disc on the other side. The boundary curves drawn in black are identified with each other to form an annulus.

Application of the grafting operation to the fundamental domain depicted in Figure 2 results in the fundamental domain depicted in Figure 5. The surfaces produced by grafting will generically cover the grey region in Figure 5 multiple times.

Given a lamination $\Lambda \in \mathcal{ML}_C(\frac{1}{2}\mathbb{Z})$, we can start from the uniformizing oper and apply the grafting operation to each homotopy class of closed curves in $\Lambda$. The result is independent of the order of the grafting operations. As intersections with the real axis represent the zero locus of $\chi$, the real projective created by grafting along $\Lambda$ structure will have zero locus given by $\Lambda$.

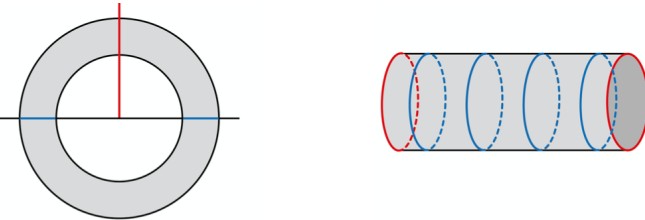

Figure 4: Result of a grafting operation producing four singularity lines.

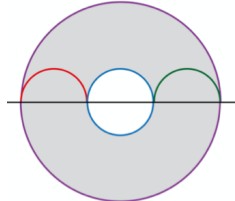

Figure 5: A fundamental domain for grafted once-punctured tori.

Conversely, one can show that starting from any real projective structure, we can remove pairs of trumpets until we reduce it to a uniformizing real projective structure which is non-singular. We can thereby demonstrate that all real opers are obtained by the grafting operation from the opers associated to the projective structures from uniformisation.

Goldman's theorem classifies real projective structures having holonomy in $\mathrm{PGL}(2,\mathbb{R})$ that may not be conjugate to a subgroup in $\mathrm{PSL}(2,\mathbb{R})$, in general. It has been pointed out by B. Goldman[15] that the subset of laminations corresponding to real projective structures with holonomy in $\mathrm{PSL}(2,\mathbb{R})$ can be described as follows. Let $H : \Lambda \in \mathcal{ML}_C(\frac{1}{2}\mathbb{Z}) \to H_1(C,\mathbb{Z}/2)$ be defined by summing the homology classes of the curves representing the elements in $\mathcal{ML}_C(\frac{1}{2}\mathbb{Z})$ with twice the assigned weights. An element $\lambda \in \mathcal{ML}_C(\frac{1}{2}\mathbb{Z})$ will then correspond to an oper with holonomy in $\mathrm{PSL}(2,\mathbb{R})$ iff $H(\lambda) = 0$. The set of all such laminations will be denoted by $\mathcal{ML}_C^0(\frac{1}{2}\mathbb{Z})$. The corresponding functions $\chi$ are single-valued on $C$, not just up to a sign.

### 4.3 Grafting opers

A projective structure canonically defines a complex structure, being represented by an atlas of a very special type with transition functions all being Möbius transformations. One should note, however, that grafting changes the atlas representing a projective structure substantially. The complex structure defined by the result of a grafting operation will differ from the complex structure defined by the projective structure to which the grafting operation has been applied.

One should keep in mind, on the other hand, that an oper is defined with respect to an underlying complex structure on $C$. The projective structure associated to a given oper using (20) will of course coincide with the complex structure underlying the definition of this oper. However, applying a grafting operation to the projective structure associated to the given oper yields a projective structure defining a complex structure which will be different from the complex structure defining the given oper. For our goals we will need a variant of the grafting operation that preserves the complex structures.

In order to define this variant, we will, following [44], introduce the projection $\pi$, mapping projective structures to the underlying complex structure. The grafting operation constructs a new projective structure from the hyperbolic metrics associated to points in the Teichmüller

---

[15]The manuscript with title "Geometric Structures And Varieties Of Representations" is available online.

space $\mathcal{T}(C)$, thereby defining a family of maps $\mathsf{Gr}_\Lambda : \mathcal{T}(C) \to \mathcal{P}(C)$, labelled by laminations $\Lambda \in \mathcal{ML}_C(\frac{1}{2}\mathbb{Z})$. $\mathsf{Gr}_\Lambda$ projects under $\pi$ to a map $\mathsf{gr}_\Lambda = \pi \circ \mathsf{Gr}_\Lambda : \mathcal{T}(S) \to \mathcal{T}(S)$. This map is known to be a diffeomorphism for each $\Lambda \in \mathcal{ML}_C(\frac{1}{2}\mathbb{Z})$ by [44, Theorem 4.3]. One may therefore consider the inverse map $\mathsf{gr}_\Lambda^{-1} : \mathcal{T}(S) \to \mathcal{T}(S)$, and form the composition

$$\sigma_\Lambda := \mathsf{Gr}_\Lambda \circ \mathsf{gr}_\Lambda^{-1}, \qquad \Lambda \in \mathcal{ML}_C(\tfrac{1}{2}\mathbb{Z}). \tag{23}$$

Acting with $\sigma_\Lambda$ on a point in $\mathcal{T}(C)$ creates a family of projective structures labelled by $\Lambda \in \mathcal{ML}_C(\frac{1}{2}\mathbb{Z})$. All these projective structures define the same complex structure. It therefore follows from Goldman's theorem that the fiber of the space of all real projective structures over a fixed complex structure $C$ is a countable discrete set isomorphic to $\mathcal{ML}_C(\frac{1}{2}\mathbb{Z})$.

## 4.4 Fenchel-Nielsen type coordinates

To each oper we may associate its holonomy, a representation of $\rho : \pi_1(C) \to \mathrm{PSL}(2, \mathbb{C})$. The space of all such representations modulo conjugation is the character variety $\mathfrak{X}(C)$. We shall here introduce coordinates called Fenchel-Nielsen (FN) type coordinates for $\mathfrak{X}(C)$ which are particularly well-suited for the description of grafting close to a degeneration limit of $C$.

### 4.4.1 Refined pants decompositions

The definition of FN type coordinates uses pants decompositions of $C$, defined by a collection of $3g - 3 + n$ simple, mutually non-intersecting closed curves $\gamma_r$, $r = 1, \ldots, 3g - 3 + n$. We will introduce seams on each of the pairs of pants appearing in the pants decomposition, a collection of three arcs connecting all different boundary components pairwise and decomposing the pairs of pants into two simply-connected regions of hexagonal shape. We assume that the pairs of pants are glued in such a way that the end-points of seams get pairwise identified.

It will the convenient to thicken the curves defining the pants decomposition into annuli. The end-points of the seams get stretched into arcs connecting the two boundaries of the annuli. In order to get a uniform description, we shall introduce additional annuli contained in sufficiently small discs around each of the $n$ punctures such that the punctures themselves are not contained in these annuli. The seams on the pairs of pants introduced above will cut these additional annuli into two halfs. The additional annuli will be called external annuli, while the annuli obtained by thickening curves defining the given pants decomposition are called internal annuli. For each internal annulus we shall choose a distinguished boundary, referred to as the inner boundary.

The result of this construction is a decomposition of $C$ into simply connected regions. Any flat connection $\nabla$ can be trivialised within such a simply connected region.

### 4.4.2 Definition of Fenchel-Nielsen type coordinates

A concrete parameterisation of the representations $\rho : \pi_1(C) \to \mathrm{PSL}(2, \mathbb{C})$ representing the holonomy of a connection $\nabla$ can be defined by choosing particular local trivialisations in the neighbourhoods of the curves separating pairs of pants and annuli. The parallel transport along a curve $\delta$ defined by $\nabla$ can then be represented by decomposing $\delta$ as concatenation of arcs contained in the annuli and pairs of pants appearing in the given pants decomposition, associating parallel transport matrices to each arc, and taking the product of these matrices in the order of appearance along $\delta$.

To each internal annulus $A$ one may associate four local trivialisations, associated to the two segments on the boundary curves separated by the seams. The parallel transport on each internal annulus is generated by two types of transport matrices. There is, on the one hand, a matrix $K_A$ representing the parallel transport from the inner boundary to the other boundary

component along a contour which does not cross a seam. One has, on the other hand, the parallel transport along the inner boundary curve of an annulus $A$. It relates the local trivialisations defined on the two segments of the boundary curves, and is represented by a matrix $L_A$. Orientations and local trivialisations can be chosen such that the two types of transport matrices $K_A$ and $L_A$ can be parameterised as follows,

$$K_A = \begin{pmatrix} e^{\kappa_A/2} & 0 \\ 0 & e^{-\kappa_A/2} \end{pmatrix} \begin{pmatrix} 0 & 1 \\ 1 & 0 \end{pmatrix}, \qquad L_A = \begin{pmatrix} e^{\lambda_A/4} & 0 \\ 0 & e^{-\lambda_A/4} \end{pmatrix}. \tag{24}$$

We are assuming that the two boundary curves of the annulus have opposite orientations, explaining the matrix $\begin{pmatrix} 0 & 1 \\ 1 & 0 \end{pmatrix}$ in (24). One thereby assigns complex numbers called FN-type length coordinates to each internal annulus. To each external annulus $A$ one can furthermore assign a complex number $\lambda_A$ allowing one to describe the parallel transport along the boundary of $A$ in the form (24).

To each pair of pants one may furthermore associate three matrices representing the parallel transport from one boundary component to another. While we won't need explicit formulae in the following, it is worth noting that these matrices can be represented as rational functions of the variables $U_A = e^{\lambda_A/2}$ associated to the annuli $A$ glued to a given pair of pants. It follows from the description above that the monodromy matrices characterising the representations $\rho : \pi_1(C) \to \mathrm{SL}(2, \mathbb{C})$ defined in this way can be represented as rational functions of the variables $U_A = e^{\lambda_A/2}$ and $V_A = e^{\kappa_A/2}$.

The holomorphic symplectic form $\Omega_{\mathfrak{X}}$ takes a particularly simple form in FN-type coordinates,

$$\Omega_{\mathfrak{X}} \propto \sum_A d\lambda_A \wedge d\kappa_A, \tag{25}$$

identifying $(\boldsymbol{\lambda}, \boldsymbol{\kappa})$ as a system of Darboux coordinates.

There exist certain one-parameter families of deformations called bending [44] which admit simple representations in terms of FN-type coordinates. Given a representation of $\pi_1(C)$ parameterised by the construction above, one may define a one-parameter deformation by replacing $\kappa_A$ by $\kappa_A + it$, $t \in \mathbb{R}$, for one of the annuli $A$ appearing on the refined pants decomposition. Grafting along a simple closed curve winding around $A$ with weight $k \in \frac{1}{2}\mathbb{Z}$ induces a bending deformation with $t = \pi k$.

## 4.5 Generating function of opers

For any fixed $C$ one may consider the image of the set of all opers within $\mathfrak{X}(C)$. This set is known to be a holomorphic Lagrangian sub-manifold $\mathfrak{Op}(C)$ within $\mathfrak{X}(C)$. Being a Lagrangian sub-manifold, there exists locally a generating function $\mathcal{Y}$ for $\mathfrak{Op}(C)$ given a choice of Darboux coordinates such as the FN coordinates introduced above.

The definition of $\mathcal{Y}$ can be refined by noting that the map associating to an oper its holonomy defines a locally biholomorphic symplectomorphism between the total space of the affine bundle $\mathfrak{Op}(C) \to \mathcal{T}(C) \simeq T^*\mathcal{T}(C)$ and the character variety $\mathfrak{X}(C)$ [45,46]. A choice of local coordinates $z_1, \ldots, z_{3g-3+n}$ for $\mathcal{T}(C)$ defines a dual basis $\{Q_1, \ldots, Q_{3g-3+n}\}$ for the cotangent fibre $T^*_C\mathcal{T}(C)$. Having chosen a reference projective structure, one may represent opers by quadratic differentials, defining coordinates $\boldsymbol{E} = (E_1, \ldots, E_{3g-3+n})$ via

$$t = t_0 + \sum_{r=1}^{3g-3+n} E_r Q_r. \tag{26}$$

The coordinates $(\boldsymbol{z}, \boldsymbol{E})$ are local Darboux coordinates for $\mathfrak{Op}(C) \to \mathcal{T}(C)$

$$\Omega = \sum_{r=1}^{3g-3+n} dz_r \wedge dE_r.$$

By introducing FN-type coordinates $(\boldsymbol{\lambda}, \boldsymbol{\kappa})$ associated to the cut system $\gamma$ we may represent the generating function $\mathcal{Y}$ of the symplectomorphism from $\mathfrak{O}\mathfrak{p}(C) \to \mathcal{T}(C)$ to the character variety $\mathfrak{X}(C)$ concretely as a function $\mathcal{Y}_\gamma(\boldsymbol{\lambda}, \boldsymbol{z})$ defined by the pair of equations[16]

$$\frac{\partial}{\partial z_r} \mathcal{Y}_\gamma(\boldsymbol{\lambda}, \boldsymbol{z}) = E_r(\boldsymbol{\lambda}, \boldsymbol{z}), \qquad r = 1, \dots, 3g - 3 + n, \tag{27a}$$

$$4\pi i \frac{\partial}{\partial \lambda_l} \mathcal{Y}_\gamma(\boldsymbol{\lambda}, \boldsymbol{z}) = \kappa_l(\boldsymbol{\lambda}, \boldsymbol{z}), \qquad l = 1, \dots, 3g - 3 + n, \tag{27b}$$

where the functions $E_r = E_r(\boldsymbol{\lambda}, \boldsymbol{z})$ are defined by the condition that the oper $\partial_y^2 + t(y)$ has monodromy with FN-type length coordinates given by the tuple $\boldsymbol{\lambda}$, and $\kappa_l(\boldsymbol{\lambda}, \boldsymbol{z})$ are the FN-type twist coordinates characterising the monodromy of $\partial_y^2 + t(y)$.

We conjecture that the functions $E_r(\boldsymbol{\lambda}, \boldsymbol{z})$ and $\kappa_l(\boldsymbol{\lambda}, \boldsymbol{z})$ defining $\mathcal{Y}_\gamma(\boldsymbol{\lambda}, \boldsymbol{z})$ up to a constant via (27) can be defined in a neighbourhood of the boundary of the Deligne-Mumford compactification associated to the cut system $\gamma$ by analytic continuation in $\lambda_l$, $l = 1, \dots, 3g - 3 + n$. These functions will not be single-valued functions of the variables $U_l = e^{\lambda_l/2}$. Existence of such an analytic continuation is supported by the analysis in the following Section 4.6, the relation to classical Virasoro conformal blocks discussed in Section 9, and the analysis in [47]. One may note that the definition of the functions $\kappa_l(\boldsymbol{\lambda}, \boldsymbol{z})$ involves an ambiguity of shifts by integer multiples of $2\pi i$ which can be fixed by requiring that $\kappa_l(\boldsymbol{\lambda}, \boldsymbol{z})$ is real on the uniformising oper.

### 4.5.1 Fenchel-Nielsen type coordinates of real opers

It follows from the above that real opers are characterised by the following system of equations,

$$\operatorname{Im}(\lambda_r) = 2\pi n_r, \qquad \operatorname{Im}(\kappa_r(\boldsymbol{\lambda}, \boldsymbol{z})) = \nu_r \pi m_r, \tag{28}$$

where

$$\nu_r = \begin{cases} 2 & \text{if } \gamma_r \text{ is non-separating,} \\ 1 & \text{if } \gamma_r \text{ is separating,} \end{cases} \qquad m_r, n_r \in \mathbb{Z}. \tag{29}$$

These equations ensure that the monodromy matrices parameterised by FN-type coordinates are real. The distinction between seperating and non-separating case is due to the fact that a given curve $\gamma$ crosses the annulus $A = A_r$ an even number of times if $\gamma_r$ is separating.

Once we have set our conventions for the generating function $\mathcal{Y}_\gamma$, each real oper defines a collection of integers $(\boldsymbol{n}, \boldsymbol{m})$, with $\boldsymbol{n} = (n_1, \dots, n_{3g-3+n})$, and $\boldsymbol{m} = (m_1, \dots, m_{3g-3+n})$. We will argue that this map is a bijection, so that the integers $(\boldsymbol{n}, \boldsymbol{m})$ are analogs of Bethe quantum numbers determining single-valued solutions to the Hitchin eigenvalue problem. We will later see that the Bethe quantum numbers coincide with the Dehn-Thurston parameters associated to the singularity lines of the hyperbolic metric defined by a real oper.

A preliminary observation is that a bending deformation indeed shifts the integers $\boldsymbol{m}$ by integral amounts.

## 4.6 Real opers and grafting lines

We will now discuss how the Bethe quantum numbers $(\boldsymbol{n}, \boldsymbol{m})$ associated to a given oper with monodromy in $\operatorname{PSL}(2, \mathbb{R})$ are related to the numbers of grafting lines.

---

[16]To avoid confusion one should bear in mind that the normalisation used in the rest of this paper differs from the one used in the introduction by a factor including the imaginary unit i.

### 4.6.1 Opers on degenerating families of surfaces

To this aim we shall consider degenerations of $C$ corresponding to a component of the boundary of the moduli space of Riemann surfaces. As a typical example we shall consider $C = C_{0,4}$. In this case we can represent opers explicity in the form $\partial_y^2 + t(y)$, with

$$t(y) = \frac{\delta_1}{y^2} + \frac{\delta_2}{(y-z)^2} + \frac{\delta_3}{(y-1)^2} + \frac{\delta_1 + \delta_2 + \delta_3 - \delta_4}{y(1-y)} + \frac{z-1}{y(y-1)}\frac{H}{y-z}. \tag{30}$$

This representation introduces coordinates $(z, H)$ for the space of pairs $(C_{0,4}, \nabla_{\mathrm{op}})$ with $C_{0,4}$ being a four-punctured sphere, and $\nabla_{\mathrm{op}}$ being an oper on $C_{0,4}$.

As an example we shall study the limit $z \to 0$. In the region on $C$ with $y = \mathcal{O}(1)$ for $z \to 0$ one may get a finite limit iff $H = h_0 + \mathcal{O}(z)$. It will be useful to represent $h_0$ in the form $h_0 = \delta - \delta_1 - \delta_2$ with $\delta \in \mathbb{C}$. The oper defined by (30) can then, to leading order in $z$, be approximated by

$$t_{0,3}(y) = \frac{\delta}{y^2} + \frac{\delta_3}{(y-1)^2} + \frac{\delta + \delta_3 - \delta_4}{y(1-y)}.$$

There is another region on $C$ where $y = wz$ with $w$ finite for $y, z \to 0$. We then have

$$z^2 t(y) = \frac{\delta_1}{w^2} + \frac{\delta_2}{(w-1)^2} + \frac{\delta_1 + \delta_2 - \delta}{w(1-w)} + \mathcal{O}(z). \tag{31}$$

This leads to a representation of the degeneration limit of $C_{0,4}$ as union of two three-punctured spheres glued at a double point.

The monodromy of $\partial_y^2 + t(y)$ around this double point is easy to compute in this representation. Representing $\delta$ in the form

$$\delta = \delta(l) := \frac{1}{4} + \left(\frac{l}{4\pi}\right)^2, \tag{32}$$

one finds two linearly independent solutions $f_1$, $f_2$ to $(\partial_y^2 + t(y))f(y) = 0$, having leading behavior of the form

$$f_1(y) = y^{\frac{1}{2}+i\frac{l}{4\pi}}(c_1 + \mathcal{O}(y)), \qquad f_2(y) = y^{\frac{1}{2}-i\frac{l}{4\pi}}(c_2 + \mathcal{O}(y)). \tag{33}$$

The monodromy around this cycle has eigenvalues $-e^{\pm\frac{\lambda}{2}}$ if we identify the parameter $l$ describing the asymptotic behavior of $H$ to be equal to the monodromy parameter $\lambda$.

### 4.6.2 Generating function for degenerating families of surfaces

The computation of the coordinate $\kappa(\lambda, z)$ turns out to be somewhat nontrivial even in the limit $z \to 0$. We shall here quote the results from [48–50],

$$\mathcal{Y}(\lambda, z) = (\delta(\lambda) - \delta(\lambda_1) - \delta(\lambda_2))\log(z) + \mathcal{Y}_0(\lambda) + \mathcal{O}(z), \tag{34a}$$

where

$$\mathcal{Y}_0(\lambda) = \frac{1}{2}\big(C_{\mathrm{cl}}(\lambda_4, \lambda_3, \lambda) + C_{\mathrm{cl}}(-\lambda, \lambda_2, \lambda_1)\big), \tag{34b}$$

with special function $C_{\mathrm{cl}}(l_3, l_2, l_1)$ defined as

$$C_{\mathrm{cl}}(l_3, l_2, l_1) = \sum_{s_1, s_2 = \pm} \Upsilon_{\mathrm{cl}}\big(\tfrac{1}{2} + \tfrac{i}{4\pi}(l_3 + s_1 l_1 + s_2 l_2)\big) - \sum_{i=1}^{3} \Upsilon_{\mathrm{cl}}(1 + \tfrac{i}{2\pi}l_i), \tag{34c}$$

using the notations

$$\log \Upsilon_{\mathrm{cl}}(x) = \int_{1/2}^{x} du \, \log \frac{\Gamma(u)}{\Gamma(1-u)}. \tag{34d}$$

Equation (34) implies the following formula for $\kappa(\lambda, z)$:

$$\kappa(\lambda, z) = \frac{\lambda}{2\pi i} \log z - \pi i + \log S_{21}(\lambda) + \log S_{43}(\lambda) + \mathcal{O}(z), \tag{35a}$$

$$S_{ij}(\lambda) := \frac{\Gamma\left(1 - \frac{i}{2\pi}\lambda\right)^2 \prod_{s_1, s_2 = \pm} \Gamma\left(\frac{1}{2} + \frac{i}{4\pi}(\lambda + s_1\lambda_i + s_2\lambda_j)\right)}{\Gamma(1 + \frac{i}{2\pi}\lambda)^2 \prod_{s_1, s_2 = \pm} \Gamma\left(\frac{1}{2} - \frac{i}{4\pi}(\lambda + s_1\lambda_i + s_2\lambda_j)\right)}. \tag{35b}$$

The corrections are expected to be represented by a convergent series in powers of $z$.

### 4.6.3 Real opers near degeneration limits

The equations

$$l := \text{Im}(\lambda) = 2\pi n, \qquad k(\lambda, z) := \text{Im}\big(\kappa(\lambda, z)\big) = \pi m, \tag{36}$$

can be used to determine real and imaginary parts of $\lambda = l + 2\pi i n$. For $z \to 0$ we find

$$-\frac{l}{4\pi} \log |z|^2 = \pi(m+1) - n \arg(z) + \mathcal{O}(l^2) + \mathcal{O}(z). \tag{37}$$

The indicated nature of the corrections follows from (35). Solutions $l$ that do not diverge for $z \to 0$ must vanish like $1/\log|z|$. The solutions to (36) take the form

$$l = l_{n,m} := \frac{2\pi}{\log(1/|z|)}\big(\pi(m+1) + n \arg(z)\big) + \dots, \tag{38}$$

$$k = k_{n,m} := \frac{\arg(z)}{\log(1/|z|)}\big(\pi(m+1) + n \arg(z)\big) + n \log|z|^2 + \dots, \tag{39}$$

with corrections in (38) vanishing for $z \to 0$ at least as fast as $1/(\log|z|)^2$. We note that the equations (36) have unique solutions for small $|z|$. To each solution (38) there corresponds an oper $\partial_y^2 + t(y)$ with $t(y)$ of the form (30) with $H = \delta(l_{n,m}) - \delta_1 - \delta_2 + \mathcal{O}(z)$ when $z \to 0$.

Recall that the corresponding hyperbolic metrics are found by forming the ratio $A = f_2/f_1$ of two solutions to $(\partial_y^2 + t(y))f(y) = 0$, and defining

$$g = \frac{dy \, d\bar{y}}{(\chi(y, \bar{y}))^2}, \qquad \chi \equiv \chi_{n,m} := 2\frac{\text{Im}(A)}{|\partial A|}. \tag{40}$$

For $z \to 0$, $y \to 0$ with $z/y \to 0$ one may use (33), and approximate $A(y)$ by $y^{i\lambda/2\pi}$, giving

$$\text{Im}(A(y)) = \exp\left(n\rho - \sigma \frac{l}{2\pi}\right) \sin\left(\frac{l}{2\pi}\rho + n\sigma\right), \qquad y = e^{\rho + i\sigma}. \tag{41}$$

The solution is simplest for $m = n = 0$. The solution (38) for $l_* \equiv l_{0,0}$ then recovers results previously derived in [51] for the uniformising metric. The approximation (41) will be useful in a range of values for $\rho$ determined by the condition that the metric should be completely smooth on $C$. This forces us to restrict attention to the annulus $\mathbb{A} = \{y = e^{\rho + i\sigma}; \rho \in I, \sigma \in [0, 2\pi)\}$, with $I$ being a subset of $(\log|z|, 0)$. The geodesic separating the punctures $z_1$ and $z_2$ from $z_3$ and $z_4$ on $C_{0,4}$ will then be approximated by the circle in $\mathbb{A}$ defined by the equation $\rho = \frac{1}{2}\log|z|$ on which the metric (40) gets minimised.

Considering cases with $(n, m) = (n, 0)$, $n \neq 0$, we may easily see that the number of zeros of $\text{Im}(A(e^{\rho + i\sigma}))$ for $\sigma \in [0, 2\pi)$ is equal to $2|n|$. By varying $\rho$, one defines singularity lines traversing $\mathbb{A}$. This is equal to twice the number of the singularity lines of corresponding hyperbolic metric on $C_{0,4}$, as each singularity line will intersect the cutting curve $\gamma$ twice.

Turning attention to the cases $(n, m) = (0, m)$, $m \neq 0$, one may notice that $\text{Im}(A(y))$ will for fixed $\sigma$ have zeros as function of $\rho$. As long as $m$ is much smaller than $\log(1/|z|)$, we find $m$ circular singularity lines in the interior of $\mathbb{A}$. It is instructive to compare these findings with the discussion in section 4.3. One may, in particular, note that

- The mapping $(l_*, k_*) \to (l_{0,m}, k_{0,m})$ can be represented as a deformation within the Teichmüller space of Riemann surfaces $C_{0,4}$ parametrised by Fenchel-Nielsen coordinates $(l, k) \in \mathbb{R}^2$ representing $\mathsf{gr}^{-1}$.

- $(l_{0,m}, k_{0,m}) \to (l_{0,m}, k_{0,m} + \pi i m)$ is the bending flow associated to $\mathsf{Gr}$.

In the general case associated to integers $(m, n)$ which are both non-vanishing, one may observe that the we still have $2|n|$ zeros of $\mathrm{Im}(A(y))$ at fixed $\rho$. However, having non-vanishing $m$ introduces an additional winding of the singularity lines around $\mathbb{A}$, related to the action of Dehn twists on $\mathbb{A}$.

The discussion above suggests that there exist unique real opers labelled by the integers $m$ and $n$ in the neighbourhood of the boundary to $\mathcal{M}_{0,4}$ associated to $z \to 0$. One may expect that this conclusion can be extended to all $z \in \mathcal{M}_{0,4}$ by real analytic continuation.

A similar analysis can be carried out for $C = C_{1,1}$, the once-punctured torus. Any grafting line will then only pass once through an annulus representing the part of the surface getting pinched in the factorization limit. For essentially the same reason one finds that monodromy matrices depend on $e^{\frac{1}{2}\kappa}$ rather than $e^\kappa$. Reality of the monodromy therefore requires that $\mathrm{Im}(\kappa) \in 2\pi\mathbb{Z}$.

By means of the gluing construction of Riemann surfaces one can then generalise this analysis to arbitrary surfaces of the type $C = C_{g,n}$.

### 4.6.4 Relation to the Dehn-Thurston parameters

One may note, in particular, that there exists a direct relation between the Bethe quantum numbers $(\boldsymbol{n}, \boldsymbol{m})$ and the Dehn-Thurston coordinates of the grafting curves. The constructions of Dehn and Thurston[17] assign collections of integers to homotopy classes non self-intersecting closed curves $\gamma$ on a Riemann surface $C$, depending on a choice of a pants decomposition of $C$. If the given pants decomposition is defined by cutting along the simple closed curves $\gamma_1, \ldots, \gamma_{3g-3+n}$, we can assign to $\gamma$ and $r \in \{1, \ldots, 3g-3+n\}$ a non-negative integer $n_r(\gamma)$ by counting the number of intersections of $\gamma$ with $\gamma_r$. To any given triple $n, n', n''$ of non-negative integers with even sum $n + n' + n''$ there exists a collection of basic arcs on a pair of pants such that $n$, $n'$ and $n''$ are equal to the numbers of intersections with the three boundary circles, respectively. Gluing pairs of pants decorated with such systems of basic arcs in such a way that the end points of basic arcs get identified with each other can produce closed curves. It can be shown that a generic closed curve $\gamma$ can be obtained from a closed curve defined by gluing arcs on three-holed spheres by doing a collection of Dehn-twists along $\gamma_1, \ldots, \gamma_{3g-3+n}$. Denoting the number of Dehn twists along $\gamma_r$ needed to represent $\gamma$ in this way by $m_r \in \mathbb{Z}$ we can assign to a closed curve $\gamma$ a collection $(\boldsymbol{n}, \boldsymbol{m})$ of integers called Dehn-Thurston parameters, with $\boldsymbol{n} = (n_1, \ldots, n_{3g-3+n})$ $\boldsymbol{m} = (m_1, \ldots, m_{3g-3+n})$.

By means of the analysis in Section 4.6.3 we are led to the conclusion that the Dehn-Thurston parameters $(\boldsymbol{n}, \boldsymbol{m})$ associated to the singularity lines of the hyperbolic metric defined by a real oper coincide with the Bethe quantum numbers associated to same oper using (28), as anticipated in the notations.[18]

---

[17] A review can be found in [52].

[18] It would be interesting to study quantum numbers associated to other canonical Darboux coordinate systems such as cluster coordinates [53] associated to a triangulation of $C$.

# 5 Gauge theory proposal for a quantum analytic Langlands program

Recall that the 4d gauge theory description of the Analytic Langlands Correspondence (ALC) involves the Kapustin-Witten A-twist of $\mathcal{N} = 4$ SYM with compact gauge group $G_c$. The $A$-twist is a $\Psi \to 0$ limit of a continuous family of twists KW[$\Psi$]. The $\Psi \to \infty$ limit recovers the $B$-twist. The KW twists have special properties when $\Psi$ is real and rational. In the following, we deform away from $\Psi = 0$ along the imaginary axis.

The theory KW[$\Psi$] occurs naturally when one attempts to define a path integral for Chern-Simons theory at generic (critically shifted) level $\kappa - \kappa_c = \Psi$ [32]. In other words, KW[$\Psi$] is equipped with a "Neumann" boundary condition which plays the role of the $B_{cc}$ boundary conditions, so that a segment compactification with Neumann b.c. at one end can define contour-path integrals with a CS action [32, 54]. As the CS action is topological, so is the Neumann boundary condition.

Mimicking the ALC setup and taking pure imaginary $\Psi$, we can use a folding trick and a $[1, -1]$ compactification with Neumann boundary conditions at both ends to engineer the path integral for a 3d "complex Chern Simons theory" with gauge group $G$ and classical action [24]:

$$\frac{\Psi}{4\pi} \int_{C \times \mathbb{R}} \omega_{\mathrm{CS}}(A) - \text{c.c.}, \tag{42}$$

where $A$ is a complex $G$ connection. We propose that this setup and its S-dual image should lead to a "quantum analytic Geometric Langlands" correspondence [18–20].

We will still refer to the Neumann boundary conditions at 1 and at $-1$ as holomorphic and anti-holomorphic, respectively. This choice of terminology now reflects the fact that gauge-invariant functionals of the complex connection $A$ define holomorphic coordinates on the moduli space of complex flat connections with respect to its natural complex structure. It is not related to a notion of holomorphicity referring to a choice of complex structure on $C$.

We can take a $\Psi \to 0$ limit of this setup to recover the ALC setup, but the limit has important subtleties. In particular, the limit must re-introduce a complex structure dependence on the Neumann boundary conditions. The limiting procedure can be described easily at the level of the 3d actions: we sent $\Psi \to 0$ while rescaling $\Psi A_z \to \varphi_z$. Applying this limit to the CS action yields the HT-BF theory action. Conversely, the 3d CS action is a deformation of the HT-BF theory action by a term of the schematic form $\Psi \mathrm{Tr} A \partial_z A$. Analogously, in 4d we can take a scaling limit which sends the (anti)holomorphic Neumann b.c. at $\Psi \neq 0$ to the (anti)holomorphic Neumann b.c. at $\Psi = 0$.

At the level of the phase space, the deformation away from $\Psi = 0$ is related to a hyper-Kähler rotation: we deform the space of Higgs bundles to the moduli space of complex flat connections, also known as the character variety $\mathfrak{X}(G, C)$ of $G$ local system on $C$. This phase space does not depend on a choice of complex structure on $C$.

Although the space of flat connections is independent of the complex structure on $C$, a choice of complex structure on $C$ allows one to define a natural polarization describing the phase space as a twisted cotangent bundle to $\mathrm{Bun}_G(C)$. Representing $\mathrm{Bun}_G(C)$ by complex gauge equivalence classes of the $(0, 1)$ component $\mathcal{A}_{\bar{z}}$ of a flat connection $\mathcal{A}$ on $C$, one may parameterize the fiber direction in this polarisation by the choices of $(1, 0)$ component $\mathcal{A}_z$, which transforms affine-linearly under holomorphic gauge transformations fixing the bundle.

Following reference [24], the space of states of complex CS theory can be represented as space of twisted half-densities on $\mathrm{Bun}_G(C)$, i.e. sections of $|K_{\mathrm{Bun}}|^{1+is}$ for real $s$, with $is = \frac{\Psi}{k_c}$. The space of twisted half-densities is equipped with a natural positive-definite inner product

when $\Psi$ is pure imaginary and can thus be completed to an Hilbert space $\mathcal{H}_s$.[19] From the 3d perspective one may note that partition functions on 3-manifolds having boundaries with boundary conditions $A_{\bar{z}}|_\partial = \mathcal{A}_{\bar{z}}^{\text{2d}}$ can be interpreted as wave-functions representing states in the Hilbert space $\mathcal{H}_s$ of quantised complex CS-theory. The wave-functions in this representation depend on the choice of a complex structure on $C$.

A 4d version of this statement requires us to elaborate a bit on the 4d lift of Dirichlet boundary conditions. First, consider $\Psi = 0$. The analytic continuation strategy applies to the HT-BF theory on a manifold with boundary as well. The Dirichlet boundary condition in the 3d theory lifts to a 4d Dirichlet boundary condition which can meet $B_{\text{cc}}$ at a corner. This Dirichlet boundary condition boundary condition is topological away from the ends of the segment $[-1, 1]$. It is defined by specifying an unitary 3d background $G_c$ connection $a^{\text{3d}}$ at the boundary, but the currents describing deformations of the choice vanish[20] and thus the specific choice of $a^{\text{3d}}$ only matters up to homotopy. At the corner with the $B_{\text{cc}}$ boundary the holomorphic components of the currents survives and thus we have a dependence on the $(0, 1)$ part $\mathcal{A}_{\bar{z}}^{\text{2d}}$ of $a^{\text{3d}}$.

Turning on $\Psi$ does not change these statements. While both $B_{\text{cc}}$ and 4d Dirichlet admit topological descriptions, the corners where they can meet require a choice of complex structure.

We will now employ the 3d and 4d perspectives to formulate the properties of $\mathcal{H}_s$ which appear to be the most central ingredients of the quantum analytic Langlands correspondence. As we make $\Psi$ non-zero, the twisted theory seems to loose both the Hitchin Hamiltonians and the Hecke operators, in the sense that the corresponding local operators cease to be gauge-invariant. We will see shortly how the Hamiltonians re-emerge in a different guise, but this already indicates some important qualitative differences between the cases $\Psi = 0$ and $\Psi \neq 0$.

On the other hand, the theory at $\Psi \neq 0$ also gains new observables: topological Wilson lines supported on the Neuman boundary conditions. More precisely, we gain "holomorphic" Wilson lines built from $A$ supported at one boundary, and "anti-holomorphic" Wilson lines built from $\bar{A}$ and supported at the other boundary of the $[-1, 1]$ segment. Again, the nomenclature only refers to the fact that these observables, which are topological in the 3d geometry, depend holomorphically or anti-holomorphically on the connection $A$. We can wrap them on closed curves or certain networks[21] $a$ on $C$ in order to define operators $W_a$ and $\widetilde{W}_a$ acting on $\mathcal{H}_s$. Classically, i.e. in an $s \to \infty$ limit, these become represented as multiplication with holonomies of $A$ and $\bar{A}$, respectively. We anticipate that we can label the operators $W_a$ and $\widetilde{W}_a$ by networks $a$ in such a way that classically $\widetilde{W}_a = W_a$ whenever the connection $A$ is unitary.

Quantum-mechanically, the operators $W_a$ and $\widetilde{W}_a$ will define two copies of the quantised algebra of holomorphic functions $\mathcal{O}_q(\mathfrak{X})$ and $\widetilde{\mathcal{O}_q(\mathfrak{X})}$ on the character variety $\mathfrak{X}$, respectively, $q = e^{\frac{2\pi}{s}}$. One may note that $\widetilde{\mathcal{O}_q(\mathfrak{X})} \simeq \mathcal{O}_{q^{-1}}(\mathfrak{X})$. We shall use the notation $\mathfrak{A}_q \simeq \mathcal{O}_q(\mathfrak{X}) \times \widetilde{\mathcal{O}_q(\mathfrak{X})}$ for the algebra generated from both $W_a$ and $\widetilde{W}_a$. Determining the action of these operators on $\mathcal{H}_s$ will be an important part of the quantum analytic Langlands correspondence.

As the holomorphic and anti-holomorphic Neumann boundary conditions are equivalent in KW[$\Psi$] for generic $\Psi$, we can define a state for the system on $[-1, 1] \times C$ as a path integral over an $HD^2 \times C$ geometry, where $HD^2$ is an half-disk with Neumann boundary conditions on the circumference. We will denote this very special state as $|1\rangle$. Properties of this state are discussed at great length in the companion paper [25]. By construction, this state will relate

---

[19]As for $\mathcal{H}_0$, a careful definition requires one to deal with the singularities of Bun. For example, we could start with twisted half-densities on the space of very stable bundles, etc.

[20]More precisely, they are BRST-exact in the twisted theory.

[21]We use the term "network" to refer to a graph on $C$ whose edges are open Wilson lines in various representations and vertices are gauge-invariant intertwiners.

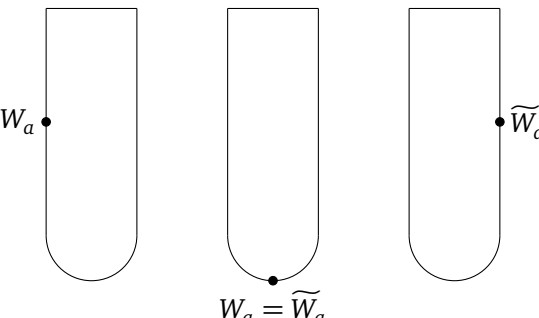

Figure 6: Holomorphic and anti-holomorphic Wilson lines with appropriately related labels act in the same manner on the state $|1\rangle$ created by the half-disk and define the $|a\rangle$ states.

the action of space-like holomorphic and anti-holomorphic Wilson lines as follows:

$$W_a|1\rangle = \widetilde{W}_a|1\rangle \equiv |a\rangle \,. \tag{43}$$

The thesis of our companion paper [25] is that $\mathcal{H}_s$ is the closure of the span of this collection of states. In particular, this statement fully characterizes the action of the algebra $\mathfrak{A}_q$ on $\mathcal{H}_s$. An abstract definition of $\mathcal{H}_s$ as a space of states of complex CS theory on $C$, or of the KW theory on $[-1, 1] \times C$, should therefore depend only topologically on $C$. This is not manifest in the representation introduced above, since $\mathcal{H}_s$ was defined by a polarization which depends on the complex structure on $C$. We will momentarily review how the complex structure dependence is controlled by the KZB equations [24].

Much of the rest of the paper is devoted to understanding how the wave-functions of $|a\rangle$ can be represented as twisted half-densities $\psi_{s,a}$ in $\mathcal{H}_s$ and how they behave as $s \to 0$.

In the case $G = \mathrm{SL}(2)$ studied in the rest of this paper one may identify the algebra $\mathcal{O}_q(\mathfrak{X}_C)$ as the skein algebra studied in [55–57]. This algebra has a linear basis labelled by the same laminations $\Lambda$ we had employed to label real opers. We will often use $\Lambda$ as a more concrete piece of data replacing the previously used labels $a$ in the following. One of the main objectives of the current paper is to describe the wave-functions $\psi_{s,\Lambda}(\boldsymbol{x}; \boldsymbol{z}) \in \mathcal{H}_s$ for the corresponding states $|\Lambda\rangle$ in a polarization defined by a family of complex structures on $C \equiv C_{\boldsymbol{z}}$ which is parameterised by a collection of variables $\boldsymbol{z} = (z_1, \ldots, z_{3g-3+n})$, and to relate them to the Hecke eigenfunctions $\psi_\Lambda(\boldsymbol{x}; \boldsymbol{z}) \in \mathcal{H}_0$ labelled by the same laminations. The statement that the $\psi_{s,\Lambda}$ give a basis for $\mathcal{H}_s$ is thereby recognised as a quantum analogue of the relation between real opers and eigenstates of the Hitchin Hamiltonians.

Finally, S-duality relates this to a setup in the Kapustin-Witten B-twist of $\mathcal{N} = 4$ SYM with Langlands dual compact gauge group $G_c^\vee$. S-duality of the physical theories descends to a relation between KW twists for Langlands dual groups with inverse values of $\Psi$.[22] S-duality maps the topological Neumann boundary conditions to topological "Nahm pole" boundary conditions. At the end of this section we will apply the analytic continuation technology along the time direction to arrive at an alternative duality statement: $\mathcal{H}_s$ controls the path integration contours for Liouville/Toda theory on $C$.

## 5.1 Unitary flat connections and boundary states

In order to make the topological nature of 3d CS theory manifest, we can look for boundary conditions which are topological and do not require a choice of complex structure. By

---

[22]Up to a numerical factor for non-simply-laced gauge groups. In the following, we will assume $G$ simply laced.

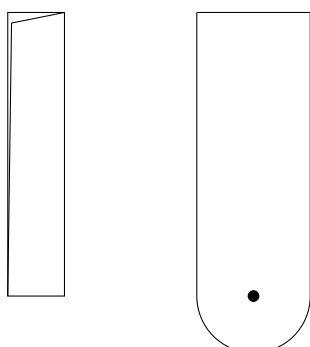

Figure 7: The $\frac{\mathbb{R}\times[-1,1]}{\mathbb{Z}_2}$ geometry realizing the 4d lift of $B_c$ has an orbifold singularity. It can be smoothed out to the shape on the right hand side of the picture, with the orbifold singularity mapped to an interior point. We pick the initial corner so that no co-dimension 2 defect is present at the interior point.

construction, such a boundary condition $B_{3d}$ defines a boundary state $|B_{3d}\rangle \in \mathcal{H}_s$ which is a one-dimensional representation of the mapping class group. If $B_{3d}$ supports topological line defects, they can be wrapped on curves or networks on $C$ to give rise to a larger collection of boundary states. These states generate a non-trivial representation of the mapping class group, induced from the realisation of the mapping class group as diffeomorphisms of $C$.

Topological boundary conditions in Chern-Simons theory can be produced by restricting the $G$ connection to a subgroup $H$ such that the CS form restricted to $H$ vanishes. In particular, we can pick $H = G_c$: the compact real form of $G$. Classically, the resulting boundary condition $B_c$ gives the space of unitary flat connections on $C$ as a Lagrangian sub-manifold of the character variety. As the space of unitary flat connections is compact, we expect the associated boundary state to be actually normalizable, rather than distributional.

We can now lift the 3d $B_c$ boundary condition to 4d via the analytic continuation construction. Notice that $B_c$ essentially identifies the connection $A$ with its complex conjugate. We thus have a $\mathbb{R}^+\times[0,1]$ geometry with two copies of the 4d theory glued together both along $\mathbb{R}^+\times 0$ and along $0\times[0,1]$. We can also describe this theory as a single copy of the 4d theory placed on an "orbifold" $\frac{\mathbb{R}\times[-1,1]}{\mathbb{Z}_2}$ geometry.

This geometry has a "topological" corner at $0\times 0$. This sort of $\mathbb{Z}_2$ orbifold geometries were consider in [17], although involving a reflection on $C$. They require one to specify an automorphism $g$ the Lie algebra to describe the gluing of the gauge fields. In that language, the $B_c$ boundary condition corresponds to $g = 1$.[23] Precisely for this choice of $g$, it was argued in [17] that the "corner" can be smoothened out. The configuration is therefore topologically equivalent to the smooth half-disk which produces the $|1\rangle$ state. We have found a way around a problematic aspect of the direct definition of $|1\rangle$ as a half-disk path integral: it does not take the form of an "analytic continuation of path integral" configuration and thus does not have an immediate 3d interpretation.

The main payoff of the 3d presentation of $|1\rangle$ is that we can draw from the theory of boundary conditions in Chern-Simons theory to identify the wave-function for $|1\rangle$ with the partition function of a 2d CFT: the WZW model with target $G/G_c$. This is called the $H_3^+$ WZW model for $G = SL(2)$ and is closely related to Liouville theory. With a bit more effort, one can also identify the wave-functions for $|a\rangle$ as partition functions decorated by certain topological line defects, also known as "Verlinde line defects" [58–60].

---

[23]For other choices of $g$ we will get a restriction to other real forms $G_{\mathbb{R}}$ of $G$. It would be interesting to explore the corresponding distributional boundary states and their duality properties, e.g. in relation to Lorentzian 3d quantum gravity with positive cosmological constant.

## 5.2 Nahm vs Dirichlet

It will be instructive to introduce one more element in in our discussion of the appearance of 2d CFTs partition functions in our story. Namely, there is a family of variants of Dirichlet boundary conditions called Nahm pole boundary conditions, which are labelled by an $\mathfrak{sl}(2)$ embedding $\rho$ in the Lie algebra of $G$. In the following we will mostly focus on the Nahm pole corresponding to a regular embedding and refer to it simply as a Nahm pole boundary condition.

The Nahm pole boundary conditions admit corners with (anti)holomorphic Neumann boundary conditions for every $\rho$, and can arise from an analytic continuation construction. These corners, as well as the corresponding 3d boundary conditions in complex HT-BF theory, support a quantum Drinfeld-Sokolov reduction of critical Kac-Moody, a W-algebra $W_G^{\rho}$. Turning on $\Psi$ makes the Kac-Moody level non-critical [61,62]. 't Hooft line defects survive at a Nahm pole boundary condition, giving rise to degenerate primary fields in the W-algebra. We thus get a family of distributional boundary states $|W_{\boldsymbol{u}}\rangle$, $\boldsymbol{u} = \{u_1, \ldots, u_d\}$, created by the regular Nahm pole boundary condition modified at points $u_i$, $i = 1, \ldots, d$, by 't Hooft operator insertions.

A prescient paper [63] employed this setup to engineer the 2d Liouville CFT within the KW[$\Psi$] theory for $G = SL(2)$, with $c = 13 + 6\Psi + 6\Psi^{-1}$. In our language:

$$Z_{\text{Liou}} = \langle W|1\rangle, \qquad \text{with} \qquad W \equiv W_{\emptyset}. \tag{44}$$

This is expected to generalize to Toda theory for other $G$'s. The pairing $Z_{\text{Liou}}(\boldsymbol{u}) = \langle W_{\boldsymbol{u}}|1\rangle$ will be related to Liouville correlation functions with certain degenerate field insertions.[24]

Our claim is similar in nature: the partition function of the WZW model is obtained by pairing $|1\rangle$ with Dirichlet boundary conditions

$$Z_{\text{WZW}}(\boldsymbol{x}) = \langle \text{Dir}_{\boldsymbol{x}}|1\rangle, \tag{45}$$

i.e. the WZW partition function is the wave-function for $|1\rangle$ in a holomorphic polarization, with $\boldsymbol{x} = (x_1, \ldots, x_d)$ being coordinates on $\text{Bun}_G$ for $G = SL(2)$. This is a non-compact example of relations between WZW models and Chern-Simons theories. As a simple semiclassical check (at large $s$), observe that the problem of finding classical solutions of the equations of motion with the specified boundary conditions is the same as finding a flat $G$ connection on $C$ which is unitary, and projects to a specified bundle. This problem is precisely solved by the equations of motion of the $G/G_c$ WZW model, as reviewed later.

We will later discuss also relations between $Z_{\text{WZW}}(\boldsymbol{x})$ and $Z_{\text{Liou}}(\boldsymbol{u})$ of the form

$$Z_{\text{WZW}}(\boldsymbol{x}) = \int d\mu(\boldsymbol{u}) \, K(\boldsymbol{x}, \boldsymbol{u}) \, Z_{\text{Liou}}(\boldsymbol{u}), \tag{46}$$

known to exist for $G = SL(2)$ if $d = 3g - 3 + n$. The context of quantised complex CS theory suggests to interpret the kernel $K(\boldsymbol{x}, \boldsymbol{u})$ as the overlap $\langle \text{Dir}_{\boldsymbol{x}}|W_{\boldsymbol{u}}\rangle$.

Both constructions allow for the inclusion of Wilson lines. We will argue that $\langle W_{\boldsymbol{u}}|a\rangle$ and $\langle \text{Dir}_{\boldsymbol{x}}|a\rangle$ are partition functions modified by Verlinde line defects and study their $\Psi \to 0$ limit.

## 5.3 Polarization dependence and the KZ equations

We should elaborate on the physically expected polarization-independence of $\mathcal{H}_s$, following [24]. An important feature of the description of $\mathcal{H}_s$ as a space of twisted half-densities

---

[24]The symbol "W" for the states is motivated by an expected interpretation as an analog of the Whittaker functionals appearing in other variants of the Langlands correspondence.

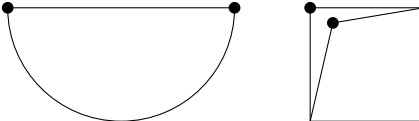

Figure 8: Left: the configuration employed to define 2d CFTs. The curved edge has Neumann b.c., the straight edge has either Nahm or Dirichlet b.c. The chiral and anti-chiral algebras live at the corners. Right: a more symmetric version of the same geometry, described as $\frac{[-1,1]\times[-1,1]}{\mathbb{Z}_2}$.

is that it admits a straightforward $s \to 0$ limit. The price to pay is that the definition depends on the choice of complex structure on $C$, even though the complex Chern-Simons theory should be topological. The complex structure dependence is locally trivialized by a unitary flat connection on the moduli space of complex structures. This connection can be described in a 2d CFT language. It is closely related to the KZB connection defined on the spaces of Kac-Moody conformal blocks using the Sugawara construction, having the schematic form

$$\nabla_{\text{KZB}} = \mathrm{i}s\frac{\partial}{\partial z_r} - \mathsf{H}_{r,s}\,. \tag{47}$$

$\nabla_{\text{KZB}}$ controls the behaviour of wave-functions under changes of polarization induced by a change $\delta z_r$ in the complex structure of $C$, with $z = (z_1,\ldots,z_{3g-3+n})$ being local coordinates for the Teichmüller space of $C$, and $\mathsf{H}_{r,s}$ being second order differential operators acting on sections of $|K_{\text{Bun}}|^{1+\mathrm{i}s}$, deforming the Hitchin Hamiltonians $\mathsf{H}_r$ in the sense that $\mathsf{H}_{r,0} = \mathsf{H}_r$.

If $\mathcal{H}_s$ is polarization-independent, the KZB connection should preserve the inner product on $\mathcal{H}_s$. This statement generalizes the normality of $\mathsf{H}_r$ acting on $\mathcal{H}_0$. We propose that one of the problems in a quantum analytic Langlands program should be to prove the unitarity of the KZB connection on $\mathcal{H}_s$ and to compute the resulting representation of the mapping class group of $C$.

We have formally introduced the states $|a\rangle \in \mathcal{H}_s$. We will concretely represent them by wave-functions $\psi_{s,a}$ which will be identified with the partition functions of the $G/G_c$ WZW model decorated by Verlinde line defects. This representation implies that the wave-functions satisfy the KZB equations. Furthermore, the mapping class group acts on these states by the natural action on the curves defining $a$. It is natural to wonder if this information is sufficient to actually demonstrate unitarity of the KZB equation and describe the action of the mapping class group. This would be the case if $|a\rangle$ provided a complete basis of normalizable states with norms which are independent of the complex structure on $C$. We will provide arguments in support of this conjecture both in this paper and in the companion paper [25].

## 5.4 The action of Wilson/Verlinde lines

As discussed, we expect that the algebra of observables is generated by quantized analogs $W_a$ and $\widetilde{W}_a$ of the traces of holonomies defined from complex connections $\mathcal{A}$ and $\bar{\mathcal{A}}$ on a Riemann surface $C$, respectively. This algebra can be identified with the algebra $\mathfrak{A}_q = \mathcal{O}_q(\mathfrak{X}_C) \times \overline{\mathcal{O}_q(\mathfrak{X}_C)}$, where $\mathcal{O}_q(\mathfrak{X}_C)$ is the algebra of quantised functions on the character variety $\mathfrak{X}$. The algebra $\mathcal{O}_q(\mathfrak{X}_C)$ for $G = \text{SL}(2)$ has a natural integral linear basis labelled by laminations $\Lambda$. We thus have states $|\Lambda\rangle$ labelled by laminations, acted by the mapping class group in a natural way.

This raises an obvious question: How is the algebra $\mathfrak{A}_q$ represented on the Hilbert space $\mathcal{H}_s$ defined by the complex-structure dependent polarization?

One might approach the definition by generalising a strategy that has been successfully applied to Chern-Simons theories associated to compact groups. It is based on the observation that topological invariance should allow us to elongate the closed curves $\gamma$ in the time direction,

describing them as a braid along time closed by cups and caps in the past and future. This leads to a description of the operators $W_a$ and $\widetilde{W}_a$ as a composition of maps between spaces of states assigned by the Chern-Simons theory to surfaces with varying number of punctures.

In our case one has to note that the space of states for a system with time-like Wilson lines lacks a positive-definite inner product, as finite-dimensional representations of $SL(2,\mathbb{C})$ are not unitary. Instead, it has a pairing with a dual vector space. As a consequence, one cannot work with Hilbert spaces at intermediate steps of the calculation. The proper functional-analytic framework for such definitions remains to be found.

It seems nevertheless reasonable to expect that the wave-functions in such a representation will behave as conformal blocks of Kac-Moody currents in the presence of Weyl modules associated to the finite-dimensional representations. The space of such wave-functions will be equipped with a KZB connection allowing for the parallel transport of these new punctures on $C$. Pairwise creating, braiding and annihilating such modules then closely resembles the definition of Verlinde lines operators for a rational 2d CFT [58]. This leads us to expect that the application of a variant of the definition of Verlinde line operators in the $G/G_c$ WZW model should provide a definition of the operators $W_a$ and $\widetilde{W}_a$ in complex Chern-Simons theory.

We will verify later that this approach defines operators having the expected properties. We can apply this idea, in particular, to the WZW partition function as a wave-function. We will see that a remarkable property of the WZW partition function is precisely that the action of holomorphic and anti-holomorphic Verlinde line defects coincide. This is consistent with the proposal that the WZW partition function represents the wave-function of the state $|1\rangle$. Analogously, the WZW partition function with a Verlinde line insertion $W_a$ should give the wave-function of $|a\rangle$.

## 5.5 S-duality and thimbles

We reformulate the 4d description of

$$Z_{\text{Toda}} = \langle W | 1 \rangle, \tag{48}$$

as the partition function on the orbifold geometry $\frac{[-1,1]\times[-1,1]}{\mathbb{Z}_2} \times C$ with Neumann b.c. for the first factor in the quotient and Nahm pole b.c. for the second factor.

As S-duality permutes Neumann and Nahm pole boundary conditions, this is a very economical demonstration of self duality of Toda: Toda theory at level $\Psi$ associated to a Lie group $G$ coincides with the theory for Langlands dual groups and levels $\Psi^\vee$, $G^\vee$.

S-duality provides an alternative description of $\mathcal{H}_s$: the space of states of the S-dual KW theory on $[0,1]\times C$, with Nahm pole boundary conditions. In the S-dual context, we can apply the analytic-continuation-of-path-integral perspective along the time direction, with a space-like Neumann boundary condition: $\mathcal{H}_s$ is thus described as a space of integration contours for the path integral of $G^\vee$ Chern-Simons theory at level $\Psi^\vee$ on $[-1,1]\times C$. The boundary conditions for the interval fix the connection to be an oper or an anti-oper, as in the B-twist setup.

The condition for a flat connection to be an oper and an anti-oper leads to real opers and classical solutions of Liouville/Toda theory. For $s=0$ each classical solution gives a ray in $\mathcal{H}_0$. For $s \neq 0$, instead, each classical solution is mapped to the corresponding "thimble" path-integration contour in $\mathcal{H}_s$ for the $G^\vee$ Chern-Simons theory. Based on the realization of quantum Liouville/Toda in the 4d setting, we expect that the "thimble" path integral contour corresponds to a modification $Z_{\text{Toda}}^{\text{thimble}[a]}$ of the partition function $Z_{\text{Toda}}$ depending on the lamination $\Lambda$,

$$\langle \text{Neumann} | \text{Real Oper}[\Lambda] \rangle = Z_{\text{Toda}}^{\text{thimble}[\Lambda]}, \tag{49}$$

with $s \to 0$ asymptotics controlled by the real oper. This is a 4d argument predicting the existence of states in $\mathcal{H}_s$ with $s \to 0$ asymptotics controlled by the real oper associated to $\Lambda$. We are going to propose that the partition functions $Z_{\text{Toda}}^{\text{thimble}[\Lambda]}$ can be represented as Liouville/Toda partition function modified by the insertion of a collection of Verlinde lines described by $\Lambda$. The 2d CFT analysis later in the paper matches $|\Lambda\rangle \in \mathcal{H}_s$ with the thimble state associated to the real oper labelled by $\Lambda$:[25]

$$|\Lambda\rangle = |\text{Real Oper}[\Lambda]\rangle . \tag{50}$$

In particular, for $G = \text{SL}(2)$,

$$|1\rangle = |\text{Uniformizing Real Oper}\rangle . \tag{51}$$

On the $B$ side, $|1\rangle$ is engineered from the "smooth" ending for the $[-1, 1] \times C$ geometry, with the oper and anti-oper boundary conditions recognized as equivalent versions of the same Nahm pole boundary condition. The states $|\Lambda\rangle$ are obtained by adding 't Hooft lines.

In such a situation one might generically expect that the basis of thimble states in $\mathcal{H}_s$ could jump across Stokes lines where the action of different solutions have the same real part. It is part of our proposal above that this does not happen. As discussed in more detail later, the definition of $Z_{\text{Toda}}^{\text{thimble}[\Lambda]}$ as Liouville partition function modified by the insertion of Verlinde lines admit real analytic continuations over the Teichmüller spaces of Riemann surfaces. The same is true for the real opers labelled by a lamination $\Lambda$. We are proposing that the real oper associated to $\Lambda$ controls the asymptotics $s \to 0$ of $Z_{\text{Toda}}^{\text{thimble}[\Lambda]}$ over all of Teichmüller space.

The following heuristic argument offers a hint why this can be expected to hold in this case. Transformations such as the mapping class group action on $\mathcal{H}_s$ would be decomposed into a "formal monodromy" keeping track of semi-classical signs and potential multi-valuedness of the action, combined with the integral Stokes matrices describing changes of the basis of thimbles. The multi-valuedness of the Chern-Simons action only gives factors of $q = e^{2\pi i \Psi^\vee}$, which is a real number for imaginary level. Here we are in a very special situation: $\mathcal{H}_s$ is an Hilbert space and monodromies act by unitary transformations. It is very hard to build unitary monodromy matrices with coefficients which are integral-valued polynomials of an arbitrary real variable $q$. We thus expect Stokes phenomena not to occur for the path integral of Liouville/Toda as long as we keep the level pure imaginary. In particular, the "real oper" thimble basis should be canonical and single-valued on Teichmüller space and the monodromy of the KZ connection should take the form of a permutation matrix acting on the basis of thimbles.

## 5.6 Final statement of the conjecture

In the case $G = \text{SL}(2)$ we are thus led to the conclusion that $\mathcal{H}_s$ includes a collection of states $|\Lambda\rangle$ having wave-functions given by the 2d CFT correlation functions $\Psi_{s,\Lambda}$, labelled by laminations $\Lambda$, covariant under the KZB connection and acted upon naturally by the mapping class group. Furthermore, these states have an $s \to 0$ limit which gives a basis of $\mathcal{H}$. It is then natural to conjecture that the $|\Lambda\rangle$ give a basis for $\mathcal{H}_s$ for all values of $s$.

This conjecture has several non-trivial aspects. Even normalizability of $|\Lambda\rangle$ is far from obvious. One would need to understand the possible singular loci of sections $\Psi_{s,\Lambda}$ within $\text{Bun}_G$. Such singular loci are determined by the singularities of the KZB equations. It is known that singularities can occur at the so-called wobbly bundles, bundles admitting nilpotent Higgs fields. Normalisability has been verified in [30] in some non-trivial cases. An alternative approach could be based on the expected unitarity of the SOV-transformation.[26]

---

[25] Up to a small ambiguity which is compatible with the mapping class group action on the two sides.

[26] D. Kazhdan, private communication, and work in progress by F. Ambrosino, D. Gaiotto, J. Teschner.

Remembering that the laminations $\Lambda$ classify real opers with the help of grafting, we can schematically represent the conjectured correspondence between real opers and square-integrable sections $\Psi_{s,\Lambda}$ in a form which resembles (10), as a correspondence between

$$
\boxed{\text{Real opers}} \quad \longleftrightarrow \quad \boxed{\begin{Bmatrix} \text{single-valued and} \\ L^2\text{-normalisable} \end{Bmatrix} \begin{matrix} \text{solutions to the} \\ \text{KZB-equations.} \end{matrix}} \tag{52}
$$

We will refer to this correspondence as quantum analytic Langlands correspondence.

In our companion paper [25], we consider a disk geometry, consisting of two half-disks glued together. This computes inner products $\langle 1|1 \rangle$ or more generally $\langle \Lambda'|\Lambda \rangle$. We argue that it coincides with the Schur index of class S theories, leading to a natural algebraic definition of $\mathcal{H}_s$ and of the $\mathcal{O}_q(\mathfrak{X}_C) \times \widetilde{\mathcal{O}_q(\mathfrak{X}_C)}$ action which is independent of the complex structure on $C$. In particular, this identification supports the conjecture that the $|\Lambda\rangle$ states are square-normalizable.

# 6 WZNW model associated to hyperbolic three-space

For $G = SL(2)$, the quotient $G/G_c = SL(2)/SU(2)$ is the hyperbolic space $H_3^+$. This subsection will offer a very brief summary of relevant results and conjectures on the WZNW model with target $H_3^+$. The heuristics offered by the path integral for the $H_3^+$ WZNW model will be translated into the mathematical problem to find single-valued real-analytic solutions of the Knizhnik-Zamolodchikov-Bernard (KZB) equations. The solutions of this problem called correlation or partition functions characterise the $H_3^+$ WZNW model completely.

We will furthermore introduce certain generalisations of the correlation functions labelled by half-integer measured laminations. The generalised correlation functions will form the basis for the generalisation of the analytic Langlands correspondence proposed in this paper.

## 6.1 $H_3^+$ WZNW model on surfaces of genus zero

We will start by formulating expectations from theoretical physics based on the path integral. The description will be schematic, referring to [64–66] for further background.

The CFT called $H_3^+$ WZNW model is characterised by the collection of its correlation functions. The path integral is expected to provide a definition of the correlation functions schematically represented as

$$
\mathcal{G}(\boldsymbol{z}, \boldsymbol{x}; \boldsymbol{j}) := \left\langle \prod_{k=1}^{n} \Theta^{j_k}(x_k|z_k) \right\rangle_{\mathrm{W}} = \int_{h:C \to \mathbb{H}_3^+} \mathcal{D}[h]\, e^{-S_{\mathrm{WZ}}[h]} \prod_{k=1}^{n} \Theta^{j_k}(h(z_k); x_k), \tag{53}
$$

where

$$
S_{\mathrm{WZ}}[h] = -\frac{k}{\pi} \int dx \left( \partial\phi \bar{\partial}\phi + |\bar{\partial}\gamma|^2 e^{2\phi} \right), \qquad h = \begin{pmatrix} e^{-\phi} + |\gamma|^2 e^{\phi} & \bar{\gamma}e^{\phi} \\ \gamma e^{\phi} & e^{\phi} \end{pmatrix}, \tag{54}
$$

and the fields get represented by the functions $\Theta^j(h; x)$ defined as

$$
\Theta^j(h; x) = \frac{2j+1}{\pi} \left( (1, -x) \cdot h \cdot \begin{pmatrix} 1 \\ -\bar{x} \end{pmatrix} \right)^{2j}. \tag{55}
$$

The matrices $h$ are hermitian, and have determinant equal to one, representing elements of the three-dimensional hyperbolic space $H_3^+$. This space is a symmetric space with symmetry group

$\mathrm{SL}(2) \equiv \mathrm{SL}(2,\mathbb{C})$ acting as $h \mapsto g^{-1} \cdot h \cdot (g^{\dagger})^{-1}$. This symmetry gets enhanced to a symmetry represented by a central extension of the loop group $\widetilde{\mathrm{SL}}(2)$ in the $H_3^+$ WZNW model.

The notations used in (53) are adapted to the case of Riemann surfaces $C = C_{0,n}$ of genus 0 with $n$ punctures at the positions $z_1, \ldots, z_n$. Equation (55) identifies the variables $x$ and $\bar{x}$ as auxilliary parameters, allowing us to represent the transformation law of the generating functions $\Theta^j(h; x)$ under the SL(2)-symmetry in the form

$$
\Theta^j(h; x) \mapsto \Theta^j\big(g^{-1}h(g^{\dagger})^{-1}; x\big) = |d + cx|^{4j}\,\Theta^j\Big(h; \frac{ax+b}{cx+d}\Big), \qquad g = \begin{pmatrix} a & b \\ c & d \end{pmatrix}, \tag{56}
$$

characteristic for the spherical principal series $\mathcal{P}_j$ of the group $\mathrm{SL}(2,\mathbb{C})$ if $j \in -\frac{1}{2} + i\mathbb{R}$.

The $H_3^+$ WZNW model has the parameter $k$. For many applications one is interested in real values of $k \in (-\infty, -2)$, but let us anticipate that we will later consider the analytic continuation to values where $k + 2 \in i\mathbb{R}$.

## 6.2 $H_3^+$ WZNW model on surfaces of higher genus

When the surface $C = C_{g,n}$ has genus $g > 0$, it is natural to generalise the definition of correlation functions by twisting with holomorphic $G = \mathrm{SL}(2)$-bundles. Describing holomorphic bundles $\mathcal{E}$ in terms of an atlas of local trivialisations, one assigns fields $h_U \in H_3^+$ to open subsets $U \subset C$, related to each other as $h_{U'} = g_{U'U} \cdot h_U \cdot g_{UU'}^{\dagger}$, with $g_{UU'}$ being the transition function of the atlas defining $\mathcal{E}$ on the intersection of the open sets $U$ and $U'$.

It is possible to construct families $\mathcal{E}_x$ of holomorphic $SL(2)$-bundles labelled by collections $x = (x_1, \ldots, x_{3g-3+n})$ of complex parameters. This can be done in such a way that $n$ out of the $h := 3g - 3 + n$ parameters coincide with the variables $x_1, \ldots, x_n$ introduced with the help of (53) and (55). We will in the following use the notation $x$ for tuples of complex local coordinates on the moduli space $\mathrm{Bun}_G$ of $G = SL(2)$-bundles generalising the variables $x_1, \ldots, x_n$ explicitly introduced using (53) and (55).

The variables $z_1, \ldots, z_n$ used in (53) get generalised to $h = 3g - 3 + n$ complex parameters $z = (z_1, \ldots, z_h)$ for families of complex structures of $C$ if $C = C_{g,n}$ has $g > 0$. The correlation functions defined in (53) are generalised to functions $\mathcal{G}(x, z; j)$ of $2h$ complex variables $x$ and $z$, parametrically depending on $j = (j_1, \ldots, j_n)$. Away from certain singular loci, one expects a dependence with respect to $x = (x_1, \ldots, x_h)$ and $z = (z_1, \ldots, z_h)$ which is real-analytic, but not holomorphic.

The heuristics provided by (53) leads to the formulation of a precise mathematical problem using the affine Lie algebra symmetry discussed next.

## 6.3 Affine Lie algebra symmetry

The $\mathrm{SL}(2,\mathbb{C})$-symmetry of the $H_3^+$-model is accompanied by a symmetry under the Lie-algebra $\mathfrak{sl}(2,\mathbb{C})$ which becomes enhanced to a symmetry under the direct sum of affine Lie algebras $\widehat{\mathfrak{sl}}_{2,k} \oplus \widehat{\mathfrak{sl}}_{2,k}$ extending the complexification $\mathfrak{sl}(2,\mathbb{C})_{\mathbb{C}} \simeq \mathfrak{sl}_2 \oplus \mathfrak{sl}_2$ of $\mathfrak{sl}(2,\mathbb{C})$.

The affine Lie-algebra symmetry implies that the correlation functions locally define modules of $\widehat{\mathfrak{sl}}_{2,k} \oplus \widehat{\mathfrak{sl}}_{2,k}$, represented by $\mathfrak{sl}_2$-valued differential operators $\mathcal{J}(z)$ and $\bar{\mathcal{J}}(\bar{z})$, with $z$ being a local coordinate on $C$. The operator $\mathcal{J}(z)$ is a first order differential operator in the variables $x$, while $\bar{\mathcal{J}}(\bar{z})$ is the complex conjugate of $\mathcal{J}(z)$. The action of $\mathcal{J}(z)$ and $\bar{\mathcal{J}}(\bar{z})$ on $\mathcal{G}$ are interpreted as the current expectation values.

The Sugawara construction allows one to derive differential equations of the following form

$$(k+2)\frac{\partial}{\partial z_r}\mathcal{G}(\boldsymbol{x},\boldsymbol{z};\boldsymbol{j}) = \mathsf{H}^{\text{KZB}}_{\boldsymbol{x},r}\mathcal{G}(\boldsymbol{x},\boldsymbol{z};\boldsymbol{j}),$$

$$(k+2)\frac{\partial}{\partial \bar{z}_r}\mathcal{G}(\boldsymbol{x},\boldsymbol{z};\boldsymbol{j}) = \bar{\mathsf{H}}^{\text{KZB}}_{\bar{\boldsymbol{x}},r}\mathcal{G}(\boldsymbol{x},\boldsymbol{z};\boldsymbol{j}),$$

(57)

where $\mathsf{H}^{\text{KZB}}_{\boldsymbol{x},r}$ are certain second order differential operators with respect to the variables $\boldsymbol{x}=(x_1,\ldots,x_h)$, and the operators $\bar{\mathsf{H}}^{\text{KZB}}_{\bar{\boldsymbol{x}},r}$ are obtained from $\mathsf{H}^{\text{KZB}}_{\boldsymbol{x},r}$ by complex conjugation.

The differential operators $\mathsf{H}^{\text{KZB}}_{\boldsymbol{x},r}$ reduce to the Hitchin Hamiltonians when $k=-2$ [67], and represent quantisations of the isomonodromic deformation Hamiltonians [67–69]. These relations suggest that the quantum geometric Langlands correspondence is a part of a larger picture discussed in [12] allowing two types of quantisation and deformation parameters.

## 6.4 Holomorphic factorisation

Given that the correlation functions of the $H_3^+$ WZNW model represent solutions to the KZB-equations (57), it is natural to expect that these functions can be represented in a holomorphically factorised form,

$$\mathcal{G}(\boldsymbol{x},\boldsymbol{z};\boldsymbol{j}) = \int_{\mathbb{R}^h} d\mu_{\text{w}}(\boldsymbol{p})\ \mathcal{G}_\gamma(\boldsymbol{x},\boldsymbol{z};\boldsymbol{p};\boldsymbol{j})\mathcal{G}_\gamma(\bar{\boldsymbol{x}},\bar{\boldsymbol{z}};\boldsymbol{p};\boldsymbol{j}).$$

(58)

The functions $\mathcal{G}_\gamma$ in (58) are series solutions of the KZB equations (57) having power-like singular behaviour parameterised by the variables $\boldsymbol{p}$ near the boundary of the moduli space of Riemann surfaces labelled by $\gamma$. The measure $d\mu_{\text{w}}(\boldsymbol{j})$ can be represented in terms of two explicitly known functions $B_{\text{w}}(j)$ and $C_{\text{w}}(j_3,j_2,j_1)$ [66].

We shall illustrate the main idea underlying the definition of the functions $\mathcal{G}_\gamma$ in the case $C=C_{0,4}$, allowing us to replace the variables $\boldsymbol{x}$ and $\boldsymbol{z}$ by the cross-ratios $x$ and $z$, formed out of the variables $x_1,\ldots,x_4$ and $z_1,\ldots,z_4$ appearing in (53), respectively. The KZB-equations (57) may then be represented explicitly in the form

$$(k+2)\frac{\partial}{\partial z}\mathcal{G}_\gamma(x,z;\boldsymbol{j}) = \left(\frac{\mathcal{P}_x}{z} + \frac{\mathcal{Q}_x}{1-z}\right)\mathcal{G}(x,z;\boldsymbol{j}),$$

(59)

with $\mathcal{P}_x$ and $\mathcal{D}_x$ being second order differential operators in $x$ independent of $z$. There exist power series solutions $\mathcal{G}_\gamma(x,z;p;\boldsymbol{j})$ of the form

$$\mathcal{G}(x,z;p;\boldsymbol{j}) = z^{\Delta_{j_p}-\Delta_{j_1}-\Delta_{j_2}}\sum_{k=0}^{\infty}z^k\mathcal{G}_{\gamma,k}(x;p;\boldsymbol{j}),$$

(60)

with $\mathcal{G}_{\gamma,0}(x;p;\boldsymbol{j})$ being eigenfunctions of $\mathcal{P}_x$ with eigenvalue $(k+2)(\Delta_{j_p}-\Delta_{j_1}-\Delta_{j_2})$, where

$$\Delta_j = \frac{j(j+1)}{k+2},\qquad j_p = -\frac{1}{2}+\mathrm{i}p,$$

(61)

and $\mathcal{G}_{\gamma,k}(x;p;\boldsymbol{j})$ for $k>0$ determined recursively in terms of $\mathcal{G}_{\gamma,0}(x)$ by (59). The series are expected to be absolutely convergent for $|z|<|x|<1$.

The papers [65,66] presented evidence that there exist functions $B_{\text{w}}(j)$ and $C_{\text{w}}(j_3,j_2,j_1)$ such that the linear combination $\mathcal{G}(\boldsymbol{z},\boldsymbol{x};\boldsymbol{j})$ defined in (58) admits a real-analytic and single-valued continuation to the total space of the fibration with fibers $\text{Bun}_G\simeq\mathbb{P}^1\setminus\{0,z,1,\infty\}$ over the moduli space $\mathcal{M}_{0,4}$ of four-punctured spheres parameterised by the cross-ratio $z$.

This conjecture can be proven with the help of the relation to Liouville theory [70] where the analogous problem has been completely solved, as will be reviewed later in this paper.

## 6.5 Limits of the $H_3^+$ WZNW model

We will here review some predictions concerning the behaviour of the $H_3^+$ WZNW model in the limits $k \to \infty$ and $k \to -2$ that can be supported by fairly standard arguments from theoretical physics. These limits will admit more explicit descriptions revealing relations of the WZNW model to the harmonic analysis of $SL(2, \mathbb{C})$, to the mathematical theory of holomorphic $SL(2)$-bundles on Riemann surface, and to the analytic Langlands correspondence. Relating these limits to the solutions of mathematical problems which have been previously investigated, will offer valuable insights into the mathematical nature of the $H_3^+$ WZNW model.

### 6.5.1 Mini-superspace limit

The limit $k \to \infty$ with representations labels $j_r$, $r = 1, \ldots, n$ kept finite leads to the suppression of the derivatives of the field $h(z)$ with respect to $z$ and $\bar{z}$. The consequence is a reduction to the quantum mechanical system discussed in [71]. This limit reduces the correlation functions defined in (53) to the finite-dimensional integrals

$$G(\boldsymbol{x}; \boldsymbol{j}) = \int_{H_3^+} dh \prod_{r=1}^{n} \Theta^{j_r}(h; x_k), \tag{62}$$

where $dh$ is the $SL(2, \mathbb{C})$-invariant measure on $H_3^+$.

This establishes a link with the harmonic analysis on the symmetric space $H_3^+$. In order to describe this link more explicitly, one may expand

$$\Theta^j(h; x) = \sum_{2l \in \mathbb{Z}_{\geq 0}} \sum_{m=-l}^{l} \Psi_{lm}^p(h) \Xi_{lm}^p(x), \qquad j = -\frac{1}{2} + \mathrm{i}p, \tag{63}$$

where the functions $\Xi_{lm}^p(x)$, $l \in \frac{1}{2}\mathbb{Z}_{\geq 0}$, $m \in \{-l, -l+1, \ldots, l\}$ generate a basis in $\mathcal{P}_j \simeq L^2(\mathbb{C})$ transforming as the irreducible representation with spin $l$ under the compact subgroup of $SL(2, \mathbb{C})$, and the functions $\Psi_{lm}^p(h)$ represent distributions on $L^2(H_3^+)$, satisfying

$$\int_{H_3^+} dh \left( \Psi_{l'm'}^{p'}(h) \right)^* \Psi_{lm}^p(h) = \delta_{l',l} \delta_{m,m'} \delta(p - p'). $$

The collection of functions $\Psi_{lm}^p(h)$ forms a basis for the space of square-integrable functions $L^2(H_3^+)$. The transformation (63) allows us to express the function $G(\boldsymbol{x}; \boldsymbol{j})$ in terms of similar integrals over products of the functions $\Psi_{lm}^p$.

The products $\Psi_{l_2,m_2}^{p_2}(h)\Psi_{l_1,m_1}^{p_1}(h)$ are normalisable in $L^2(H_3^+)$. Representing such products as integrals over the functions $\Psi_{lm}^p(h)$ leads to expansions which become equivalent to to (58) in the mini-superspace limit.

### 6.5.2 Classical limit

The limit $k \to \infty$ with representations labels $j_r$ diverging such that $j_r/k$ stays finite for $r = 1, \ldots, n$ will be referred to as the classical limit.

$$\lim_{k \to \infty} \frac{1}{k} \log \mathcal{G}(z, x; \boldsymbol{j}) = S(\boldsymbol{x}, \boldsymbol{z}; \boldsymbol{j}), \tag{64}$$

where

$$S(z, x; \boldsymbol{j}) = S_{\mathrm{WZ}}\big[h(.; \boldsymbol{x}, \boldsymbol{z}; \boldsymbol{j})\big], \tag{65}$$

and $h(z) \equiv h(z; \boldsymbol{x}, \boldsymbol{z}; \boldsymbol{j})$ is the unique solution of the following problem: For given families $\mathcal{E}_{\boldsymbol{x}}$ of holomorphic bundles, the function $h$ is the single-valued solution to the equation

$$\bar{\partial}_{\bar{z}}\big[(\partial_z h)h^{-1}\big] = 0 \,, \tag{66}$$

which can be represented in the form

$$h(z) = g(z) \cdot (g(z))^{\dagger} \,, \tag{67}$$

where $g(z)$ is a holomorphic multi-valued section of the stable parabolic bundle $\mathcal{E}_{\boldsymbol{x}}$ having power-like growth at $z = z_r$, $r = 1, \ldots, n$ determined by the parameters $\boldsymbol{j}$.[27] In order for $h_{\mathrm{cl}}(z)$ to be single-valued, it is necessary that the monodromy of $g(z)$ is contained in SU(2). This means that solutions to this problem correspond to the pairs $(\mathcal{E}, \nabla)$ of holomorphic bundles $\mathcal{E}$ with connection $\nabla$ having unitary holonomy.

This is precisely the classical form of the $B_c$ boundary condition we introduced in complex Chern-Simons theory. Indeed, we propose that the WZNW partition function at pure imaginary $k+2$ coincides with the wave-function of the boundary state $|1\rangle$ created by $B_c$ in the polarisation representing states by densities on $\mathrm{Bun}_G$.

Important results about this problem for $C = C_{0,n}$ can be found in [72] and references therein.

### 6.5.3 Critical level limit

The limit $k \to -2$ of the KZB-equations has first been analysed in [73]. It is elementary to verify that an Ansatz of the form

$$\mathcal{G}(\boldsymbol{x}, \boldsymbol{z}) \sim e^{-b^2 S(\boldsymbol{z})} \Psi(\boldsymbol{x}, \boldsymbol{z})\big(1 + \mathcal{O}(b^{-2})\big) \,, \tag{68}$$

yields solutions to (57) if

- the functions $\Psi(\boldsymbol{x}, \boldsymbol{z})$ are eigenfunctions $\Psi$ of the Hitchin Hamiltonians,

$$\mathcal{H}_x^{(r)} \Psi(\boldsymbol{x}, \boldsymbol{z}) = E_r \Psi(\boldsymbol{x}, \boldsymbol{z}), \qquad \overline{\mathcal{H}}_{\bar{x}}^{(r)} \Psi(\boldsymbol{x}, \boldsymbol{z}) = \bar{E}_r \Psi(\boldsymbol{x}, \boldsymbol{z}), \qquad \text{and}$$

- the function $S$ is a generating function of the eigenvalues $E_r$ and $\bar{E}_r$, $r = 1, \ldots, h$,

$$\frac{\partial}{\partial z_r} S(\boldsymbol{z}) = E_r(\boldsymbol{z}), \qquad \frac{\partial}{\partial \bar{z}_r} S(\boldsymbol{z}) = \bar{E}_r(\boldsymbol{z}), \qquad r = 1, \ldots, h\,. \tag{69}$$

This observation suggests that the problem to construct single-valued solutions to the KZB-equations represents a natural deformation of the eigenvalue problem for the Hitchin Hamiltonians with deformation parameter $k$.

### 6.6 Verlinde line operator insertions

We will now introduce a family of modification of the correlation functions of the $H_3^+$-WZNW model labelled by half-integer measured laminations $\Lambda$. For a given lamination $\Lambda$, let $\gamma$ be a boundary component in the moduli space of curves $C$ corresponding to vanishing length of the geodesics representing $\Lambda$. In the neighbourhood of such a boundary one may start with (58), and simply modify it to

$$\mathcal{G}_{\Lambda}(\boldsymbol{x}, \boldsymbol{z}; \boldsymbol{j}) := 2^h \int_{\mathbb{R}^h} d\mu_{\mathrm{W}}(\boldsymbol{p}) \prod_{r=1}^{h} \cosh(4\pi w_r p_r)\, \mathcal{G}_{\gamma}(\boldsymbol{x}, \boldsymbol{z}; \boldsymbol{p}; \boldsymbol{j}) \mathcal{G}_{\gamma}(\bar{\boldsymbol{x}}, \bar{\boldsymbol{z}}; \boldsymbol{p}; \boldsymbol{j}) \,, \tag{70}$$

---

[27]The parameters $\boldsymbol{j}$ are related to the data called parabolic weights in the mathematical literature. See e.g. [72] for a thorough review of relevant background in a related context.

where $w_r$, $r = 1, \dots, h$, are the weights of $\Lambda$. The definition (70) may look somewhat ad hoc at this point. We will explain later in this paper why it is natural, relating the modification in (70) relative to (58) to the Verlinde line operators.

It will in particular turn out that the correlation functions (70) admit a continuation over the moduli space of curves and $\mathrm{Bun}_G$ which is single valued and real-analytic over $\mathrm{Bun}_G$ away from the locus represented by wobbly bundles. This continuation is uniquely determined by the KZB-equations (57). A crucial result of this paper may be concisely formulated as follows:

> *The critical level limit $k \to -2$ of $\mathcal{G}_\Lambda$ takes a form similar to (68), with $\Psi(\boldsymbol{x}, \boldsymbol{z})$ replaced by the eigenfunctions $\Psi_\Lambda(\boldsymbol{x}, \boldsymbol{z})$ of the Hitchin-Hamiltonians associated by the analytic Langlands correspondence to the real oper obtained from the uniformizing oper by grafting along $\Lambda$.*

# 7 A deformation of the separation of variables

This section will review the integral transformation relating correlation functions of the $H_3^+$-WZNW model to Liouville correlation functions. We will see that this transformation can be interpreted as a deformation of the Separation of Variables transformation (12).

## 7.1 Liouville theory

The Liouville field theory can be defined on the classical level using the action

$$S_{\mathrm{Liou}} = \frac{1}{4\pi} \int_{\mathbb{C}} dx \left( |\nabla \phi(x)|^2 + 4\pi \mu e^{2b\phi(x)} \right). \tag{71}$$

For Riemann surfaces of genus zero one may then define the correlation functions by the path integral [74]

$$\left\langle \prod_{k=1}^n V_{\alpha_k}(z_k) \right\rangle_{\mathrm{L}} := \int_{\phi : \mathbb{C} \to \mathbb{R}} \mathcal{D}[\phi] \, e^{-S_{\mathrm{Liou}}[\phi]} \prod_{k=1}^n e^{2\alpha_k \phi(z_k)}. \tag{72}$$

The definition can be extended to surfaces $C$ of genus $g = 1$ [75] and of genus $g \geq 2$ [76].

It has been conjectured in [77] that there exists a range of values for $\alpha_1, \dots, \alpha_n$ for which these correlation functions have a representation of the form,

$$\left\langle \prod_{k=1}^n V_{\alpha_k}(z_k) \right\rangle_{\mathrm{L}} = \int_{\mathbb{S}^h} dC_\mu(\boldsymbol{\alpha}) \, \mathcal{F}_\mu(\boldsymbol{\alpha}; \boldsymbol{z}) \mathcal{F}_\mu(\boldsymbol{\alpha}; \bar{\boldsymbol{z}}), \tag{73}$$

using the following notations:

- $\mathbb{S}_{\mathrm{L}} = \frac{Q}{2} + i\mathbb{R}_+$, $Q = b + b^{-1}$, and $h = 3g - 3 + n$ if $C$ has genus $g$ and $n$ punctures.

- The integration is extended over the space $\mathbb{S}_{\mathrm{L}}^h$, the $h$-fold Cartesian product of $\mathbb{S}_{\mathrm{L}}$, and the integration variable $\boldsymbol{\alpha}$ is an $h$-tuple $\boldsymbol{\alpha} = (\alpha_{n+1}, \dots, \alpha_{n+h})$ of elements $\alpha_r \in \mathbb{S}$.

- $\mu$ is a marking, a pair $\mu = (\Gamma, \Delta)$ consisting of a collection $\Gamma = (\gamma_1, \dots, \gamma_h)$ of non-intersecting simple closed curves defining a pants decomposition of $C$ and a trivalent graph $\Delta$ on $C$ having exactly one vertex $v$ in each of the pairs of pants defined by $\gamma$. Each edge of $\Delta$ is labelled by a number $r \in \{1, \dots, n + h\}$ in such a way that $\alpha_r$ gets assigned to the edge with number $r$.

- For a given pants decomposition of $C$ one can represent the measure $dC_\mu(\boldsymbol{\alpha})$ in terms of two known[28] functions $R(\alpha)$ and $C(\alpha_3, \alpha_2, \alpha_1)$ as

$$dC_\mu(\boldsymbol{\alpha}) = \prod_{r=n+1}^{h+1} \frac{d\alpha_r}{R(\alpha_r)} \prod_{v \in \Delta_0} C(\alpha_{r_{v,3}}, \alpha_{r_{v,2}}, \alpha_{r_{v,1}}), \tag{74}$$

where $\Delta_0$ is the set of vertices of $\Delta$, and $(r_{v,3}, r_{v,2}, r_{v,1})$ is the tuple of numbers of the edges incident to the vertex $v \in \mu_0$.

- The conformal blocks $\mathcal{F}_\mu$ are power series in $h$ gluing parameters for a family of Riemann surfaces defined by gluing pairs of pants. The coefficients of the power series $\mathcal{F}_\mu$ are recursively defined by the representation theory of the Virasoro algebra (see e.g. [15]).

A rigorous proof of the holomorphic factorisation was recently given for the case $g = 0$ in [81].

## 7.2 Degenerate fields in Liouville theory

The fields $V_{\alpha_{n,m}}(z)$ with $\alpha_{n,m} = -nb/2 - m/2b$, $m, n \in \mathbb{Z}_{\geq 0}$, have special properties. Correlation functions involving such fields satisfy partial differential equations first studied by Belavin, Polyakov and Zamolodchikov (BPZ) in the context of minimal models. Such fields are called degenerate fields. The differential equations can be used to construct the fields $V_{\alpha_{r,s}}(z)$ recursively from the two basic examples $V_{-b/2}(w)$ and $V_{-1/2b}(y)$. Of particular interest for us will therefore be correlation functions of the form

$$\left\langle \prod_{k=1}^{n} V_{\alpha_k}(z_k) \prod_{j=1}^{m} V_{-b/2}(w_j) \prod_{i=1}^{l} V_{-1/2b}(y_i) \right\rangle_{\text{L}}. \tag{75}$$

Such correlation functions satisfy a system of partial differential equations of the form

$$\left( \frac{1}{b^2} \frac{\partial^2}{\partial w_j^2} + \mathbf{D}_j^{\text{BPZ}} \right) \left\langle \prod_{k=1}^{n} V_{\alpha_k}(z_k) \prod_{j=1}^{m} V_{-b/2}(w_j) \prod_{i=1}^{l} V_{-1/2b}(y_i) \right\rangle_{\text{L}} = 0,$$

$$\left( b^2 \frac{\partial^2}{\partial y_i^2} + \tilde{\mathbf{D}}_i^{\text{BPZ}} \right) \left\langle \prod_{k=1}^{n} V_{\alpha_k}(z_k) \prod_{j=1}^{m} V_{-b/2}(w_j) \prod_{i=1}^{l} V_{-1/2b}(y_i) \right\rangle_{\text{L}} = 0, \tag{76}$$

where $j = 1, \ldots, m$, $i = 1, \ldots, l$, and $\mathbf{D}_j^{\text{BPZ}}$, $\tilde{\mathbf{D}}_i^{\text{BPZ}}$ are first order differential operators.

One may regard such correlation functions as objects associated to a surface $C = C_{g,n}$ of genus $g$ with $n$ punctures at $z_1, \ldots, z_n$, on which one has marked two further collections of distinguished points $\boldsymbol{y} = (y_1, \ldots, y_l)$ and $\boldsymbol{w} = (w_1, \ldots, w_m)$. Families of such surfaces can be constructed by the gluing construction from a collection of pants $\{S_v; v \in \mu_0\}$, each $S_v$ having two mutually disjoint sets of punctures $\boldsymbol{w}_v = (w_{v,1}, \ldots, w_{v,m_v})$ and $\boldsymbol{y}_v = (y_{v,1}, \ldots, y_{v,l_v})$.

### 7.2.1 Gluing construction

The conformal blocks appearing in Liouville correlation functions can be constructed by the gluing construction. The geometric set-up underlying this construction will be a refinement of the set-up used to define FN-type coordinates in Section 4.4. It considers a decomposition of $C$ into punctured annuli and pairs of pants, decorated with seams decomposing all annuli and pairs of pants into simply-connected pieces.

---

[28] The explicit expressions for the functions $R$ and $C$ have been conjectured in [77, 78]. Strong arguments supporting this conjecture were later given in [79]. A rigorous proof has recently been proposed in [80].

Conformal blocks for $C$ can then be constructed from conformal blocks associated to the building blocks by the gluing construction. One needs to assign a representation of the Virasoro algebra to each cutting curve appearing in this decomposition.

The assignment of representations to cutting curves must take into account the so-called fusion rules. The space of conformal blocks assigned to an annulus with degenerate representation $V_{-jb^{\pm1}}$, $j \in \frac{1}{2}\mathbb{Z}$, assigned to a single puncture is non-trivial only if the representation labels $\alpha_A$, $\alpha'_A$ associated to the two boundaries

$$\alpha' = \alpha - mb^{\pm1}, \qquad m \in \left\{-j, -j+1, \ldots, j-1, j\right\}. \tag{77}$$

Similar selection rules for conformal blocks associated to annuli with any number of degenerate punctures can be obtained by gluing annuli with lower numbers of degenerate punctures.

The factorisation expansions of Liouville correlation functions in the presence of degenerate insertions can then be represented in terms of conformal blocks associated to intermediate representations from the set

$$\hat{\mathbb{S}} := \left\{\alpha + jb + kb^{-1}; \alpha \in \mathbb{S}, j, k \in \tfrac{1}{2}\mathbb{Z}\right\}, \qquad \mathbb{S} = \frac{Q}{2} + i\mathbb{R}.$$

We will consider decorations assigning elements of $\hat{\mathbb{S}}$ to each cutting curve.

### 7.2.2 Factorised form of correlation functions with degenerate fields

Taking into account these differential equations one may then show that the holomorphically factorised representation (73) has a generalization of the following form

$$\left\langle \prod_{k=1}^{n} V_{\alpha_k}(z_k) \prod_{j=1}^{m} V_{-b/2}(w_j) \prod_{i=1}^{l} V_{-1/2b}(y_i) \right\rangle_{\text{L}} = \sum_{\iota} \int_{\mathbb{S}_{\text{L}}^h} dC_{\mu,\iota}(\boldsymbol{\alpha}) \left|\mathcal{F}_{\mu,\iota}(\boldsymbol{\alpha}; \boldsymbol{z}, \boldsymbol{w}, \boldsymbol{y})\right|^2, \tag{78}$$

with conformal blocks now denoted as $\mathcal{F}_{\mu,\iota}(\boldsymbol{\alpha}; \boldsymbol{z}, \boldsymbol{w}, \boldsymbol{y})$, and $\mu$ being a marking $\mu = (\Gamma, \Delta)$ of $C = C_{g,n}$ with numbered edges as before. The functions $\mathcal{F}_{\mu,\iota}(\boldsymbol{\alpha}; \boldsymbol{z}, \boldsymbol{w}, \boldsymbol{y})$ can be characterised as solutions to the system (76) of PDE characterised by having a particular asymptotic behaviour at the nodal degeneration of the complex structure of $C$ described in terms of the gluing parameters. There will then be unique solutions to (76) having leading behaviour in this nodal degeneration which is proportional to the product of the conformal blocks associated to the pants $S_\nu$, $\nu = 1, \ldots, 2g-2+n$, from which $C$ was built. The parameters $\alpha$ determine the labels associated to the three holes of the pants by the same rule as used already in (74). The multi-indices $\iota = (\iota_1, \ldots, \iota_{2g-2+n})$ collect the labels $\iota_\nu$ for elements of a basis for the space of solutions to the equations (76) for each sphere $S_\nu$, $\nu = 1, \ldots, 2g-2+n$.

Remarkably [82], the constraint that the solutions should have a good degeneration limit with $w$'s and $y$'s placed in various ways with respect to the degeneration locus turns out to be sufficiently strong to determine the $z$ dependence of conformal blocks completely, without explicit use of the representation theoretic definition employing a sum over Virasoro descendants in intermediate channels.

## 7.3 Relation between Liouville theory and the $H_3^+$ WZNW model

It has been shown in [70] that the correlation functions of the $H_3^+$ WZNW model on Riemann surfaces with genus $g = 0$ can be represented in terms of certain correlation functions of Liouville theory. Correlation functions of the $H_3^+$ WZNW model can be expressed in terms of an integral transformation of the following form:

$$\left\langle \prod_{k=1}^{n} \Phi^{j_k}(x_k|z_k) \right\rangle_{\text{W}} = \int d^2 y_1 \ldots d^2 y_h \, \mathcal{K}_{\kappa}(\boldsymbol{x}, \boldsymbol{y}; \boldsymbol{z}) \left\langle \prod_{k=1}^{n} V_{\alpha_k}(z_k) \prod_{l=1}^{h} V_{-1/2b}(y_l) \right\rangle_{\text{L}}, \tag{79}$$

assuming the relations $h = n-2$, $b^{-2} = \kappa - 2$, and $2\alpha_k - Q = b(2j_k + 1)$ for $k = 1, \ldots, n$. The relation (79) simplifies a bit in the case where $z_n$ and $x_n$ are sent to infinity. In this case one has $h = n-3$ in (79), and one finds the following formula for the kernel $\mathcal{K}_\kappa$ [38]:

$$\mathcal{K}_{\mathrm{SOV}}^{(n)}(\boldsymbol{x}, \boldsymbol{y}; \boldsymbol{j}, \boldsymbol{z}) = \left| \sum_{r=1}^{n-1} x_r \frac{\prod_{k=1}^{n-3}(z_r - y_k)}{\prod_{s \neq r}^{n-1}(z_r - z_s)} \right|^{2J} \prod_{k<l}^{n-3} |y_k - y_l|^{-k} \prod_{r=1}^{n-1} \left[ \frac{\prod_{s \neq r}^{n-1} |z_r - z_s|^2}{\prod_{k=1}^{n-3} |z_r - y_k|^2} \right]^{\frac{\alpha_r}{b}}, \quad (80)$$

where $J = -j_n + \sum_{r=1}^{n-1} j_r$.

Similar relations can be derived for correlation functions on surfaces $C$ of genus $g > 0$ [40]. One can use relations (79) and their higher genus generalizations to construct the correlation functions of the $H_3^+$ WZNW model from Liouville theory.

If $h = 3g-3+n$ one may solve the system of equations (76) for the derivatives $\frac{\partial}{\partial z_r}$, leading to an equivalent representation of the system of equations (76) in the form

$$\left( \frac{1}{b^2} \frac{\partial}{\partial z_r} - \mathsf{H}_{\boldsymbol{y},r}^{\mathrm{BPZ}} \right) \left\langle \prod_{k=1}^{n} V_{\alpha_k}(z_k) \prod_{l=1}^{3g-3+n} V_{-1/2b}(y_l) \right\rangle_{\mathrm{L}} = 0, \qquad r = 1, \ldots, h, \quad (81)$$

where the second order differential operators $\mathsf{H}_{\boldsymbol{y},r}^{\mathrm{BPZ}}$ are quantisations of the isomonodromic deformation Hamiltonians in Garnier form [12]. The integral transformation (79) intertwines the action of $\mathsf{H}_{\boldsymbol{x},r}^{\mathrm{KZB}}$ with $\mathsf{H}_{\boldsymbol{y},r}^{\mathrm{BPZ}}$ [40, 70]. It enjoys natural relations with the transformations for the nodal surfaces appearing at the boundaries of the Deligne-Mumford compactification [70].

## 7.4  Critical level limit

We are now going to explain how the separation of variables transformation (12) is obtained from the $H_3^+$-Liouville relation (79) in the critical level limit.

Recall that the $b \to \infty$ limit is a classical limit for Liouville theory. Accordingly, a Liouville theory correlation function behaves as

$$\left\langle \prod_{k=1}^{n} V_{\alpha_k}(z_k) \right\rangle_{\mathrm{L}} \sim e^{-b^2 S_{\mathrm{Liou}}[\phi_{\mathrm{cl}}]}, \quad (82)$$

for a classical solution $\phi_{\mathrm{cl}}$ of the Liouville equations of motion, i.e. an uniformizing oper with a specific behaviour at the punctures.

The relation (82) can be made more precise by looking at the BPZ differential equations satisfied by the "light" degenerate fields $V_{-\frac{1}{2b}}$, which reduces to an oper differential equation for a classical $t(z)$ which is the semiclassical limit of $b^{-2} T(z)$, a rescaled stress tensor. Thus the semiclassical limit of a $V_{-\frac{1}{2b}}(z)$ insertion is the single-valued flat section $\chi$ of the uniformizing oper:

$$F^{(n)}(\boldsymbol{y}, \boldsymbol{z}) \sim e^{-b^2 S(\boldsymbol{z})} Z_{1-\mathrm{loop}}^{L,n}(\boldsymbol{z}) \prod_{j=1}^{n-3} \chi(y_j, \bar{y}_j; \boldsymbol{z})\left(1 + \mathcal{O}(b^{-2})\right), \quad (83)$$

where

- The functions $\chi(y, \bar{y})$ satisfy a pair of equations of the form

$$\left( \partial_y^2 + t(y) \right) \chi(y, \bar{y}) = 0, \qquad \left( \bar{\partial}_{\bar{y}}^2 + \bar{t}(\bar{y}) \right) \chi(y, \bar{y}) = 0, \quad (84)$$

with $t(y)$ holomorphic on $C_{0,n}$. We refer to equations (84) as the oper equations.

- The function $S(\boldsymbol{z})$ is defined up to a constant by the equations (69), where $E_r(\boldsymbol{z})$ are defined by the expansion

$$t(y) = \sum_{r=1}^{n-1}\left(\frac{\delta_r}{(y-z_r)^2} + \frac{E_r(\boldsymbol{z})}{y-z_r}\right), \tag{85}$$

of the functions $t(y)$ defined by the oper equations (84).

The $b^2 \to \infty$ limit is not a semi-classical limit in the WZW theory. The relation to Liouville, though, still allows an asymptotic expansion, leading to

$$G^{(n)}(\boldsymbol{x}, \boldsymbol{z}) \sim e^{-b^2 S_{\mathrm{Liou}}[\phi_{\mathrm{cl}}](\boldsymbol{z})}\Psi(\boldsymbol{x}, \boldsymbol{z})\bigl(1 + \mathcal{O}(b^{-2})\bigr), \tag{86}$$

where

- The functions $\Psi(\boldsymbol{x}, \boldsymbol{z})$ are eigenfunctions $\Psi$ of the Gaudin Hamiltonians,

$$\mathcal{H}_x^{(r)}\Psi(\boldsymbol{x}, \boldsymbol{z}) = E_r\Psi(\boldsymbol{x}, \boldsymbol{z}), \qquad \overline{\mathcal{H}}_{\bar{x}}^{(r)}\Psi(\boldsymbol{x}, \boldsymbol{z}) = \bar{E}_r\Psi(\boldsymbol{x}, \boldsymbol{z}).$$

- The function $S$ is a generating function of the eigenvalues $E_r$ and $\bar{E}_r$, $r = 1, \ldots, h$,

$$\frac{\partial}{\partial z_r}S(\boldsymbol{z}) = E_r(\boldsymbol{z}), \qquad \frac{\partial}{\partial \bar{z}_r}S(\boldsymbol{z}) = \bar{E}_r(\boldsymbol{z}), \qquad r = 1, \ldots, n-1. \tag{87}$$

- The wavefunction is of the SoV form

$$\Psi(\boldsymbol{x}, \boldsymbol{z}) = \int d^2y_1 \ldots d^2y_{n-3}\, K_{\mathrm{SOV}}^{(n)}(\boldsymbol{x}, \boldsymbol{y}; \boldsymbol{z})\prod_{j=1}^{n-3}\chi(y_j, \bar{y}_j; \boldsymbol{z}). \tag{88}$$

Relation (79) therefore represents a natural deformation of (12).

In conclusion, the WZW partition function gives a single-valued solution of KZB equations with $b^2 \to \infty$ asymptotics controlled by the uniformizing oper.

## 7.5 Analytic continuation

Discussions of the Liouville field theory often assume real values of the parameter $b$. In order to identify the $H_3^+$ WZNW correlation functions with a wavefunction in $\mathcal{H}_s$, though, we will need to consider imaginary values of $b^{-2} = \mathrm{i}s$. Thanks to (79), we can analyse the analytic continuation relating these two regimes using the Liouville correlation functions. The conformal blocks in the factorised expression make sense for all values of $b^2$. The measure and three-point functions have poles at multiples of $b$ and $b^{-1}$. As we rotate the phase of $b^2$ by $\pi/2$, these poles move away from the real axis to rays of slope $\pi/4$, but remain well separated from the integration contour.

Another possible issue concerns the limits $b \to \infty$ of the relation between the correlation functions of the $H_3^+$ WZNW model and Liouville theory. The interplay of the limit $b \to \infty$ with analytic continuation with respect to other parameters of the correlation functions might be affected by Stokes phenomena. The line $\mathrm{i}s = b^{-2}$ is precisely the phase at which a sub-dominant saddle may start competing with the dominant one, if present. Based on the gauge-theory constructions we discussed, we expect the standard Liouville correlation functions to reproduce a path integral over a steepest descent contour associated to the uniformizing oper. This is a stronger result than just claiming that the uniformizing oper dominates asymptotics for real $b^2$: it ensures that no subdominant saddle would appear at the first Stokes line.

# 8 Verlinde line operators

The deformation of the Separation of Variables transform discussed in Section 7 so far misses an important ingredient of the analytic Langlands correspondence: The creation of generic eigenstates from a cyclic vector, geometrically represented by the grafting of real projective structures. We are here going to introduce the proposed counterparts of the grafting operation away from the critical level, represented by the Verlinde line operators. The following Section 9 will offer evidence for the claim that the insertion of Verlinde line operators reproduces the grafting operation in the critical level limit.

## 8.1 Summary

The main goal in this section is to define analogs of the Verlinde line operators acting on $H_3^+$-correlation functions satisfying the following crucial properties:

a) Verlinde line operators are invariant under deformations of the contour used as their label.

b) Verlinde lines generate representations of the quantised algebra of functions on $\mathcal{M}_{\text{flat}}$, denoted $\mathcal{O}_q(\mathfrak{X}_C)$. It follows in particular that a linear basis can be labeled by laminations.

c) Left- and right actions on correlation functions coincide.

d) The mapping class group of the surface $C$ acts naturally.

This turns out to be somewhat subtle.

The usual definition of the Verlinde line operators reviewed below is based on the holomorphic factorisation of correlation functions. The Verlinde line operators can often be defined as operators acting on spaces of conformal blocks. In our context we will find it useful to consider modifications of the correlation functions in which holomorphic or anti-holomorphic conformal blocks are replaced by the result of an action of a Verlinde line operator. However, it turns out that the subtleties with the holomorphic factorisation of $H_3^+$-correlation functions mentioned above prevent a straightforward definition of Verlinde line operators in the $H_3^+$-WZNW-model.

We will therefore begin by describing the analogous construction in the case of Liouville theory, where a fairly complete treatment is possible. We'll then note that a sub-algebra of the algebra of Verlinde line operators in Liouville theory can be mapped to operators acting on $H_3^+$-correlation functions by means of the map between the correlations functions of these two CFTs discussed above. We will briefly mention a possible definition of Verlinde line operators directly in the $H_3^+$-WZNW-model, leaving a more detailed discussion for another occasion.

Regarding correlation functions as twisted-half densities on $\text{Bun}_G$, we may create a family of descendants by the natural action of Verlinde lines on correlation functions. This action represents a natural quantum deformation of the grafting action on Hitchin eigenfunctions. One thereby gets a representation of $\mathcal{O}_q(\mathfrak{X})$ on a natural space of densities on $\text{Bun}_G$.

## 8.2 Definition of Verlinde line operators in Liouville theory

The definition of Verlinde loop operators goes back to [58] and has been applied to Liouville field theory in [59,60]. The following is a reformulation which appears to be particularly well-suited for us. It depends on a choice of pants decomposition as used in the gluing construction, and will yield a direct relation with the Fenchel-Nielsen type coordinates introduced in Section 4.4. We will later discuss in which sense this construction defines objects that do not depend on the auxiliary choice of a pants decomposition.

The definition of Verlinde line operators can be realised on spaces of conformal blocks obtained by the gluing construction outlined in Section 7.2.1. For the definition of Verlinde line operators it turns out that we may restrict attention to somewhat smaller spaces of conformal blocks described as follows. We will restrict attention to conformal blocks associated to surfaces decomposed into annuli and pairs of pants on which all punctures are located inside of pairs of pants. We shall furthermore consider the cases where the degenerate representations assigned to the punctures $P_r$ of $C$ all of the type $V_{-j_r b}$, $j_r \in \frac{1}{2}\mathbb{Z}$. The data characterising such punctures can then be collected in the divisors $D_C = \sum_{r=1}^d 2j_r P_r$.

Let $\boldsymbol{\alpha}$ be a map assigning an element $\alpha_A$ of $\mathbb{S}$ to each annulus $A$. For fixed $\boldsymbol{\alpha}$ we may consider decorations $\Delta_{\boldsymbol{\alpha}}$ assigning discrete shifts $\hat{\alpha}_A = \alpha_A + kb \in \alpha_A + \frac{b}{2}\mathbb{Z}$ of $\alpha_A$ to each annulus $A$. A discrete infinite-dimensional space of conformal blocks is defined for fixed $\boldsymbol{\alpha}$ by taking the direct sum over all possible decorations $\Delta_{\boldsymbol{\alpha}}$ of the tensor products of conformal blocks spaces assigned to the punctured annuli and pairs of pants,

$$\mathsf{CB}(C, \boldsymbol{\alpha}, D_C) = \bigoplus_{\Delta_{\boldsymbol{\alpha}}} \bigotimes_P \mathsf{CB}(P, \Delta_{P,\boldsymbol{\alpha}}, D_P), \tag{89}$$

with $\Delta_{P,\boldsymbol{\alpha}}$ being the decoration of the boundary components of a given pair of pants $P$ defined by the decoration $\Delta_{\boldsymbol{\alpha}}$. One should note that the spaces $\mathsf{CB}(P, \Delta_{P,\boldsymbol{\alpha}}, D_P)$ associated to punctured pairs of pants are all finite-dimensional.

### 8.2.1 Bubbling, de-bubbling and Verlinde line operators

Two operations are basic for the definition of Verlinde line operators, called bubbling and fusion, intuitively describing "creation" and "annihilation" of pairs of degenerate punctures.

In order to define the bubbling, let us consider two distributions $\boldsymbol{D}$ and $\boldsymbol{D}'$ of $d$ and $d+2$ punctures, respectively, having the property that $\boldsymbol{D}' - \boldsymbol{D} = p_1 + p_2$. The key observation repeatedly used in the following is that the space of conformal blocks $\mathsf{CB}(C', \Delta', \boldsymbol{D}')$ contains a subspace canonically isomorphic to $\mathsf{CB}(C, \Delta, \boldsymbol{D})$, as will now be explained.

Given a surface $C$ with $d$ punctures we may construct a surface $C'$ with $d+2$ punctures by gluing a twice-punctured disc $D$ into the open surface obtained from $C$ by cutting out a disc containing no puncture. Conformal blocks for $C'$ can be constructed by taking conformal blocks for $C$ and $D$, respectively, and applying the gluing construction. In the case of degenerate fields with label $-b/2$ assigned to both $p_1$ and $p_2$ there are only two possibilities for the representation assigned to the boundary of $D$, the representations $V_0$ and $V_{-b}$. Let $\delta_0$ be the decoration assigning $V_0$ to the boundary of $D$.

Fixing an element of the one-dimensional space $\mathsf{CB}(D_{p_1 p_2}, \delta_0, p_1 + p_2)$, we may then use the gluing construction to assign to each element of $\mathsf{CB}(C, \Delta, \boldsymbol{D})$ a unique element of $\mathsf{CB}(C', \Delta', \boldsymbol{D}')$, defining a map in the following referred to as bubbling, and denoted by $\mathfrak{B}$.

Closely related is the map $\mathfrak{D}$ defined from the projection onto the subspace of $\mathsf{CB}(C', \Delta', \boldsymbol{D}')$ which is isomorphic to $\mathsf{CB}(C, \Delta, \boldsymbol{D})$. $\mathfrak{D}$ will be referred to as de-bubbling.

Verlinde line operators may then be defined as the composition of bubbling, parallel transport of one of the degenerate punctures defined by the BPZ-equations, and de-bubbling. A concrete algorithm for constructing the parallel transport will be described in Section 8.2.2 below. This composition defines a deformed version of the trace of the holonomy of an oper connection.

### 8.2.2 Parallel transport of degenerate fields

In order to simplify the exposition we will restrict attention to the cases where $C$ has at least one non-degenerate puncture $p$. We may then apply bubbling to the pair of pants having a puncture of $C$ as one of its boundary components. By cutting $C$ along a contour $\gamma_p$ one may

get a decomposition into two connected parts, one of which is a pair of pants containing $p$ and one of the two punctures created by bubbling. It therefore suffices to define the parallel transport for conformal blocks associated to surfaces with a single degenerate puncture.

To simplify the exposition, we will furthermore restrict to the cases with a single degenerate puncture associated to the representation $V_{-b/2}$ for the moment. The puncture may be assumed to be located in any one of the pairs of pants appearing in the refined pants decomposition described above. Natural bases for the two-dimensional spaces of conformal blocks associated to a pair of pants $P'$ with one puncture can be defined by cutting $P'$ along a curve $\delta$ into a punctured annulus $A'$ glued to one of the three boundaries of a pair of pants $P$ without punctures. Assigning representations to the three boundaries of $P'$ leaves two possible choices for the representation assigned to the cutting curve $\delta$ on $P'$ according to (77).

The BPZ null vector decoupling equations define natural isomorphisms between the spaces of conformal blocks associated to different locations of the degenerate puncture. Motions of the puncture can be represented as compositions of three types of elementary moves.

$T_P^{[ij]}$   (*traversing a pair of pants*) – Traversing a pair of pants $P$ along a path connecting boundary components $i$ and $j$ without crossing any seams.

$T_P^{[i]}$   (*half-monodromy*) – Moving across the next seam along boundary $i$ of pair of pants $P$.

$T_A$   (*traversing an annulus $T_A$*) – Move across the circle in the interior of the annulus $A$.

These moves can be represented on spaces of conformal blocks of the type (89) above by taking advantage of the fact that the representation of the elementary moves on spaces of conformal blocks of the type (89) can be induced from the representation on building blocks associated to sub-surfaces of $C$.

For the sake of clarity let us stress that all these operations express the analytic continuation of conformal blocks which can be defined with the help of the gluing construction when the degenerate field is located in a specific region of $C$ in terms of other conformal blocks of this type, defined by gluing constructions that are suitable for locations of the degenerate field in other regions of $C$ rather than the original one.

### 8.2.3   Realisation of elementary moves

We shall now describe these elementary moves more precisely.

$T_P^{[ij]}$**-move:** Let $P'$ be the pair of pants containing the puncture $p$, and let $\delta$ be a curve on $P'$ such that cutting along $\delta$ yields a punctured annulus $A'$ and a pair of pants $P$. If $\gamma_i$, $i \in \{1, 2, 3\}$, is the boundary of $P'$ which becomes a boundary of $A'$ after cutting, and $\alpha_i \equiv \hat{\alpha}_{\gamma_i}$ is associated to $\gamma_i$, only the two assignments $\alpha_i \mp b/2$ of representation labels to $\delta$ yield non-trivial conformal blocks. By fixing normalisations, one may use the conformal blocks from the gluing construction to get a basis for $\mathsf{CB}(P', \Delta_{P', \boldsymbol{\alpha}}, p)$. There exist three such bases, associated to the boundary components $\gamma_i$, $i = 1, 2, 3$ of $P'$. The BPZ equations define a parallel transport pairwise relating such bases, concretely represented by $2 \times 2$ matrices $\mathsf{T}_P^{[ij]}$ having matrix elements which are functions of the three representation labels assigned to the boundaries of $P'$. By adopting suitable normalisations, one may get matrices $\mathsf{T}_P^{[ij]}$ formed out of ratios of trigonometric functions. Explicit formulae can be found e.g. in [60] or [83].

$L_A$**-move:** The monodromy of the bases introduced above, defined by analytic continuation along a contour isotopic to $\gamma$, is represented diagonally, multiplying the basis element corresponding to $\alpha_i \mp b/2$ by $e^{2\pi i b(Q/2 \pm (\alpha_i - Q/2))}$. It is then natural to introduce the half-monodromy operation, represented by the diagonal matrices $L_A$, defined as

$$L_A = e^{\frac{\pi i}{2}(1+b^2)} \begin{pmatrix} \mathsf{U}_A^{+\frac{1}{2}} & 0 \\ 0 & \mathsf{U}_A^{-\frac{1}{2}} \end{pmatrix}, \qquad \mathsf{U}_A^{\pm \frac{1}{2}} := e^{\pm \pi i b(\alpha_i - Q/2)}. \tag{90}$$

This formula holds for a certain orientation of $\gamma$ which we will not need explicitly.

$K_A$**-move:** The representation of $K_A$ follows from the observation that a basis defined in one half of the annulus $A$ trivially defines a basis in the other half by analytic continuation. One needs to note, however, that the representation label assigned to $A$ must change in this process, from $\hat\alpha_\gamma$ to $\hat\alpha_\gamma \mp b/2$, corresponding to the eigenvalues of $M_A$. We may therefore describe parallel transport traversing $A$ by a matrix

$$\mathsf{K}_A = \begin{pmatrix} 0 & \mathsf{V}_A \\ \mathsf{V}_A^{-1} & 0 \end{pmatrix}, \tag{91}$$

with $\mathsf{V}_A$ being the shift operator mapping a conformal block with intermediate representation label $\alpha_A$ to the one defined by replacing $\alpha_A$ by $\alpha_A - b/2$, with other labels unchanged.

### 8.3 Generalisations, labelling and representation by difference operators

The action of a Verlinde line operator on a conformal block $\mathcal{F}_\mu(\boldsymbol{\alpha})$ can be represented as a linear combination of the conformal blocks $\mathcal{F}_\mu(\boldsymbol{\beta})$ associated to finitely many values $\boldsymbol{\beta} \in \mathbb{C}^h$ of the intermediate dimension parameters.

We will find it convenient to replace the variables $\boldsymbol{\alpha} = (\alpha_1,\ldots,\alpha_h)$ by variables $\boldsymbol{p} = (p_1,\ldots,p_h)$, with $p_r$ related to $\alpha_r$ as $p_r = \mathrm{i}(Q/2 - \alpha_r)$. With a slight abuse of notations we shall identify $\mathcal{F}_\mu(\boldsymbol{p})$ with the function defined from $\mathcal{F}_\mu(\boldsymbol{\alpha})$ by this change of variables.

The representation is particularly simple if the contour defining the Verlinde line operator is one of the curves $\gamma_1,\ldots,\gamma_h$ defining the pants decomposition of $C$ underlying the definition of the conformal blocks under consideration. In this case one finds [59, 60] that the Verlinde line operators are represented simply by the operator of multiplication with the function

$$\xi(e^{2\pi b p_r}), \qquad \text{where} \qquad \xi(U) := U + U^{-1}. \tag{92}$$

The definition of the Verlinde line operators leaves a freedom which amounts to multiplying the function $\xi$ by a function of $b$. Formula (92) defines a convention fixing this freedom.[29]

#### 8.3.1 Generalisations

It is not hard to generalise the definition of Verlinde line operators by using the degenerate representations $V_{-bj}$, $j \in \frac{1}{2}\mathbb{Z}_+$, instead of $V_{-b/2}$. Appearance of the vacuum representation in the fusion of two representations $V_{-bj}$ allows us to generalise bubbling and de-bubbling. The parallel transport of an insertion of $V_{-bj}$ is described by operator-valued $(2j+1) \times (2j+1)$-matrices. Combined with bubbling and de-bubbling one gets a quantum trace of the parallel transport defined by the BPZ-equations. Instead of (92) one finds that the Verlinde line operators $W_j$ associated to $\gamma = \gamma_r$ are represented as multiplication by

$$\zeta_j(e^{2\pi b p_r}), \qquad \text{where} \qquad \zeta_j(U) = \sum_{m=-j}^{j} U^{2m}. \tag{93}$$

As another possibility one may consider parallel transport along contours $2j\gamma$ wrapping $2j$ times a simple closed curve $\gamma$. If $\gamma = \gamma_r$ the corresponding operator $\mathcal{L}_{2j\gamma}^\mu$ is represented by multiplication with

$$\xi_j(e^{2\pi b p_r}), \qquad \text{where} \qquad \xi_j(U) = U^{2j} + U^{-2j}. \tag{94}$$

A third possibility is to simply consider the product of $2j$ consecutive applications of the same Verlinde line operator, in the case $\gamma = \gamma_r$ represented by multiplication with $[\chi(U)]^{2j}$.

---

[29]Our convention differs from the one used in [60].

### 8.3.2 Labelling

In all three cases one finds Laurent polynomials in $U = e^{2\pi b p_r}$. The three possibilities mentioned above correspond to three different bases in the space of such Laurent polynomials, each having elements labelled by the half-integer $j = 0, 1/2, 1, \ldots$

We will later see all algebraic relations among Verlinde line operators only depend on the homotopy classes of the loops labelling them. By picking a particular basis, one may therefore use half-integer measured laminations $\mathcal{ML}_C(\frac{1}{2}\mathbb{Z})$ as labels for the Verlinde line operators $\mathcal{L}_\Lambda^\mu$. To a lamination $\Lambda$ represented by the collection of simple closed curves $\lambda_1, \ldots, \lambda_m$ with $\lambda_r$ non-homotopic to $\lambda_s$ for $r \neq s$, and weights $j_r \in \frac{1}{2}\mathbb{Z}^{>0}$ associated to $\lambda_r$ for $r = 1, \ldots, m$ we may associate the product $\prod_r (\mathcal{L}_{\lambda_r}^\mu)^{2j_r}$, for example.

Notice that these three three labelling options lead to the same action of the mapping class group on $\Lambda$. Also, the choice among the three does not appear to affect the $b^2 \to 0$ asymptotic analysis we use to relate decorated partition functions to real opers. In the context of this work, we see no compelling reason to prefer one labelling choice over the others.

### 8.3.3 Representation by difference operators

The explicit formulae for the Verlinde line operators are known in some special cases [59, 60]. As an example we shall consider the case where one of the curves $\gamma_r$ defining the pants decomposition $\mu$ is surrounded by a sphere $C_{0,4}^r$ with four holes or punctures representing a connected component of $C \setminus \bigcup_{s \neq r} \gamma_s$, and $\Lambda = \delta_r$ is a simple closed curve in $C_{0,4}^r$ intersecting $\gamma_r$ twice. The Verlinde loop operators $\mathcal{L}_{\delta_r}^\mu$ will then have a representation of the form $(\mathcal{L}_{\delta_r}^\mu \mathcal{F}_\mu)(\boldsymbol{p}) = \mathcal{D}_r^\mu \mathcal{F}_\mu(\boldsymbol{p})$, where $\mathcal{D}_r^\mu$ is a finite difference operator of the form

$$\mathcal{D}_r^\mu := \mathsf{V}_r \cdot d_+(\mathsf{U}_r) \cdot \mathsf{V}_r + d_0(\mathsf{U}_r) + \mathsf{V}_r^{-1} \cdot d_-(\mathsf{U}_r) \cdot \mathsf{V}_r^{-1}, \tag{95}$$

where $\mathsf{U}_r = e^{2\pi b p_r}$ and $\mathsf{V}_r$ is the shift operator defined as $\mathsf{V}_r \mathcal{F}_\mu(\boldsymbol{p}) = \mathcal{F}_\mu(\boldsymbol{p} + \mathrm{i}\frac{b}{2}e_r)$, with $e_r \in \mathbb{C}^h$ being the vector with components $(e_r)_s = \delta_{rs}$. The coefficients $d_\pm$ in (95) depend on the chosen normalisation of the conformal blocks. The normalisation adopted in [60], for example, leads to

$$d_\pm(U) = -\frac{\prod_{s=\pm}(1 + UM_1^s M_2^{\mp s})(1 + UM_3^s M_4^{\mp s})}{U^2(U - U^{-1})(q^{\pm 1}U - q^{\mp 1}U^{-1})}, \tag{96}$$

where $M_i$, $i = 1, \ldots, 4$ are related to the representation labels assigned to the boundary components of $C_{0,4}^r$ as $M_i = e^{-\pi \mathrm{i} b(2\alpha_i - Q)}$, assuming that $\delta_r$ separates the boundary components of $C_{0,4}^r$ with labels 2 and 3 from the other two boundary components.

General Verlinde line operators $\mathcal{L}_\Lambda^\mu$ can be represented as $(\mathcal{L}_\Lambda^\mu \mathcal{F}_\mu)(\boldsymbol{p}) = \mathcal{D}_\Lambda^\mu \mathcal{F}_\mu(\boldsymbol{p})$, with finite difference operators $\mathcal{D}_\Lambda^\mu$ of the form

$$\mathcal{D}_\Lambda^\mu = \sum_{\ell \in \mathbb{Z}^h} d_{\Lambda,\ell}^\mu(\boldsymbol{U}) \mathsf{V}^\ell, \tag{97}$$

by introducing shift operators $\mathsf{V}^\ell$, $\ell \in \mathbb{Z}^h$ which act on the conformal blocks as

$$\mathsf{V}^\ell \mathcal{F}_\mu(\boldsymbol{p}) = \mathcal{F}_\mu(\boldsymbol{p} + \mathrm{i}\frac{b}{2}\ell). \tag{98}$$

We have $d_l(\mathsf{P}) \neq 0$ for finitely many $\ell \in \mathbb{Z}^h$ only.

### 8.4 The algebra generated by the Verlinde line operators

It turns out that the algebra of Verlinde loop operators is isomorphic to a non-commutative deformation $\mathcal{O}_q(\mathfrak{X}_C)$ of the algebra of functions on the character variety that is related to the skein algebra on the one hand, and to the algebra of quantised geodesic length functions in quantum Teichmüller theory on the other hand. This subsection offers a brief review of the relevant results and conjectures.

#### 8.4.1 Skein quantisation of the character variety

The ring of regular functions of the character variety $\mathcal{O}(\mathfrak{X}_C)$ is generated by trace functions. Quantisation of the canonical symplectic structure of $\mathcal{O}(\mathfrak{X}_C)$ is known [55–57] to yield a non-commutative algebra $\mathcal{O}_q(\mathfrak{X}_C)$ also known as the skein algebra $\text{Sk}_q(C)$. This algebra has generators associated to simple closed curves $\gamma$ on a surface $C$. The relations of $\text{Sk}_q(C)$ can be described in terms of relations between the curves labelling the generators which follow from the simple local relations pictorially represented as:

$$
\underset{L_1 \qquad L_2}{\bigotimes} = q^{\frac{1}{2}} \;\bigg)\bigg( + q^{-\frac{1}{2}} \;\asymp \quad . \tag{99}
$$

The left hand side of (99) represents the product $L_1 L_2$ of two generators associated to curves $\gamma_1$ and $\gamma_2$ crossing each other inside of a disc $D \subset C$, as indicated. From the union of $\gamma_1$ and $\gamma_2$ one can define two new curves by a local surgery replacing the region inside of the disc depicted on the left by one of the regions depicted on the right side of (99). Relation (99) expresses the product $L_1 L_2$ as a linear combination of the generators associated to the curves obtained by the surgeries which can be defined in this way.

It is clear that repeated use of the skein relations allows one to express any product of generators in terms of the generators associated to non-intersecting curves. By picking a basis associated to non-intersecting curves one may construct a linear basis for the algebra $\mathcal{O}_q(\mathfrak{X}_C)$.

#### 8.4.2 Example $C = C_{0,4}$

In the case of the four-punctured sphere $C = C_{0,4}$ one may consider a set of generators $\mathcal{L}_i$ associated to curves $\gamma_i$, $i \in \{1, \ldots, 4\}$ encircling the four punctures, supplemented by generators $\mathcal{L}_{ij}$ associated to the curves $\gamma_{ij}$, $i \neq j$, $i, j \in \{1, 2, 3\}$, separating the pair of punctures labelled by $i$ and $j$ from the two other punctures.

The algebra $\mathcal{O}_q(\mathfrak{X}_C)$ can in this case be explicitly characterised by two types of relations [84]: One has quadratic relations of the form

$$
e^{\pi i b^2} \mathcal{L}_{12} \mathcal{L}_{23} - e^{-\pi i b^2} \mathcal{L}_{23} \mathcal{L}_{12} = (e^{2\pi i b^2} - e^{-2\pi i b^2}) \mathcal{L}_{31} + (e^{\pi i b^2} - e^{-\pi i b^2})(\mathcal{L}_1 \mathcal{L}_3 + \mathcal{L}_2 \mathcal{L}_4), \tag{100a}
$$

together with two similar relations for $\mathcal{L}_{12} \mathcal{L}_{31}$ and $\mathcal{L}_{31} \mathcal{L}_{23}$, respectively, and a cubic relation

$$
\begin{aligned}
e^{\pi i b^2} \mathcal{L}_{12} \mathcal{L}_{23} \mathcal{L}_{31} + \left(2 \cos \pi b^2\right)^2 =\; & e^{2\pi i b^2} \mathcal{L}_{12}^2 + e^{-2\pi i b^2} \mathcal{L}_{23}^2 + e^{2\pi i b^2} \mathcal{L}_{31}^2 \\
& + e^{\pi i b^2} \mathcal{L}_{12}(\mathcal{L}_3 \mathcal{L}_4 + \mathcal{L}_1 \mathcal{L}_2) + e^{-\pi i b^2} \mathcal{L}_{23}(\mathcal{L}_2 \mathcal{L}_3 + \mathcal{L}_1 \mathcal{L}_4) \\
& + e^{\pi i b^2} \mathcal{L}_{31}(\mathcal{L}_1 \mathcal{L}_3 + \mathcal{L}_2 \mathcal{L}_4) + \mathcal{L}_1^2 + \mathcal{L}_2^2 + \mathcal{L}_3^2 + \mathcal{L}_4^2 + \mathcal{L}_1 \mathcal{L}_2 \mathcal{L}_3 \mathcal{L}_4 .
\end{aligned} \tag{100b}
$$

These relations completely characterise the non-commutative algebra deforming the ring of regular functions $\mathcal{O}(\mathfrak{X}_C)$ on the character variety $\mathfrak{X}_C$ for $C = C_{0,4}$ [84]. One can check explicitly that the Verlinde line operators on Liouville conformal blocks define a representation of this algebra [48, Appendix A].

### 8.4.3 Relation between Verlinde line operator algebra and skein algebra

We claim that the algebra of Verlinde line operators is isomorphic to the skein algebra $\mathrm{Sk}_q(C)$ for general Riemann surfaces $C$.

A proof of this claim can be based on the relation between Liouville conformal blocks and quantum Teichmüller theory established in [48] and references therein. This relation implies a direct relation between the Verlinde line operators and the geodesic length operators of the quantum Teichmüller theory. It is known [85, 86] that the geodesic length operators in quantum Teichmüller theory generate a representation of the skein algebra $\mathrm{Sk}_q(C)$.

Another approach has been proposed in [87]. One may argue that the braid group representation generated by Virasoro degenerate representations is conjugate to the braid group representation from the quantum group $\mathcal{U}_q(\mathfrak{sl}_2)$. This relates the algebra of Verlinde line operators to the Witten-Reshetikhin-Turaev representation of the Kauffman bracket skein algebra.

### 8.4.4 Relation to Fenchel-Nielsen type coordinates

Recall that the Fenchel-Nielsen type coordinates for $\mathcal{O}(\mathfrak{X}_C)$ introduced in Section 4.4 are Darboux coordinates for the canonical symplectic structure on $\mathcal{O}(\mathfrak{X}_C)$, as expressed in (25). One should note that the description of the parallel transport of degenerate fields in the definition of the Verlinde line operators exhibits a precise correspondence with the set-up used to define the FN-type coordinates. This correspondence suggests to view the parallel transport of degenerate fields on spaces of conformal blocks as a quantised version of the holonomy of opers.

One may, in particular, compare the definition of the coordinates $(\kappa_A, \lambda_A)$ using (24) with the building blocks of the Verlinde line operators introduced in (90) and (91). This comparison suggests the identifications

$$\mathsf{U}_A = e^{\frac{1}{2}\hat{\lambda}_A}, \qquad \mathsf{V}_A = e^{\frac{1}{2}\hat{\kappa}_A}, \qquad \hat{\kappa}_A = 4\pi i\, b^2 \frac{\partial}{\partial \lambda_A}, \tag{101}$$

identifying $(\hat{\lambda}_A, \hat{\kappa}_A)$ as quantised analogs of the FN-type coordinates $(\lambda_A, \kappa_A)$.

## 8.5 Changes of pants decomposition and mapping class group

The mapping class group $\mathrm{MCG}(C)$ can be represented by diffeomorphisms mapping the underlying smooth surface $S$ to itself. An element $m \in \mathrm{MCG}(C)$ will map a curve $\gamma \subset S$ to $m\gamma \subset S$, and a marking $\mu$ to a marking $m(\mu)$. The Verlinde loop operators are, on the one hand, by definition invariant under the mapping class group action in the sense that

$$\mathcal{D}_{m(\mu)}(m(\Lambda)) = \mathcal{D}_\mu(\Lambda). \tag{102}$$

Indeed, all of the operations appearing in the definition of the Verlinde loop operators only involve the local relations between $\Lambda$ and $\mu$.

The representation of the modular groupoid on the spaces of conformal blocks [48] intertwines the different representations of the Verlinde loop operators as

$$\mathsf{F}_{\mu',\mu} \cdot \mathcal{D}_\mu(\Lambda) = \mathcal{D}_{\mu'}(\Lambda) \cdot \mathsf{F}_{\mu',\mu}. \tag{103}$$

It follows that one can map any given $\mathcal{D}_\mu^\Lambda$ to diagonal form.

The representation of the modular groupoid on the spaces of conformal blocks induces a representation of the mapping class group by taking advantage of the fact that there is a canonical isomorphism $\mathrm{CB}^\mu(C) \simeq \mathrm{CB}^{m(\mu)}(C)$, allowing one to define $\mathsf{M}_\mu(m) : \mathrm{CB}^\mu(C) \to \mathrm{CB}^\mu(C)$ as $\mathsf{M}_\mu(m) = \mathsf{F}_{m(\mu),\mu}$. It then follows from (103) and (102) that

$$\mathsf{M}_\mu(m) \cdot \mathcal{D}_\mu(\Lambda) \cdot \mathsf{M}_\mu(m)^{-1} = \mathcal{D}_{m(\mu)}(\Lambda) = \mathcal{D}_\mu(m^{-1}(\Lambda)), \tag{104}$$

expressing the equivariance of the Verlinde loop operators under the mapping class group.

## 8.6 Representation on Liouville correlation functions

The definition of Verlinde loop operators can be extended to the level of correlation functions. One may simply act on either the holomorphic or anti-holomorphic functions appearing in the holomorphically factorised representation with Verlinde loop operators. In the case $\Lambda = \gamma_r$, for example, one finds that the holomorphically factorised representation (73) of the Liouville correlation functions gets modified to

$$\left\langle \mathcal{L}_\Lambda \prod_{k=1}^n V_{\alpha_k}(z_k) \right\rangle_{\mathrm{L}} = 2 \int_{\mathbb{S}^h} dC_\mu(\boldsymbol{\alpha}) \cos(\pi b(2\alpha_r - Q)) \mathcal{F}_\mu(\boldsymbol{\alpha}; \boldsymbol{z}) \mathcal{F}_\mu(\boldsymbol{\alpha}; \bar{\boldsymbol{z}}). \tag{105}$$

More generally one finds expression of the form

$$\left\langle \mathcal{L}_\Lambda \prod_{k=1}^n V_{\alpha_k}(z_k) \right\rangle_{\mathrm{L}} = \int_{\mathbb{S}^h} dC_\mu(\boldsymbol{\alpha}) \left( \mathcal{D}_\Lambda^\mu \mathcal{F}_\mu(\boldsymbol{\alpha}; \boldsymbol{z}) \right) \mathcal{F}_\mu(\boldsymbol{\alpha}; \bar{\boldsymbol{z}}), \tag{106}$$

where $\mathcal{D}_\Lambda^\mu$ are the difference operators representing the Verlinde line operators introduced in Section 8.3. The expressions of the form (106) define the Verlinde line operator expectation values in the region around the boundary component of the Deligne-Mumford compactification associated to the marking $\mu$ within which there exist convergent power series representations for the conformal block functions $\mathcal{F}_\mu(\boldsymbol{\alpha}; \boldsymbol{z})$.

The representations of the form (106) admit analytic continuation over all of Teichmüller space. This can be used to extend the definition of correlation functions modified by the insertion of Verlinde line operators to all of Teichmüller space. The resulting objects satisfy a generalised version of modular and crossing invariance in the sense of being independent of the choice of the marking $\mu$. While holomorphically factorised representations of a given Verlinde line-operator expectation value certainly depend on the choice of a marking $\mu$, one may relate the representations associated to different pants decompositions with the help of (103).

An important feature, listed as point c) in Section 8.1 above, is the coincidence of the action of Verlinde line operators on correlation functions defined by the representations on holomorphic- and anti-holomorphic conformal blocks, respectively. This is manifest in representations where the representation of Verlinde line operators on conformal blocks is diagonal, as e.g. in (105).

## 8.7 Weak degenerate fields in the presence of strong Verlinde loop operators

The representation of correlation function in the $H_3^+$ WZNW model in terms of Liouville correlation functions uses correlation functions with insertion of the degenerate fields $V_{-1/2b}(y)$. When trying to define Verlinde line operator expectation values with the help of this correspondence, it will be important to understand if the correlation functions depend on the choice of contour representing a lamination $\Lambda$ relative to the positions $y$ of the degenerate fields $V_{-1/2b}(y)$. We will see that this dependence can be non-trivial for odd laminations, complicating the definition of the corresponding Verlinde line operator expectation value in the $H_3^+$ WZNW model in terms of Liouville correlation functions somewhat.

In order to understand the origin of this subtlety, one may consider an annulus $A$ with insertion of a Verlinde line operator of charge $j$ supported on the non-contractible cycle $\gamma$ of $A$, and degenerate field $V_{-1/2b}(y)$ inserted. Assigning representation label $\alpha$ to one of the boundaries, here denoted $\gamma_+$ one needs to have representation labels $\alpha - b^{-1}m$, $m = -j, \ldots, j$

at the other boundary $\gamma_-$ according to (77). If $\gamma$ is homotopic to $\gamma_+$, one finds that $L_{(\gamma,j)}$ is represented by multiplication with

$$\lambda_j(\alpha) := 2\cos(2\pi j b(2\alpha - Q)).$$

Otherwise one finds the factor $\lambda_j(\alpha - b^{-1}m)$ instead. We have $\lambda_j(\alpha - b^{-1}m) = \lambda_j(\alpha)$ if $j = k/2$ with $k$ even, and $\lambda_j(\alpha - b^{-1}m) = -\lambda_j(\alpha)$ if $j = k/2$ with $k$ odd.

It follows that Liouville correlation function with degenerate fields $V_{-1/2b}(y)$ and odd Verlinde line operators has a sign discontinuity when a degenerate field passes across an odd line. This discontinuity creates issues in the definition of the integral transforms relating Liouville correlation functions with Verlinde line operator insertions to solutions of the KZB equations. The definition of the integrand would require additional choices in this case, rendering the verification of some of the properties listed in Section 8.1 difficult.

One may note, on the other hand, that this subtlety does not affect the cases of even Verlinde line operators. For this class of Verlinde line operators one may conveniently use the relation to Liouville theory in order to define Verlinde line operators in the $H_3^+$-WZNW-model, and to study their properties.

## 8.8 Verlinde lines in Liouville vs WZNW models

According to the discussion above, it is certainly possible to define the Verlinde line operators with even weights through the relation with Liouville theory. The integrand is independent of the choice of representatives of the contour defining the Verlinde line operators on the surface $C_{g,n}$ obtained by "filling the degenerate punctures".

One may note, however, that this does not yield enough Verlinde line operators to account for the expected spectrum of the quantised Hitchin Hamiltonians in the critical level limit. There do exist eigen-functions associated to odd-numbers of grafting lines, which one might expect to get from the limits of Verlinde line operators with half-integer charges.

In order to account for this apparent mismatch one may recall from Sections 5 and 6.6 that it should be possible to define Verlinde line operators directly in the $H_3^+$ WZNW model, taking advantage of the holomorphic factorisation of its correlation functions. The holomorphic building blocks appearing in this factorisation should allow one to define modifications associated to insertions of degenerate representations with half-integer weights. On the level of differential equations one may expect that the strategy to define Verlinde line operators by fusing and parallel-transporting degenerate representations has a close analog in the $H_3^+$ WZNW model. More subtle appears to be mainly the interpretation of the results in terms of summing over intermediate representations induced from principal series representations of $SL(2, \mathbb{C})$.

One may, however, expect that the outcome can be described in terms of finite difference operators in a way that generalises the action of Verlinde line operators with integer charges defined from Liouville theory in a natural way. From the representation (70) it is clear, in particular, that the insertions of holomorphic and anti-holomorphic Verlinde lines in the $H_3^+$ WZNW partition function coincide for Verlinde line operators with both integer and half-integer charges.

# 9 Grafting from Verlinde line insertions

We are now going to explain how the insertion of Verlinde operators gets related to the grafting operation in the limit $b \to \infty$. The argument will in particular establish the precise link between the lamination data determining a basis for the algebra of Verlinde line operators, and the grafting data determining real projective structures.

## 9.1 The main claim

We consider the limit $k \to -2$ of the relation (79), now represented in the form

$$G_\Lambda^{(n)}(x, z) = \int d^2 y_1 \dots d^2 y_{n-3} \, \mathcal{K}_{\text{SOV}}^{(n)}(x, y; z) F_\Lambda^{(n)}(y, z), \tag{107}$$

where $F_\Lambda^{(n)}$ and $G_\Lambda^{(n)}$ are correlation functions in Liouville theory and the $H_3^+$-WZNW model, respectively, modified by insertions of Verlinde line operators labelled by a lamination $\Lambda$. The field labels $\boldsymbol{\alpha} = (\alpha_1, \dots, \alpha_n)$ and $\boldsymbol{j} = (j_1, \dots, j_n)$ are fixed and suppressed in the notations.

We are going to argue that (83) and (86) get modified to

$$F_\Lambda^{(n)}(y, z) \sim e^{-b^2 S_\Lambda(z)} Z_{1-\text{loop}}^{L(n)}(z) \prod_{j=1}^{h} \chi_\Lambda(y_j, \bar{y}_j; z)\big(1 + \mathcal{O}(b^{-2})\big), \tag{108}$$

$$G_\Lambda^{(n)}(x, z) \sim e^{-b^2 S_\Lambda(z)} Z_{1-\text{loop}}^{W(n)}(z) \Psi_\Lambda(x, z)\big(1 + \mathcal{O}(b^{-2})\big), \tag{109}$$

where

- The functions $\chi_\Lambda(y, \bar{y}) \equiv \chi_\Lambda(y, \bar{y}; z)$ are related to the metric $ds^2 = e^{2\varphi_\Lambda(y, \bar{y})} dy d\bar{y}$ defined by grafting the uniformising metric along $\Lambda$ as

$$ds^2 = \frac{dy d\bar{y}}{(\chi_\Lambda(y, \bar{y}))^2}.$$

  This implies in particular that the functions $\chi_\Lambda(y, \bar{y})$ satisfy a pair of equations of the form

$$\big(\partial_y^2 + t_\Lambda(y)\big)\chi_\Lambda(y, \bar{y}) = 0, \qquad \big(\bar{\partial}_{\bar{y}}^2 + \bar{t}_\Lambda(\bar{y})\big)\chi_\Lambda(y, \bar{y}) = 0, \tag{110}$$

  with $t_\Lambda(y) \equiv t_\Lambda(y; z)$ holomorphic on $C$.

- The function $S_\Lambda(z)$ is defined up to a constant by the equations

$$\frac{\partial}{\partial z_r} S_\Lambda(z) = E_r^\Lambda(z), \qquad r = 1, \dots, h, \tag{111}$$

  where $E_r^\Lambda(z)$ are defined by the expansion of $t_\Lambda - t_0$ into a basis for $H^0(C, K^2)$. In the case of $g = 0$, for example, it can be explicitly represented as

$$t_\Lambda(y; z) = \frac{\delta_n - \sum_{r=1}^{n-1} \delta_r}{y(y-1)} + \sum_{r=1}^{n-1} \frac{\delta_r}{(y-z_r)^2} + \sum_{r=3}^{n-1} \frac{z_r(z_r - 1)}{y(y-1)} \frac{E_r^\Lambda(z)}{y - z_r}. \tag{112}$$

  The dependence of $E_r^\Lambda(z)$ is real-analytic rather than holomorphic.

- The eigenfunctions $\Psi_\Lambda$ of the Gaudin Hamiltonians,

$$\mathcal{H}_x^{(r)} \Psi_\Lambda(x, z) = E_r^\Lambda(z) \Psi_\Lambda(x, z), \qquad \overline{\mathcal{H}}_{\bar{x}}^{(r)} \Psi_\Lambda(x, z) = \bar{E}_r^\Lambda(z) \Psi_\Lambda(x, z),$$

  are finally defined as

$$\Psi_\Lambda(x, z) = \int d^2 y_1 \dots d^2 y_h \, \mathcal{K}_{\text{SOV}}^{(n)}(x, y; z) \prod_{j=1}^{h} \chi_\Lambda(y_j; z). \tag{113}$$

- The explicit form of $Z_{1-\text{loop}}^{L(n)}(z)$ and $Z_{1-\text{loop}}^{W(n)}(z)$ won't be needed in the following.

The functions characterising the leading order behaviour of the correlation functions of the $H_3^+$-WZNW model are thereby expressed in terms of the real projective structures obtained by grafting the uniformising metric.

## 9.2 Outline of the proof

To this aim we will start by considering laminations $\Lambda = \Lambda(\gamma, \boldsymbol{m})$ specified by a cut system $\gamma$ with multiplicities $\boldsymbol{m}$. The cut system $\gamma$ defines a component $\partial_\gamma \mathcal{M}_{g,n}$ of the Deligne-Mumford moduli space $\mathcal{M}_{g,n}$. The systems of partial differential equations satisfied by Liouville and $H_3^+$-WZNW correlation functions define real analytic continuations from a neighbourhood of the boundary component $\partial_\gamma \mathcal{M}_{g,n}$ to the Teichmüller space $\mathcal{T}_{g,n}$. We will therefore consider values of the arguments $\boldsymbol{z}$ of the correlation functions associated to a sufficiently small neighbourhood of $\partial_\gamma \mathcal{M}_{g,n}$. Considering the extension to $\mathcal{T}_{g,n}$ one might expect phenomena of Stokes type complicating the relation between Verlinde line operator labels and grafting labels. As mentioned above, we suspect that such phenomena are absent in the cases of interest for us.

We may first observe that the discussion in Section 7.4 reduces the claim to a result about the semiclassical limit of certain Liouville correlation functions. This result generalises and refines previous results from [77, 88, 89].

As example we will consider the case $g = 0$, $n = 4$, and the cut system defined by the curve $\gamma$ separating the punctures at 0 and $z$ from 1 and $\infty$, referring to the description used in Section 4.6.1. The corresponding boundary component $\partial_\gamma \mathcal{M}_{0,4}$ corresponds to the limit $z \to 0$. As lamination $\Lambda$ we shall consider $\Lambda = j\gamma$. This example is sufficiently typical.

Near the boundary $\partial_\gamma \mathcal{M}_{0,4}$ one may use the holomorphically factorised representation (105), which in the case at hand simplifies to the form

$$F_\Lambda^{(4)}(z) = \int_{\mathbb{R}} dp \left| \mathcal{F}_\gamma\left(\tfrac{Q}{2} + \mathrm{i}p, z\right) \right|^2 \cosh(4\pi b j p). \tag{114}$$

We have absorbed the three-point functions into the normalisation for the conformal blocks $\mathcal{F}_\gamma$.

### 9.2.1 Classical limit of conformal blocks

The first step is to verify that the conformal blocks in (105) behave in the limit $b \to \infty$ as

$$\log \mathcal{F}_\gamma(\alpha; z) = -b^2 \mathcal{Y}_\gamma(\lambda, z) + \mathcal{O}(b^0), \tag{115}$$

with $\alpha - \frac{Q}{2} = \mathrm{i}b\frac{\lambda}{4\pi}$, and $\mathcal{Y}_\gamma(\lambda, z)$ being the generating function for the sub-manifold of opers in the character variety defined in Section 4.5.

The verification of (115) can be decomposed into three steps.

i) *Exponentiation:* The following limit exists:

$$\mathcal{Y}_\gamma(\lambda, z) := -\lim_{b \to \infty} \frac{1}{b^2} \log \mathcal{F}_\gamma(\alpha; z), \qquad \text{with} \qquad \lambda := \frac{4\pi}{\mathrm{i}b}\left(\alpha - \frac{Q}{2}\right), \tag{116}$$

kept finite for $b \to \infty$.

ii) *Relation with accessory parameters:* Let the accessory parameter $E(\lambda, z)$ be defined as the solution to the inverse monodromy problem to find a function $t(y)$ of the form

$$t(y) = \frac{\delta_1}{y^2} + \frac{\delta_2}{(z-y)^2} + \frac{\delta_3}{(1-y)^2} + \frac{\delta_n - \sum_{r=1}^3 \delta_r}{y(y-1)} + \frac{z(z-1)}{y(y-1)} \frac{E(\lambda, z)}{y-z}, \tag{117}$$

such that the monodromy of $\partial_y^2 + t(y)$ along a contour separating 0 and $z$ from 1 and $\infty$ has trace equal to $2\cosh(\lambda/2)$. The classical conformal blocks $\mathcal{Y}_\gamma(\boldsymbol{\lambda}, \boldsymbol{z})$ are then related to this accessory parameter as follows:

$$\frac{\partial}{\partial z} \mathcal{Y}_\gamma(\lambda, z) = E(\lambda, z). \tag{118}$$

iii) *Relation with FN-type coordinates:*

$$\frac{\partial}{\partial \lambda} \mathcal{Y}_\gamma(\lambda, z) = \frac{1}{4\pi \mathrm{i}} \kappa(\lambda, z).$$
(119)

Parts i) and ii) of the conjectures are due Al.B. Zamolodchikov [90]. More recent related work in this direction includes [47, 49, 91–94]. The solution to the inverse monodromy problem in ii) can be represented as power series in $z$ having coefficients which can be represented using continued fractions [47].

In order to show that the conformal blocks have a limit of the required form (116) one may start from the series expansions, use the AGT-conjectures, and analyse the limit $b^2 \to \infty$ using cluster expansions [10, 95].

One may next consider the conformal blocks $\mathcal{F}_{\gamma, \iota}(\alpha; y, y', z)$ modified by insertion of the degenerate representations $V_{-1/2b}$ at a points $y$ and $y'$. Such conformal blocks satisfy the BPZ equations (76). To leading order in $b^{-2}$ one can solve these equations in the form

$$\mathcal{F}_{\gamma, \iota}(\alpha; y, y', z) = \exp(-b^2 \mathcal{Y}_\gamma(\lambda, z)) Z^{L(4)}_{1-\mathrm{loop}}(z) \chi_i(y) \chi_{i'}(y') \big(1 + \mathcal{O}(b^{-2})\big),$$
(120)

if $\iota = (i, i')$, the functions $\chi_i(y)$, $i = 1, 2$ being linearly independent solutions to the oper equations $(\partial_y^2 + t(y)) \chi(y) = 0$, provided that the function $\mathcal{Y}_\gamma(\lambda, z)$ is related to $t(y)$ via (118), with $t(y)$ being of the form (117).

One may, on the other hand apply the definition of Verlinde line operators described in Section 8 to the conformal blocks $\mathcal{F}_\gamma(\alpha; z)$. The limit $b \to \infty$ of conformal blocks modified by Verlinde line operators can be computed in two ways. One may, on the one hand, use the representations in terms of finite difference operators discussed in Section 8, giving an expression containing the derivative on the left side of equation (119). Another way to compute this limit uses the representation of the Verlinde line operators in terms of the modified conformal blocks $\mathcal{F}_{\gamma, \iota}(\alpha; y, y', z)$. The computation of quantum monodromies underlying the definition of Verlinde line operators reduces to the computation of classical monodromies of the oper $\partial_y^2 + t(y)$ in the limit $b \to \infty$. The comparison yields (119), completing the argument.

### 9.2.2 Saddle point analysis of Liouville correlation functions

Recall that we may focus on the limit $z \to 0$. Taking into account the asymptotic behaviour (34) of $\mathcal{Y}_\gamma$, one sees that the asymptotic behaviour of the integrand in (105) has the form

$$\exp\left(-b^2 \Big[\big(\delta(\lambda) - \delta(\lambda_1) - \delta(\lambda_2)\big) \log |z|^2 + \mathcal{Y}_0(\lambda) + j\lambda + \mathcal{O}(z)\Big]\right),$$
(121)

using the notations $\alpha - \frac{Q}{2} = \mathrm{i} b \frac{\lambda}{4\pi}$. The integrals in (105) will therefore be dominated by a saddle point $\lambda_j \equiv \lambda_j(z, \bar{z})$ determined by the equation

$$2 \frac{\partial}{\partial \lambda} \mathrm{Re}\big[\mathcal{Y}_\gamma(\lambda, z)\big] \Big|_{\lambda = \lambda_j(z, \bar{z})} = j.$$
(122)

Recalling that $\kappa(\lambda, z) = 4\pi \mathrm{i} \frac{\partial}{\partial \lambda} \mathcal{Y}_\gamma(\lambda, z)$ we see that (122) is equivalent to $\mathrm{Im}(\kappa(\lambda_j, z)) = 2\pi j$. In Section 4.6.3 we have argued that there exists a unique solution of the conditions (36) which can be identified with the conditions (122) determining the saddle points of the integral (105) if we choose $n = 0$ and $m = 2j$ in (36).

In this way we are led to the conclusion that the leading behaviour of the correlation functions (105) takes the form

$$F^{(4)}_{\Lambda_j}(y, z) \sim \exp\big(-2b^2 \mathrm{Re}\big[\mathcal{Y}_\gamma(\lambda_j(z, \bar{z}), z)\big]\big) \chi_j(y, \bar{y}) Z^{L(n)}_{1-\mathrm{loop}}(z) \big(1 + \mathcal{O}(b^{-2})\big),$$
(123)

where we are using the notation $\Lambda_j = j\gamma$, and $\chi_j(y, \bar{y}) \equiv \chi_j(y, \bar{y}; z, \bar{z})$ satisfies the complex conjugate pair of oper equations

$$\left(\partial_y^2 + t_j(y)\right)\chi_j(y, \bar{y}) = 0, \qquad \left(\bar{\partial}_{\bar{y}}^2 + \bar{t}(\bar{y})\right)\chi_j(y, \bar{y}) = 0, \tag{124}$$

with $t_j(y) \equiv t_j(y; z, \bar{z})$ of the form (30), and parameter $K = z E_j(z, \bar{z})$ determined by

$$E_j(z, \bar{z}) = \frac{\partial}{\partial z}\mathcal{Y}_\gamma(\lambda, z)\bigg|_{\lambda = \lambda_j(z, \bar{z})}. \tag{125}$$

It follows that the functions $\chi$ in (123) coincide with the functions $\chi_{0,2j}$ introduced in (40).

# 10 Outlook

## 10.1 Higher rank generalisations

Although the analysis of this paper was specific to the quantum field theories having moduli spaces of vacua represented by $SL(2)$ Higgs bundles, we believe that many of the lessons we have learned can be generalized to wider classes of quantum field theories and defects in various dimensions. Perhaps confusingly, there are multiple constructions connecting complex integrable systems to quantum field theories in various dimensions. Fortunately, the moduli spaces $\mathcal{M}$ of Higgs bundles on a Riemann surface $C$ with ADE gauge group $G$, possibly modified by various types of punctures,[30] are compatible with all of these constructions. In this Section outline some of these conjectures, adopting a 2d CFT perspective.

The minimal QFT perspective involves a collection of non-chiral, non-compact 2d CFTs: the WZW models with target $G/G_c$. These 2d CFTs depend on a level $\kappa$ and admit vertex operators we associate to punctures. They have chiral and anti-chiral Kac-Moody currents at level $\kappa$. The partition function $Z_\kappa$ is a single-valued twisted half-density on an appropriate space Bun of $G$-bundles on $C$ with extra structure at the punctures. The twist is controlled by the level $\kappa$.

The 2d CFTs are also endowed with a collection of "Verlinde" topological line defects, labelled by irreducible representations of $G_c$. More precisely, they are labelled by objects in the Kazhdan–Lusztig category $\mathrm{KL}_\kappa[G]$. The Riemann surface $C$ can be decorated further by a network $\ell$ of such lines, giving rise to a modified partition function $Z_{\kappa;\ell}$. These are also single-valued twisted half-densities on Bun.

The 2d CFTs have a standard $\kappa \to \infty$ semi-classical limit. The equations of motion of the model associate to a point in Bun the corresponding unitary flat $G_c$ connection. The $\ell$ insertion simply read off the monodromy data of that connection.

The critical level limit $\kappa \to \kappa_c$ is expected to have special properties, in general. These properties should be made evident by relations to the 2d Toda theories. Recall that the regular quantum Drinfeld-Sokolov (qDS) reduction maps a Kac-Moody algebra to the corresponding W-algebra $W_\kappa^G$. The Toda theories are non-chiral, non-compact 2d CFTs with $W_\kappa^G$ algebra symmetries. One may in particular expect that there should exist non-chiral versions of the qDS reduction, relating integral transforms of the partition functions $Z_{\kappa;\ell}$ to the corresponding partition functions $Z_{\kappa;\ell}^W$ of the relevant Toda theories, possibly decorated by degenerate fields. A stronger conjecture is that a quantum Separation of Variables transformation exists, reconstructing $Z_{\kappa;\ell}$ as an integral transform of $Z_{\kappa;\ell}^W$ over a space of positions of degenerate punctures, see [98] for recent results in this direction.

---

[30]The only sharp characterization we know of $P$ is that the punctures should admit a lift to half-BPS co-dimension two defects in the six-dimensional SCFTs which are the ultimate source of the various gauge theory constructions [96,97].

A crucial property of $W_\kappa^G$ is the self-duality $(\kappa - \kappa_c) \to (\kappa - \kappa_c)^{-1}$. This implies that the critical level limit $\kappa \to \kappa_c$ is always related to a semi-classical limit for the corresponding Toda CFTs. In that limit, the Verlinde lines labelled by $\ell$ lines become "heavy" and strongly affect the outcome/semi-classical saddle. The degenerate fields appearing in the qDS reduction and possibly SoV, on the other hand, will become "light" and have a finite limit to functions of the semi-classical saddle. Accordingly, the conjectural quantum SoV transformation should have a finite limit of the form $Z_{\kappa;\ell} \sim e^{\frac{1}{\kappa - \kappa_c} S_\ell} \psi_\ell + \cdots$, where the wave-functions $\psi_\ell$ are half-densities on Bun produced by the classical SoV transform. Both $\psi_\ell$ and $S_\ell$ depend on the choice of $\ell$ in a way that might display a dependence on the given regime in parameter space through the choice of saddle points dominating in different regimes.

In the critical level limit, the Kac-Moody algebra develops a large center [99], realized as the quantum Hitchin Hamiltonians acting on sections on Bun [8]. This includes the Sugawara element, which is the limit of $(\kappa - \kappa_c)T$, where $T$ is the stress tensor. Although the stress tensor is modified by the qDS reduction, the divergent parts of the stress tensor and other similar generators matches the semi-classical limits of the $W_\kappa^G$ generators.

As a consequence, the coefficient $\psi_\ell$ above must be an eigenfunction of the quantum Hitchin Hamiltonians, with eigenvalues given by the values of the $W_\kappa^G$ determined by the semiclassical saddle. We thus get a family of eigenfunctions labelled by the networks $\ell$ of lines valued in $KL_\kappa[G]$, i.e. elements of the Skein algebra associated to $C$, $G$ and the punctures.

Up to this point, our conjectures are based on reasonably well-understood aspects of the WZW CFTs, together with the self-duality of Toda CFTs. These are naturally defined for positive level $\kappa$ and admit conjectural analytic continuation away from the $\kappa < \kappa_c$ negative real axis. Next, we specialise to the unusual choice $\kappa \in \kappa_c + i\mathbb{R}$, where the space of twisted half-densities on Bun is equipped with a hermitian inner product and can be completed to an Hilbert space $\mathcal{H}_\kappa$.

One basic conjecture is that $Z_{\kappa;\ell}$ is normalizable on Bun. Expected properties of inner products $\langle Z_{\kappa;\ell} | Z_{\kappa;\ell'} \rangle$ are discussed at length in a companion paper [25]. From a 2d CFT perspective, a proof of this statement requires a very careful analysis of the behaviour of the partition function near loci in Bun where it becomes singular. Alternatively, one could attempt to verify unitarity of a quantum SoV kernel and study the normalizability of the wavefunction in the configuration space of degenerate punctures. In [25] the authors invoke a chain of dualities mapping the inner product to the Schur index of a theory of class $\mathcal{S}$.

In the critical level limit, we learn that $\psi_\ell$ itself is a normalizable eigenfunction representing a part of the spectrum of the quantum Hitchin Hamiltonians. A further conjecture is that $Z_{\kappa;\ell}$ and thus $\psi_\ell$ give bases in the respective Hilbert spaces.

## 10.2 Real generalizations

Two-dimensional CFTs admit a rich collection of conformal boundary conditions. The $G/G_c$ WZW model, in particular, admits boundary conditions which relate the holomorphic and anti-holomorphic Kac-Moody currents by some anti-linear map $\tau$. The theory of these boundary conditions [100, 101] and of cross-caps [102] has been mostly developed for $G = SL(2)$. It generalizes the rational WZW case [103] in a manner analogous to what happens in Liouville theory [104–106]. The Separation of Variables approach is still applicable [107].

Given a Riemann surface $C$ with boundaries decorated in such a manner, we can associate to it a partition function $Z_\kappa$ which is a twisted half-density on a space of "real" bundles $Bun_\mathbb{R}$. Assuming we also understand the interplay between boundaries and topological lines, we can also produce a collection of partition functions $Z_{\kappa;\ell}$.

The logic of the previous analysis will apply verbatim to this situation, leading to Toda theories with boundaries and to normalizable eigenfunctions of quantum Hitchin Hamiltonians. The main extra piece of information needed for a detailed analysis are the dictionaries between WZW and Toda boundaries and the self-duality properties of boundary conditions.

# Acknowledgments

Preliminary versions of some of the results presented in Section 4, and a discussion of their applications in the context of the analytic Langlands correspondence have been described in the PhD thesis of Troy Figiel [28]. The existence of a relation between correlation functions of the $H_3^+$-WZNW model modified by Verlinde line operators and grafting established in our Section 9 has first been proposed in this thesis.

The authors would like to acknowledge inspiring discussions with P. Etingof, B. Feigin, E. Frenkel, and D. Kazhdan.

**Funding information** The work of J.T. was supported by the Deutsche Forschungsgemeinschaft under Germany's Excellence Strategy EXC 2121 "Quantum Universe" 390833306.

# A Real opers with more general types of singularities

## A.1 Real opers with more general types of punctures

The discussion of grafting can be readily extended to the case of a surface $C$ with punctures.

For a regular puncture at $z_a$, $t(z)$ has a double pole with a coefficient

$$\frac{1}{4} - \frac{1}{4}(k_a + 2it_a)^2, \tag{A.1}$$

controlled by the parameters $t_a$ and $k_a$ of the principal series representation there. Notice that $k_a$ can be half-integral depending on the global form of the gauge group.

Solutions of the oper differential equations will behave as

$$(z - z_a)^{\frac{1}{2} \pm \frac{k}{2} \pm it}, \tag{A.2}$$

so that $\phi$ behaves as

$$c(z - z_a)^k |z - z_a|^{1-k+2it} + \text{c.c.}, \tag{A.3}$$

for some constant $c$ and the local coordinate behaves as

$$(z - z_a)^{k_a + 2it_a}. \tag{A.4}$$

If $k_a = 0$, this is real along a semi-infinite collection of circles converging to $z_a$ as a geometric series. The parameter $t_a$ controls the density of the sequence in the radial direction. If $k_a \neq 0$, the zero locus of $\phi$ includes $2k_a$ curves spiraling into $z_a$.

If all $k_a = 0$, the zero locus minus the components wrapping around a single puncture forms what is can be reasonably called an half-integer measured lamination on the punctured Riemann surface. We still have a notion of uniformizing oper: the real oper which only has the semi-infinite sequence of circles around each regular puncture. This is a well-known construction of hyperbolic metrics with geodesic boundaries of fixed length. Grafting along closed geodesics then adds extra closed curves with a more general homotopy.

We can also connect the standard grafting operation to a form of gluing along regular punctures. First, imagine cutting some $C$ along a geodesic, but grafting in an infinite sequence of new lines of zeroes. The result will be a real projective structure on a surface $C'$ obtained from $C$ by replacing an handle with two regular punctures, with a parameter $t$ which encodes the length of the original geodesic, i.e. the monodromy eigenvalue for the connection around the geodesic we cut.

If some $k_a \neq 0$, the story is somewhat different, as the zero locus now includes open lines ending on punctures. We can define a notion of *relative* half-integer measured lamination, including both closed curves and open curves, with a specified number $k_a$ of curves ending at a given puncture. We still lose some of the nice features we discussed above: there is no natural notion of "minimal" real oper analogous to the uniformizing oper and grafting only allows one to add closed lines. We thus are left with no proof of the following natural conjecture: for every $t_a$ and $k_a$, each relative lamination labels exactly one real oper.

## A.2 Irregular singularities

One can also discuss irregular punctures. At an irregular puncture, we have a behaviour

$$e^{\pm(z-z_a)^{-\mathrm{rk}_a}+\cdots}. \tag{A.5}$$

The natural generalization of the single-valuedness condition on $\phi$ is that $\phi$ should not grow exponentially fast at infinity. This imposes a reality condition on the Stokes data of the flat connection. Then $\phi$ behaves roughly as

$$c e^{(z-z_a)^{-\mathrm{rk}_a}-(\bar{z}-\bar{z}_a)^{-\mathrm{rk}_a}\cdots} + \text{c.c.}, \tag{A.6}$$

and the local coordinate as

$$e^{2(z-z_a)^{-\mathrm{rk}_a}+\cdots}, \tag{A.7}$$

which is real along $2\mathrm{rk}_a$ infinite collections of lines approaching $z_a$ along the rays where $(z-z_a)^{\mathrm{rk}_a}$ is real, i.e. within specific Stokes sectors of the irregular puncture.

Semi-infinite collections of open lines will thus stretch between consecutive Stokes sectors. Additionally, some finite collection of open lines may stretch between non-consecutive sectors or between different regular punctures. We can define irregular versions of half-integer measured lamination by including such non-trivial open curves ending at Stokes sectors. Irregular punctures of integral rank are labelled by "formal monodromy" parameters $k_a$ and $t_a$. One can argue that $2k_a$ control the difference between non-trivial curves ending at even or odd Stokes sectors.

The data of laminations in the presence of punctures may seem intricate. Remarkably, in all the situations described above, the lamination data coincides with the data one would employ to label holomorphic functions (or sections of line bundles) on the corresponding character varieties. As we saw in the main text, this is no coincidence and we expect it to remain remain true even for general gauge group $G$.

A simple example of an irregular real oper, for $t(z) = -z$, is

$$\mathrm{Ai}(z)\mathrm{Bi}(\bar{z}) + \mathrm{Ai}(\bar{z})\mathrm{Bi}(z), \tag{A.8}$$

leading to the pattern of zeroes in Figure 9.

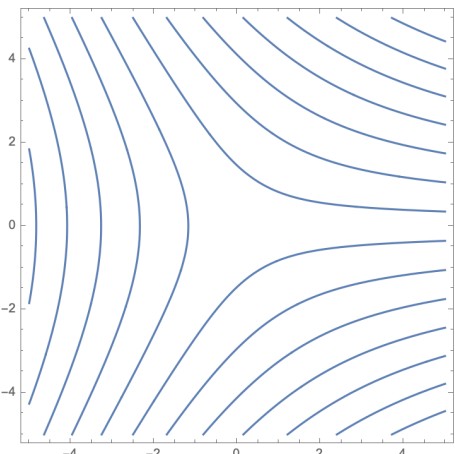

Figure 9: The pattern of zeroes for the "Airy" real oper.

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
