# Peer review of "Quantum Analytic Langlands Correspondence"

_SciPost Physics, doi:SciPost Phys. 18, 144 (2025)_

## Round 1 · Referee Report · Anonymous (Referee 1) · 2025-2-10

Strengths

  1. This paper introduces an important and intriguing extension - the quantum deformation - of the analytic Langlands correspondence.

  2. The physical account of the (quantum) analytic Langlands correspondence is also well-analyzed in 2d/3d/4d perspectives, which would help readers unfamiliar with the subject.

Weaknesses

  1. While the paper is well-structured, some sections are dedicated to reviewing results from previous works, which extends its length. However, I do not see this as a weakness, as it ensures the paper includes all the necessary details which would eventually help readers.

Report

The results of the paper provide interesting development of the quantum deformation of the analytic Langlands correspondence. I support publication of the paper.

Requested changes

  1. It could be helpful to include a figure in Section 4.4.1 or 4.4.2 intuitively showing how the FN coordinates are defined.

  2. In Section 2.3, the authors discuss the physical account of the Hecke operators. It could be worth mentioning this paper(https://inspirehep.net/literature/2650150) studying the class S theory counterpart.

  3. In Section 4.6.2, the authors present the twist coordinate $\kappa(\lambda, z)$ for the Heun oper in the limit $z \to 0$ at the order of $\log z$, following [48,49]. It may also be worth mentioning this paper (https://inspirehep.net/literature/1678845) for this small z analysis from the class S theory perspective.

  4. (6.18) is called the Verlinde line operator insertion in the sense that 1) $H_3 ^+$-WZNW correlation function admits an integral transform presentation (7.9), in which the integrand is given by the Liouville correlation function; 2) The Verline line operators for that Liouville correlation function produces the additional insertion in the integrand of (6.18). Although this is explained well at the beginning of Section 8, it could be helpful to restate this explicitly by adding an equation at the end of Section 8.6.

  5. Similar to the comment 1, a figure in Section 8.2.2 or 8.2.3 may be helpful.

  6. As authors summarized in page 17, the analytic Langlands correpsondence involved Hecke operators sharing the eigenspace with Hitchin Hamiltonians. Could there be any comment on the role of Hecke operators after the quantum deformation (or just Hecke modification)?

  7. In Section 10.1, the authors discuss the higher-rank generalization. It may be worth mentioning this paper (https://inspirehep.net/literature/2760389) studying the higher-rank version of the integral transformation (7.9) in the class S theory perspective.

Recommendation

Publish (surpasses expectations and criteria for this Journal; among top 10%)

---

## Round 1 · Referee Report · Anonymous (Referee 2) · 2025-3-30

Strengths

This is actually a magnificent paper. The relation that the authors make between quantization of SL(2,C) Chern-Simons theory and the H_3^+ WZW model is pathbreaking. The same is true of their generalization of the ``analytic'' form of the geometric Langlands correspondence to the ``quantum'' case.

As well as being a pathbreaking paper, this is a very difficult one. A lot of different things go into it. Indeed, I described myself as ``expert,'' but realistically I am only an expert on some aspects of the background and quite a beginner on other aspects. SciPost would have a lot of trouble to find a referee who was really knowledgeable on all of the things that go into this paper.

Weaknesses

No relevant weakness. The authors have done a laudable job of trying to explain some of the background needed for this paper. What they've offered is certainly less than a complete explanation of the necessary background. However trying to give a more complete account of the background would certainly have made the paper even longer.

The reason that I give the paper ``high'', not ``top'' grades for clarity is that indeed there are places where the explanation of the background could have been more complete.

Report

Most definitely this paper should be published.

Requested changes

I don't really want to recommend changes, but for the benefit of the authors, I'd like to mention that the work of Beilinson and Drinfeld to quantize Hitchin's integrable system was undoubtedly stimulated by Hitchin's prior work doing this for SL_2.

Recommendation

Publish (surpasses expectations and criteria for this Journal; among top 10%)

---

## Editorial Decision

published